Registered report 

psychology

the spread of COVID-19, moral identity, moral self-image, labelling, persuaded communications

**Author for correspondence:**
Fumiya Yonemitsu
e-mail: y.fumiya.0408@gmail.com

# Warning 'Don't spread' versus 'Don't be a spreader' to prevent the COVID-19 pandemic

Fumiya Yonemitsu[1,4], Ayumi Ikeda[1,4], Naoto Yoshimura[1,4], Kaito Takashima[1], Yuki Mori[1], Kyoshiro Sasaki[5], Kun Qian[2] and Yuki Yamada[3]

[1]Graduate School of Human-Environment Studies, [2]Institute of Decision Science for a Sustainable Society, and [3]Faculty of Arts and Science, Kyushu University, Fukuoka, Japan
[4]Japan Society for the Promotion of Science, Tokyo, Japan
[5]Faculty of Informatics, Kansai University, Osaka, Japan

FY, 0000-0001-8774-4499; AI, 0000-0002-1688-2875;
NY, 0000-0002-2656-4432; KS, 0000-0002-5496-3748;
KQ, 0000-0002-0625-1834; YY, 0000-0003-1431-568X

The coronavirus disease 2019 (COVID-19) outbreak is threatening not only health but also life worldwide. It is important to encourage citizens to voluntarily practise infection-prevention (IP) behaviours such as social distancing and self-restraint. Previous research on social cognition suggested that emphasizing self-identity is key to changing a person's behaviour. The present study investigated whether reminders that highlight self-identity would be effective in changing intention and behaviour related to the COVID-19 outbreak, and hypothesized that those who read reminders highlighting self-identity (Don't be a spreader) would change IP intention and behaviour better than those who read 'Don't spread' or no reminder. We conducted a two-wave survey of the same participants with a one-week interval, during which we assigned one of three reminder conditions to the participants: 'Don't spread' (spreading condition), 'Don't be a spreader' (spreader condition) and no reminder (control condition). Participants marked their responses to IP intentions and actual behaviours each week based on the Japanese Ministry of Health, Labour and Welfare guidelines. While the results did not show significant differences between the conditions, the *post hoc* analyses showed significant equivalence in either IP intentions or behavioural scores. We discussed the results from the perspective of the effect size, ceiling effects and ways of manipulation checks as future methods with more effective

persuasive messaging. Following in-principle acceptance, the approved Stage 1 version of this manuscript was pre-registered on the OSF at https://doi.org/10.17605/OSF.IO/KZ5Y4. This pre-registration was performed prior to data collection and analysis.

# 1. Introduction

## 1.1. Background and main research question

The spread of the coronavirus disease (COVID-19) has caused an acute public health crisis that is threatening lives globally. To tackle this dire situation and help stop the spread of the disease, governments have been encouraging citizens to change their behaviours by providing specific guidelines (such as social distancing and refraining from venturing out). However, these guidelines are likely to be ignored by some citizens. Furthermore, it is legally impossible for some governments (for example, the Japanese government) to forbid people to leave their homes, suspend attendance at work or school, or lock down an entire city. For example, the Japanese government declared a national state of emergency on 7 April 2020, but only requested that people follow the suggested control measures on a voluntary basis. There are no laws to force citizens to follow these measures. Therefore, behavioural modifications depend entirely on an individual's ethical beliefs. Perhaps for this reason, the number of COVID-19 infections in Japan increased daily, with more than 100 cases every day from 31 March to 6 May 2020. Therefore, psychologically based persuasive communication is required to effectively change individual attitudes and behaviours, thereby encouraging people to conform to the guidelines.

The main research question answered in this study is whether the difference in infection-prevention (IP) reminders (Don't spread/Don't be a spreader) can influence Japanese citizens to modify their behaviours. In other words, the self-identity-related 'Don't be a spreader' would be more likely to stop readers from engaging in high-risk behaviours that promote the spread of infection. IP was measured from two perspectives—behavioural intentions and reported behaviour of IP, using the IP-intention and IP-behaviour scale. Research on social cognition has demonstrated that the presentation of a specific word can change a person's behaviour. For example, slightly different descriptions in instructions can change the deterrent rate of the reader's unethical behaviour: A description of 'Don't be a cheater' reduced cheating more than 'Don't cheat' [1]. Their findings suggest that slogans with suffixes that represent agents (-er) modulate behaviour.

The virtue of manipulating the wording of instructions lies in its simplicity and general versatility. In addition to highlighting self-identity, a recent paper reported that manipulating the expressions of messages to appeal to citizens' emotions is effective in promoting behaviour to self-isolate [2]. However, it is important to find more useful methods, given the circumstances in different countries such as cultural, linguistic and legal differences. Making self-identity salient in messages is also one of the simplest ways of intervening with large numbers of people. Thus, we investigated the effectiveness of messages highlighting the role of self-identity in changing people's behaviour.

Generally speaking, spreading infection is negative and an undesirable behaviour. Nominalizing the verb indicating such a behaviour with suffixes that represent agents (-er) creates a strong link between an identity and negative self-image. In other words, people who read the reminder 'Don't be a spreader' come to consider infection spreader as part of their self-identity. As this threatens positive self-image, the avoidant motivation to circumvent such labels is driven to protect one's image, which makes ignoring the reminder difficult. As a result, people who read the reminder are more likely to modify their behaviours to avoid spreading infection.

In addition, the effectiveness of this intervention has been repeatedly verified in other studies [3,4]. In particular, Savir & Gamliel [4] successfully replicated the study by Bryan et al. [1]. Before Bryan et al. [1], Bryan and colleagues investigated the effect of highlighting self-identity reminders on voting behaviours (voting versus being a voter), finding that the reminder 'be a voter' elevated actual voter turnout [5]. On the other hand, another intervention is to evoke mortality in messages based on terror management theory [6]. However, studies [7,8] have also pointed out that mortality salience effects are small or not robust compared to the original study. Accordingly, in the present study investigating IP reminders, we avoided the use of mortality salience as its manipulation can be considered dangerous. Therefore, we considered highlighting self-identity reminders as an obviously effective way to promote behavioural change that can also be applied in the COVID-19 pandemic.

Our study aimed to show whether it may be possible to reduce the spread of COVID-19 using this stimulus manipulation. This is important because such simple changes in language may still serve to

encourage people to change their behaviour. If effective, this method might help to successfully contain the pandemic even in countries where 'a lockdown could not be enforced by external forces such as the army and the police'. Therefore, our research is suggestive regarding the practical implications.

## 1.2. Hypotheses

The main hypotheses of this research are as follows:

— Hypothesis 1: If highlighting self-relevance corrected our intention, the change of the IP-intention score for behavioural intentions from the first wave (baseline) to the second wave would be significantly larger in the spreader condition than in the spreading and control conditions.
   H0: If highlighting self-relevance did not influence our intention, the change of the IP-intention score for behavioural intentions from the first wave (baseline) to the second wave in the spreader condition would not differ from either the spreading or the control condition.
— Hypothesis 2: If highlighting self-relevance corrected our behaviour, the change of the IP-behaviour score from the first wave (baseline) to the second wave would be significantly larger in the spreader condition than in the spreading and control conditions.
   H0: If highlighting self-relevance did not influence our behaviour, the change of the IP-behaviour score from the first wave (baseline) to the second wave in the spreader condition would not differ from either the spreading or the control condition.

Additionally, as our secondary hypothesis, we predicted that the scores on the perceived vulnerability to disease (PVD) scale would be higher during the COVID-19 pandemic than before the pandemic.

# 2. Method

## 2.1. Participants

We recruited participants via Yahoo! Crowdsourcing (http://crowdsourcing.yahoo.co.jp/). Participants must be able to understand Japanese well and be residents of Japan.[1] We also confirmed their residence in the survey questionnaire. Participants were informed regarding the need to understand Japanese well in the recruitment requirements. In addition, we asked questions about participants' native language and nationality in the demographic information.

As ethical considerations, we conducted our survey carefully to ensure informed consent. We stated that the data obtained from these surveys would be anonymized so that it could be made public without identifying individuals. Moreover, we ensured that the survey requests stated that the survey would be conducted anonymously with no personal information, such as COVID-19-related data or email addresses, being linked to it. As mentioned in Data analysis, we performed two one-way analyses of variance (ANOVA) with the reminder condition (spreading, spreader and control conditions) for both different dependent variables (i.e. change of IP-intention and IP-behaviour scores). Guo *et al.* [9] calculated the effect size of the original study [1] for a value of Cohen's $f = 0.302$. However, the small sample size might overestimate the effect size. Therefore, as a replication convention [9,10], we halved the effect size of the original experiment (Cohen's $f = 0.151$), and used $G^{*}$ Power [11] to conduct a power analysis (Cohen's $f = 0.151$, $\alpha = 0.05$, $1 - \beta = 0.99$, number of groups = 3), as a result of which we set $N = 942$ as the required sample size. Moreover, considering that many participants are often excluded in online surveys because of satisficing [12–14], and the dropout rate from the second wave, we have set almost double the required sample size ($N = 1890$) as the maximum sample size. If the number of final participants was below the required sample size after excluding participants according to the criteria detailed in Data exclusion criteria, we planned to collect data from additional participants to reach the required sample size. Additionally, for ethical reasons, even if more than 1890 people participated in the survey, we would include the excess data in the analysis. Furthermore, based on our maximum sample size, we performed a sensitivity analysis to calculate the effect size in the case that statistical power was 0.80 ($\alpha = 0.05$, $1 - \beta = 0.80$, $N = 1890$, number of groups = 3), yielding an estimate of Cohen's $f$ of 0.071.

[1]In the pre-registered procedure, we planned to recruit participants who are residing in Tokyo. However, fewer people accessed our survey page than we expected; only 260 people participated in our survey for the 2 days. Thus, it seemed difficult to collect the planned sample size (i.e. $N = 942–1890$) stated in the pre-registered procedure. Hence, prior to data analysis, we expanded the survey area from Tokyo to all prefectures in Japan. We executed this deviation after approval from the action editor on 13 June 2020.

**Table 1.** Items in the IP-intention scale based on the guidelines provided by the Japanese Ministry of Health, Labour and Welfare.

| no. | item |
| --- | --- |
| 1 | I will avoid going out as much as possible. |
| 2 | When I have to go out, I will wear a mask. |
| 3 | I will avoid the 'Three-Cs' (closed spaces with poor ventilation, crowded places with many people nearby and close-contact settings such as close-range conversations). |
| 4 | I will ventilate regularly. |
| 5 | I will not speak loudly. |
| 6 | I will avoid conversations where people are within hand-to-hand distance. |
| 7 | I will follow proper coughing etiquette (using a mask, tissue, handkerchief, sleeve, inside of elbow, etc. to cover my mouth and nose when I cough or sneeze). |
| 8 | I will wash my hands often. |
| 9 | I will not touch the mucous membranes of my eyes, nose and mouth. |

**Table 2.** Items in the IP-behaviour scale based on the guidelines provided by the Japanese Ministry of Health, Labour and Welfare. (**R**) at the end of the items indicates a reverse-code item.

| no. | item |
| --- | --- |
| 1 | How often did you go out? (**R**) |
| 2 | How often did you avoid going to the 'Three-Cs' place (closed spaces with poor ventilation, crowded places with many people nearby and close-contact settings such as close-range conversations)? |
| 3 | How often did you ventilate your home? |
| 4 | How often were you careful not to speak loudly? |
| 5 | How often did you have conversations where people were within hand-to-hand distance? (**R**) |
| 6 | How often did you wash your hands? |
| 7 | How often did you touch the mucous membranes of your eyes, nose and mouth? (**R**) |

There were three reminder conditions: 'Don't spread' (spreading condition), 'Don't be a spreader' (spreader condition) and no reminder (control condition). We would recruit 630 participants for each condition (1890 people in total).

## 2.2. Measures

To measure behavioural intentions and reported behaviour of IP, we developed two scales based on the COVID-19 IP policy of the Japanese Ministry of Health, Labour and Welfare [15]. We defined the scale to measure behavioural intentions and reported behaviours of IP as the IP-intention scale and the IP-behaviour scale, respectively. These scales consisted of nine and seven items, respectively (tables 1 and 2). The items of the IP-intention scale were scored on a 7-point scale ranging from 1 (*strongly disagree*) to 7 (*strongly agree*). The IP-behaviour scale was scored on 7-point scale ranging from 1 (*not at all*) to 7 (*very often*). Item numbers 1, 5 and 7 of the IP-behaviour scale were the reverse-code items.

In addition, the side hypothesis was that there was an increased concern for infection during the COVID-19 pandemic compared to life before COVID-19. Here, we used the Japanese version of the PVD scale, which comprises 15 items [16,17]. These items were scored on a 7-point scale ranging from 1 (*strongly disagree*) to 7 (*strongly agree*). The PVD scale reflects 'perceived infectability', which is related to the beliefs of one's own susceptibility to infectious diseases, and 'germ aversion', which is related to an awareness of discomfort in situations with a high likelihood of infection from a pathogen. We measured the PVD scale scores and statistically compared the PVD scale scores collected in the present experiment with those collected before COVID-19 situation [18] as a check on the growing concern for infection. The comparison of the PVD scores between the current and non-pandemic situations should advance the discussion of the effect of self-related reminders based on the comparisons.

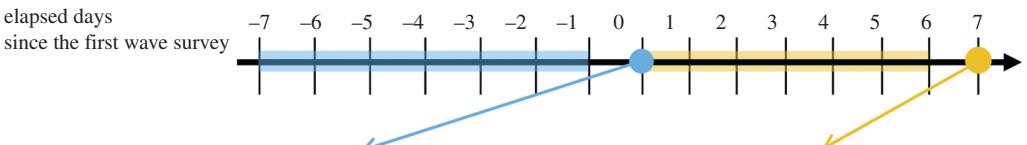

**Figure 1.** The timeline of the survey schedule for sending the form for the second wave.

## 2.3. Materials and procedure

We conducted a two-wave online survey for the same participants (figure 1). In the first wave, we asked the participants for their email addresses. A week after they have completed the first wave, we sent them the form for the second wave via email. In the first wave, we initially asked participants to respond to the IP-intention scale, IP-behaviour scale and the PVD scale, and then asked them to provide demographic information (age, sex and residence) as well as their email addresses on the face sheet. Participants' intention of upcoming behaviours (regardless of when they behave) was assessed in the IP-intentions scale, whereas their actual behaviours in the past week were recorded in the IP-behaviour scale.

After completing these questionnaires, we visually presented gratitude messages including one of the IP reminders to the participants in the spreading and spreader conditions, whereas the participants in the control condition were given the gratitude messages without any reminder. The gratitude message was presented on the last page after the completion of the questionnaire using visual images in white letters on a green background (figure 2). The randomized assignment for the three reminders was based on the participant's birthday: Participants whose birthday is the 1st–10th of the month were assigned to the spreader condition, those with the 11th–20th to the spread condition and those with the 21st–31st to the control condition. The presentation order of the items in each scale was randomized across the participants but the execution order of the scales was fixed as follows: the IP-intention scale, the IP-behaviour scale, the PVD scale and demographic information. In the second wave, the procedure was identical to the first with the exception that we did not ask participants to answer their email address and the PVD scale and presented no reminder in any of the conditions.

In addition, we included some questions as quality checks for our survey. We inserted a simple attention check question (ACQ) in the IP-behaviour scale in each survey wave to identify distracted respondents or satisficers [12–14]. The ACQ in the first wave was 'In this survey, how often have you been asked about your blood type so far?' and the ACQ in the second wave was 'How often did you go to Mars?'. The answer to both these ACQs must be '1 (not at all)'. We also set up instructional manipulation checks (IMC) to exclude the participants who did not actually see/read the gratitude messages including the reminder message in the spreader and spreading conditions. The IMC was placed on the same page as the gratitude messages and said, 'Do you like the font used in the above message? Be sure to answer N/A'. We considered non-N/A respondents as not having read the message.

## 2.4. Data analysis

The anonymized dataset of the present study is publicly available at https://osf.io/dc7rs/.

## 2.5. Data exclusion criteria

As mentioned in the Participants section, we excluded the participants whose answers for the questions about native language and nationality are not Japanese and Japan, respectively.[2] In addition, we

---

[2]Although we planned to exclude participants who are not residents of Tokyo, we removed the criterion from the exclusion criteria due to the change.

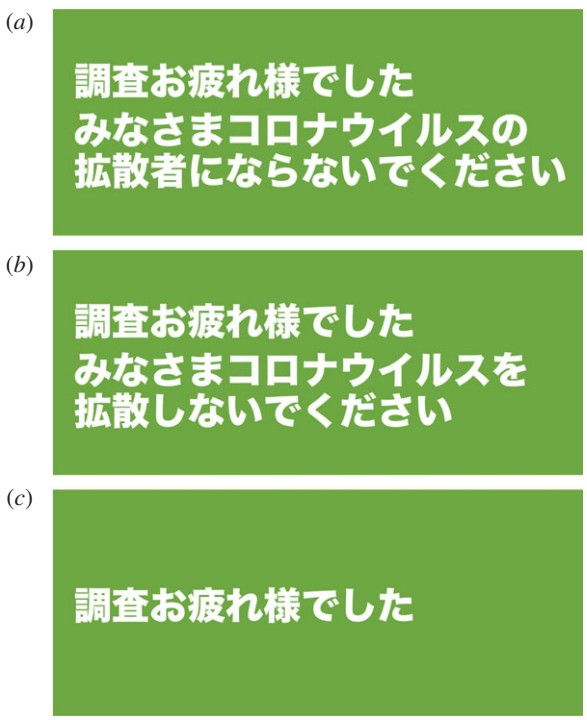

**Figure 2.** Images presenting the reminders. (*a*) indicates the spreader condition, (*b*) the spreading condition and (*c*) the control condition.

excluded participants who made mistakes on the ACQs or IMC. We excluded participants who did not answer '1 (not at all)' for ACQs and N/A for the IMC. Furthermore, we excluded the data of participants who failed to respond within 24 h of the request for the second-wave survey.

## 2.6. Quality checks for providing a fair test

Other than COVID-19, we do not have any modern examples of data from a time requiring major changes in human behaviour where it was not possible to set positive controls for any pandemic effect. In this case, to confirm that there was a significantly growing concern for infection during the COVID-19 pandemic, we used the PVD scale [16,17]. We used a dataset collected before COVID-19 that has been published elsewhere [18]. This dataset is not related to the main purpose of this study, the IP-intention scale, or the IP-behaviour scale, and none of the authors conducted any statistical analysis on this dataset. Therefore, it did not bias our interpretation of the newly acquired data. We statistically compared the PVD scale scores collected in the present study with those collected before COVID-19 [18] using a two-sample *t*-test and reported the *t*-values and *p*-values. If the mean present score was significantly higher, then we would interpret this as indicating that there was an increased concern for infection by COVID-19. We planned to perform equivalence tests with two one-sided tests for equivalence in R (TOSTER; [19]) if there was no significant difference in the PVD scales between the current and pre-pandemic situations. If there was no significant variation, we would interpret this as indicating that the perceived concern for infection was essentially the same as before COVID-19 even under the influence of the COVID-19 pandemic.

## 2.7. Main analysis

First, for each of the conditions, we calculated each participant's individual mean scores in the IP-intention as well as the IP-behaviour scales, for each of the first and second waves. Next, we calculated the changes on the IP-intention and IP-behaviour scale by subtracting the mean score of the first wave from that of the second wave in each condition. When testing the hypotheses, we used the change in the mean scores for each IP-intention scale and IP-behaviour scale as the key dependent variables.

We performed two one-way ANOVAs on the change of the mean scores for each IP-intention scale and IP-behaviour scale under each reminder condition (spreading, spreader and control conditions). We set the alpha level at 0.05 and reported *F*-values, *p*-values and effect sizes ($\eta_p^2$). When the main

effect was significant, we confirmed which pairs showed significant differences, for which purpose we performed multiple comparisons using Scheffé's $F$-test and reported $t$-values, $p$-values and effect sizes (Cohen's $d$). Among the orthogonal contrasts test, Scheffé's $F$-test was more appropriate for contrasting all the conditions because we were interested in the differences between the reminder conditions (spreading, spreader and control conditions) [19].

# 3. Results

In total, 2536 individuals participated in the first and second waves of the experiment. Based on the pre-registered exclusion criteria, we excluded participants who were not Japanese or not native Japanese speakers, provided a wrong answer on the ACQs or an IMC, or failed to respond within 24 h of the request for the second-wave survey. We also excluded participants whose data did not match the criteria but who had responded to the second wave of our survey twice.[3] As a result, we submitted data from 1104 participants (648 males, 456 females, $M_{age} = 46.45$) for further statistical analyses.

## 3.1. Pre-registered analyses

### 3.1.1. The effect of the reminders on the infection-prevention intention

We performed a one-way ANOVA on the change of the mean scores for the IP-intention scale with the reminder condition as a between-participant factor. The results showed that the main effect was not significant ($F_{2,1101} = 0.73$, $p = 0.49$, $\eta_p^2 = 0.001$). We, therefore, performed an equivalence test to determine the equivalences between the spreading and control conditions and set the Cohen's $d$ (value of equivalence bounds) to ±0.5,[4] which revealed that there was a significant equivalence ($t_{747} = 5.633$, $p < 0.001$).

### 3.1.2. The effect of the reminders on infection-prevention behaviour

We performed a one-way ANOVA on the change of the mean scores for the IP-behaviour scale with the reminder condition as a between-participant factor. The results showed that the main effect was not significant ($F_{2,1101} = 0.51$, $p = 0.60$, $\eta_p^2 = 0.0009$). Thus, we performed an equivalence test to determine the equivalences between the spreading and control conditions and set the Cohen's $d$ (value of equivalence bounds) to ±0.5, which established that there was a significant equivalence ($t_{747} = 6.073$, $p < 0.001$) (figure 3).

### 3.1.3. The differences between the PVD scores during and before the COVID-19 pandemic

We calculated each participant's mean scores on perceived infectability and germ aversion. Further, we performed a two-sample $t$-test to confirm the difference between each mean score during the COVID-19 pandemic and before its onset. The mean score on perceived infectability during the pandemic was lower than that prior to its onset ($t_{3177} = 8.49$, $p < 0.001$, Cohen's $d = 0.30$). By contrast, the mean score on germ aversion was higher during the pandemic than before its onset ($t_{3177} = 2.41$, $p = 0.02$, Cohen's $d = 0.09$). In addition, for each, we calculated the means and standard deviation, before and during the COVID-19 pandemic ('pi' and 'ga' indicate perceived infectability and germ aversion, respectively; $N = 1382$, $M_{pi} = 3.83$, s.d.$_{pi} = 1.14$, $M_{ga} = 4.33$, s.d.$_{ga} = 0.98$; $N = 1304$, $M_{pi} = 3.57$, s.d.$_{pi} = 0.59$, $M_{ga} = 4.36$, s.d.$_{ga} = 0.60$). We also calculated the average inter-item correlation for the perceived infectability and germ aversion data before and during the pandemic, which were Ave$_{pi} = 0.46$, Ave$_{ga} = 0.24$; and Ave$_{pi} = 0.48$, Ave$_{ga} = 0.29$, respectively. Please refer to our electronic supplementary material for more detailed information at https://osf.io/dc7rs/files/ (figure 4).

## 3.2. Post hoc analyses

Although we planned to perform only the equivalence tests between the spreading and control conditions, the tests for the other pairs are informative for discussing the salience of the effect of the

---

[3]We had not anticipated that there would be participants responding to the second wave of the survey twice. Thus, this exclusion criteria deviated from the pre-registered method.

[4]The values for the equivalence bounds were not pre-registered but were determined when we performed the analysis.

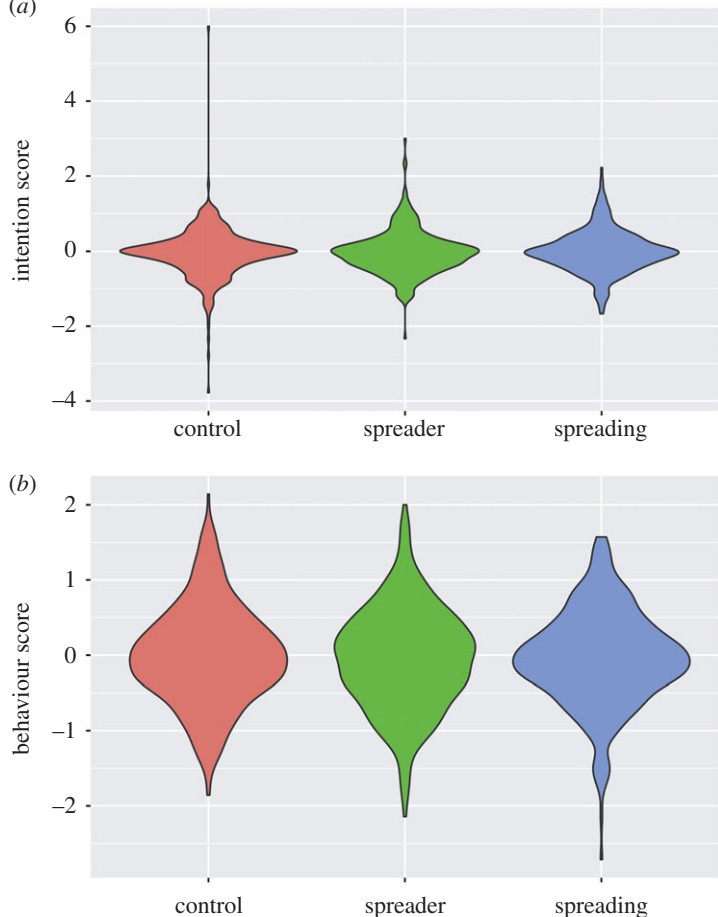

**Figure 3.** Results of the changes in the IP scores from the first wave to the second wave: violin plots for the mean of the changes of the IP-intention scores (*a*) and the IP-behaviour scores (*b*).

reminders. We, therefore, performed an equivalence test on the changes between the mean scores for the IP-intention scale and the remaining condition pairs and set Cohen's *d* (the value of equivalence bounds) to ±0.5. The results showed significant equivalence in both pairs: spreader versus spreading: $t_{693} = 5.89$; $p < 0.001$ and spreader versus control: $t_{762} = 6.39$, $p < 0.001$).

We also performed an equivalence test on the change of the mean scores for the IP-behaviour scale between the remaining condition pairs and established that there was significant equivalence in both pairs (spreader versus spreading: $t_{693} = 6.387$, $p < 0.001$; spreader versus control: $t_{762} = 5.955$, $p < 0.001$).

## 4. Discussion

This study aimed to investigate whether reminders that highlighted self-identity would affect intentions and behaviours relating to the spread of COVID-19. In addition, the results showed that there was no difference between the reminder conditions in both the IP-intention and IP-behaviour scores. As such, there was no significant statistical support for conclusions relating to either Hypothesis 1 or Hypothesis 2. Conversely, the *post hoc* equivalence tests showed that there were significant equivalences between the reminder conditions in both the IP-intention and IP-behaviour scores, indicating that there was no support against either Hypothesis 1 or 2.

Additionally, we compared the PVD scores collected in the present study with those obtained before the COVID-19 pandemic situation. The results showed that perceived infectability was lower during COVID-19 than before its onset. At the same time, germ aversion was higher during the pandemic than before its onset. Thus, the COVID-19 pandemic had an asymmetric influence on the subscales of the Japanese people.

The present study did not show the effect of highlighting self-identity reminders. Why did highlighting of self-identity reminders have little effect on IP intention and behaviour in our study? It was probably because our study had several limitations. One of its limitations was that it had a longer time period until data collection than previous studies [4,9]. In particular, in an original study

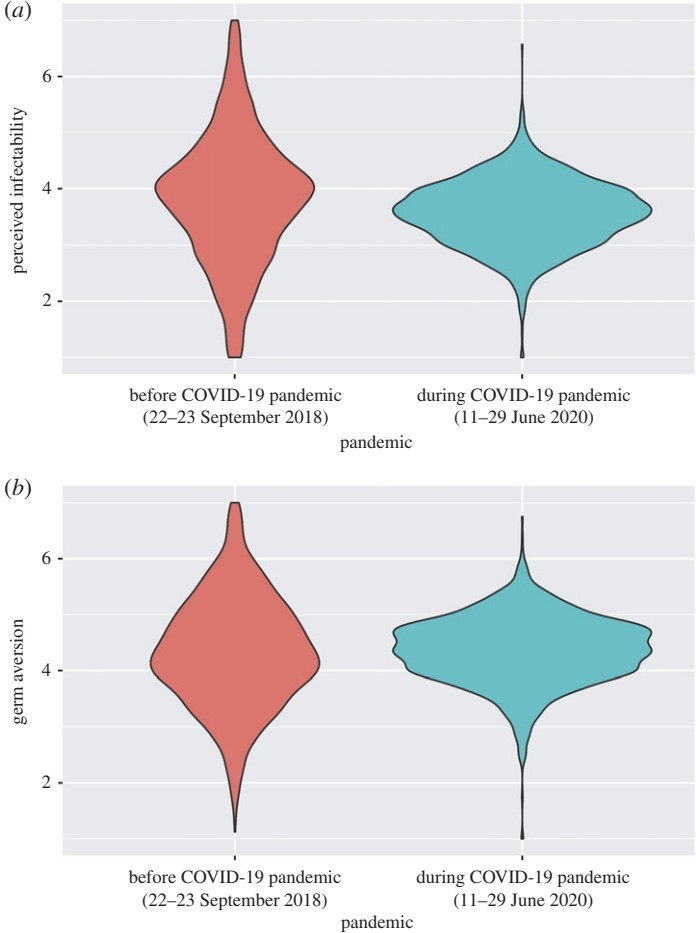

**Figure 4.** Results of the subscales in the PVD scale: violin plots for the means of the perceived infectability score (*a*) and germ aversion score (*b*).

[5], which had shown that people with an emphasis on self-identity were more likely to engage in voting behaviour, experimental manipulation of reminders had been conducted on the day before or the day of the voting, whereas in our study, the participants were required to perform the second wave a week after the experimental manipulation of a reminder. The length of the period in our study may have contributed to the saliency of the reminders' effect. Highlighting self-identity could have been more effective for IP intention and behaviour if we had presented the reminders for seven consecutive days instead of just once, or if we had presented the reminders immediately before the second survey [5]. Moreover, the replication study on highlighting self-identity [4] showed that their effect sizes were small, with an overall estimate of Cohen's *d* being 0.17. The small effect size could have made it difficult to detect the impact of highlighting self-identity in a real-life situation.

Another possibility of the null effect of highlighting self-identity was that ceiling effects could mediate in the present results. Indeed, in each wave, the mean scores on IP intention and behaviour were high; especially, for IP intention, which nearly reached the maximum value of the scale in both the first and second waves (IP intention $M_{1st} = 6.01$ (s.d. = 0.87), $M_{2nd} = 5.98$ (s.d. = 0.84); IP behaviour $M_{1st} = 4.97$ (s.d. = 0.79), $M_{2nd} = 4.94$ (s.d. = 0.78)). Considering all these factors, we speculate that ceiling effects could have occurred in the present study. In this case, there might be little room for the highlighting of self-identity to have an effect on IP intention and behaviour. At this point of time, although it is difficult to specify the factor causing the ceiling effects, one plausible speculation is that a high level of germ aversion, as shown through the PVD scores, might be related with these ceiling effects. At least, an improvement of the measurements might be desirable to properly measure IP intention and behaviour.

Moreover, one might argue that our IMC did not work sufficiently to check whether the participants had carefully read the reminders.[5] If so, there may have been a lot of participants who provided the

---

[5]We thank Reviewer 2 for pointing out this possibility.

correct answer to the IMC although they had not carefully read the reminders. This possibly caused the null effect of the reminders in the present study. To fully eliminate this possibility, it is necessary for further research to establish more effective methods in presenting the reminders or manipulation checks.

In the present study, there were differences in directions between the PVD's two subscales. Specifically, germ aversion was higher during the pandemic than prior to its onset, while perceived infectability was lower during the pandemic than in the pre-pandemic situation. Makhanova & Shepherd [20] investigated whether PVD was linked with responses to COVID-19 mainly for American participants, and established a positive correlation in both the PVD's subscales with the threat to COVID-19. In other words, during the pandemic, the higher the danger of COVID-19, the more they felt that being infected was more likely. This finding seems to be different and one can argue that this discrepancy between the studies is derived from the Japanese culture of having less physical contact. However, Makhanova & Shepherd [20] investigated the PVD only during the pandemic. Furthermore, since to the best of our knowledge, no studies have compared PVD scores during COVID-19 with those in non-pandemic situations; it is too early to conclude as to whether these tendencies of PVD are specific to the Japanese people. There is not enough comparative data relating to this study's PVD scale results; hence it is not possible to identify the causes of the different directions in the subscales and to generalize the tendency. It remains unclear what effects mediated this tendency. Further investigations would be necessary to specify the causes of divergence in the PVD subscales.

Additionally, Japan effectively controlled the spread of infection when we conducted the survey. For example, the number of cases during the survey execution period was in the range of 40 and 268, whereas the number of cases before the survey execution period exceeded 700, and that number after the survey execution period reached a record high (i.e. 1605 cases). Since the number of cases was relatively low during our survey; it is unclear whether the obtained results would be true in the event of the COVID-19 situation being severe. Based on this view, further investigations in more severe situations would be warranted to confirm whether slight differences in linguistic expressions generally have no effect on our IP intentions and behaviours.

While the present study was conducted only with Japanese people; the previous study reported that the effects of the highlighting of self-identity occurred not only in Japan [9] but also in America and Israel [1,4]. Based on these findings in American and Israeli studies, the effects of highlighting self-identity on IP intention and behaviour might occur in these countries; and its null effects in the present study may perhaps have been due to potential factors peculiar to Japan that were related to the severity of COVID-19, the quality of the medical system and so on. Further investigations in countries other than Japan would confirm this speculation.

The present study showed that highlighting self-identity by manipulating linguistic expressions had no impact on IP intention and behaviour. Although both the messages in our stimuli were general, and governments often use similar messages, our findings reveal that our manipulations would not contribute to a real messaging campaign. Then, what should be the effective reminders for encouraging IP intention and behaviour in a real messaging campaign? For example, the repetition of messages might be a good method for a real messaging campaign. Indeed, moderate repetition of messages boosts their agreements [21]. Thus, if our messages, which highlight self-identity, are moderately repeated, they could influence IP intention and behaviour during the COVID-19 pandemic situation. Alternatively, instead of manipulating linguistic expressions, considering other ways to enhance the self-relevance of messages might also be beneficial for a real message campaign. One possible and simple way to enhance the self-relevance of messages is to generate one's own reminder messages. We shall continue to explore and validate methods of high practical significance (e.g. repetitive presentation and self-generation of reminder messages).

Ethics. The present study received approval from the psychological research ethics committee of the Faculty of Human-Environment Studies at Kyushu University (approval number: 2019–034).

Data accessibility. All relevant protocols on this study are freely available in the OSF repository located at https://osf.io/kz5y4. Its anonymized dataset is publicly available at https://osf.io/dc7rs/.

Authors' contributions. F.Y.: conceptualization, data curation, formal analysis, funding acquisition, investigation, methodology, software, visualization, writing—original draft, review and editing. A.I.: conceptualization, data curation, funding acquisition, methodology, software, writing—original draft, review and editing. N.Y.: conceptualization, funding acquisition, methodology, writing—original draft, review and editing. K.T.: conceptualization, writing—original draft, review and editing. Y.M.: conceptualization, writing—original draft, review and editing. K.S.: conceptualization, formal analysis, funding acquisition, methodology, visualization, writing—original draft, review and editing. K.Q.: conceptualization, funding acquisition, methodology, writing—original draft,

review and editing. Y.Y.: conceptualization, funding acquisition, investigation, methodology, project administration, supervision, writing—original draft, review and editing.

Competing interests. All authors declare there are no competing interests.

Funding. This research was supported by JSPS KAKENHI grant nos. JP17H00875 (Y.Y.), JP17H06342 (K.Q.), JP18K12015 (Y.Y.), JP19J11199 (F.Y.), JP19J21874 (N.Y.), JP19K14482 (K.S.), JP20H04581 (Y.Y.), JP20J21976 (A.I.), JP20K03479 (K.Q.). The funders had no role in study design, data collection and analysis, decision to publish or preparation of the manuscript.

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
