## [Reviewer comments · Royal Society Open Science]

Review History

RSOS-171019.R0 (Original submission)

Review form: Reviewer 1

Do you have any ethical concerns with this paper?

No

Recommendation?

Major revision

Comments to the Author(s)

In this manuscript and in the context of COVID-19 sanitary crisis, the authors want to compare the efficacy of prevention messages making self-identity salient (“don’t be a spreader”), compared to messages making self-identity less salient (“don’t spread”). When it comes to this crisis, I do think that it is a worthy goal for the field to conduct psychological research appropriately. I think that this research program is the kind of program that we need in these times, it is a straightforward intervention that could easily be adopted. However, even if the hypothesis the authors make sounds plausible, the current amount of evidence in their literature

review is too light to consider testing this intervention right now. Overall, I have concerns that it is too soon to accept this paper, but if enough evidence is found to test this hypothesis, I think the results could be very interesting. Besides the literature review, the procedure seems fair regarding the test of the hypothesis authors want to test. It also has to be noted that the manuscript is well written and is easy to understand. Below, I describe several suggestions that, I think, could improve this program.

In their introduction, the authors discuss work by Bryan, Adams, and Monin (2013) to evoke the possibility that framing a prevention message in a way that makes identity salient should be more effective compared to a message that does not. Unfortunately, Bryan et al.'s (2013) empirical paper is the only work cited as the rationale for this study. Authors should mention whether further research has followed Bryan et al.'s (2013) work and they should also discuss the kind of effect size one can expect from such intervention. Depending on the kind of effect sizes that we can expect from this literature, it is an occasion to build a case for their research program. In the current version of the manuscript, the readers would understand why it is important to investigate processes making people more likely to adopt behavior limiting the spread of the pandemic, but I am afraid they might still wonder why focusing on this intervention and not another one. I think that this manuscript would gain in quality if authors conduct a more in-depth review of the literature, as it would help the readers (including policymakers) to know what they can expect from it.

Besides the literature review, I think that authors could improve some part of their protocol. In their experiment, authors want to compare the two messages described above ("don't be a spreader" vs. "don't spread"), along with a control condition, in an experiment with a longitudinal design (wave 1: exposure to the message; wave 2: measures, one week later). Authors will recruit their participants using their email address, so they can send a reminder for the second wave. Given the sensitive nature of information related to the adoption of behaviors limiting the spread of COVID-19, it is critical that the authors do everything they can, so the identifying information is not linked to the data on COVID-19. Authors should be explicit about it.

Regarding the data collected during the experiment, we don't know whether the data set collected for this experiment will be publicly available. Authors should mention it in their manuscript.

Authors conduct an a priori power analysis to decide what kind of sample size they need. They initially decide to compute the sample size needed based on Bryan, Adams, & Monin (2013) and other considerations, but end up doubling this sample size (for valid reasons). I think the reader would appreciate a sensitivity analysis on top of the information given. Given the sample size authors decided to use, what is the effect size authors have 80% statistical power to detect?

Regarding the analysis of their data, I think some parts could be improved. Authors will use a longitudinal design where they expect participants to come back 1-week after a first wave. However, we don't know if the authors will adopt exclusion criteria depending on when participants decide to complete the second part of the study. It seems especially important because we know that things can move quickly in the current time, and the time when participants decide to take part in the second part of the study could carry a lot of variance). Authors should address this point in their preregistration.

Page 6, authors note that if they end up recruiting more than 1380 participants, they would use only the data from the first 1380 participants (based on timestamp), hence excluding some participants. From an ethical point of view, I would argue against the exclusion of the participants. Indeed, removing the participants from the analysis would 1) decrease the statistical power of the analysis, 2) waste participants' time and the resources engage in this study. I suggest adopting a more inclusive criterion.

Authors suggest using an ANCOVA (instead of an ANOVA) to test their hypothesis while controlling by the perceived vulnerability to disease. However, it is not entirely clear why it is important to do so, as authors are not controlling for a confound (the key independent variable is experimentally manipulated) and the experiment seems appropriately powered.

Moreover, regarding the statistical power of the preregistered analysis, authors are planning on testing the omnibus effect related to their experimental manipulation and follow up with a Tukey's multiple comparison test. Tukey's test is a correction of the alpha rate that is needed when one test multiple non-orthogonal hypotheses to control for the type I error. If authors want to test their hypotheses while controlling for an alpha rate of 5% and without being too conservative, they should consider using contrasts to test orthogonal hypotheses (see Rosnow & Rosenthal's work). A contrast that seems appropriate here would be the treatment contrast, as it could answer both a pragmatic question (i.e., is a prevention message making self-identity salient better than nothing?) and a theoretical one (i.e., is a prevention message making self-identity salient better than one that does not; but see also, Helmert contrasts).

Review form: Reviewer 2 (Jackie Thompson)

Do you have any ethical concerns with this paper?

No

Recommendation?

Major revision

Comments to the Author(s)

See attached document (Appendix A).

Review form: Reviewer 3

Do you have any ethical concerns with this paper?

No

Recommendation?

Major revision

Comments to the Author(s)

This is a useful extension of a well-known effect in social ethics communication that has also been applied successfully in environmental attitudes work. To my knowledge, it has not been applied explicitly to infectious disease communication, and thus it has a good amount of applied potential.

From a basic science perspective, the main theoretical merit is knowing whether these effects from ethics and environmental research generalize to infectious disease communication. On this note, the contribution doesn't offer much in the way of explanation of why it wouldn't generalize to infectious disease, or measures that will help to explain why it didn't generalize if it doesn't. Thus, a primary weakness of the present report is that it doesn't really offer a "back-up" plan if they don't find the results they expect. All we will know is that, in this case, the previous results don't generalize. I don't have a great solution to this, but to the extent the authors could think of ways to anticipate what it might mean theoretically about infectious disease communication if their predictions don't hold, it could be helpful and might steer them toward additional variables to measure. For example, there might be differences in perceived agency or inevitability when it

comes to infectious diseases that might reduce the effect of such a manipulation. Ethics and environmental stewardship are much more agent oriented domains, whereas infectious disease tends to lack the same level of controllability. Perceived control can have a large impact on behavioral intentions and attitude-behavior alignment.

No psychometric criteria are described for averaging over the questions in Table 1 or Table 2. I assume this is the intention as the statistical methods describes using difference scores for these tables as a whole as dependent variables. It would be useful to look at whether they do measure consistent psychological constructs by looking at item analyses and alphas. If not, whether the items will be looked at individually (instead of means) should be discussed and whether there are plans to correct for multiple comparisons if they move to an item-specific strategy. For example, what if people treat being in a low ventilation area as fundamentally different from going out to buy necessities? Or wearing ventilators as different from not speaking loudly? It seems these scales have the possibility of measuring multiple different constructs.

I don't understand why the authors need to set an upper bound on sample size. If they go over planned sample size, they could theoretically keep to the original planned t criteria (if they are worried about effect sizes that are too low being accepted as significant), for example, but the added precision would always seemingly be worthwhile.

Review form: Reviewer 4 (Onurcan Yilmaz)

Do you have any ethical concerns with this paper?

No

Recommendation?

Accept with minor revision

Comments to the Author(s)

The authors set out to determine whether different messages (the spreading, the spreader vs. control) would prevent high-risk behaviors during the pandemic. They use a mixed design where the dependent variable will be measured in a within-subjects manner, and the manipulation will be between-subjects (if I understood correctly).

I like the proposal for its simplicity and applicability, and I think it will be a timely contribution. But the authors should clarify some points before I recommend it for stage 1 acceptance.

If I understood correctly, the authors will first measure preventive behavioral intentions, and then in Wave 2, they will use the same participants and before measuring their preventive behavioral intentions again, the participants will be randomly assigned to three different conditions (the spreading, the spreader, or the control). They also measure PVS as a covariate and will look at the difference between a two-time frame due to the saliency of the pandemic. However, In P6, L47-59, there is no reference to experimental manipulations. The authors even explicitly state that the participants will be presented with "no reminder in all conditions" in Wave 2. So I think I'm misunderstanding the design. I think a separate design section should help the authors clarify their points.

There are also some inconsistencies. For example, in P6, L22, the authors are saying that they will take the participants' email only in the first Wave but in the following paragraph, they report that they will measure several other variables including COVID-19 IP. This inconsistency should be solved. Relatedly, the hypotheses should be clearer.

If I were you, in Wave 1, I would measure the dependent variable and the covariates together with the demographics without experimental manipulation (i.e., baseline). And in Wave 2, the same participants would be randomly assigned to three conditions and then I would measure the same DV and covariate again. And to analyze this, I would conduct 3 (manipulation: spreading, spreader, control) by 2 (time: pretest, posttest) mixed ANOVA (where the latter factor was within-subjects) and look at whether there is any significant interaction. If this is the design the authors proposed, it should be better clarified. If not, I would like to quickly review a revised version of the manuscript.

A minor point is when COVID-19 IP abbreviation first appeared in the hypotheses section, the unabridged version should be reported in parenthesis.

I always sign my reviews,
Onurcan Yilmaz

Review form: Reviewer 5

Do you have any ethical concerns with this paper?

No

Recommendation?

Major revision

Comments to the Author(s)

Please find the review of your submitted report in the attached PDF document.

Decision letter (RSOS-200793.R0)

Dear Dr Yonemitsu,

The Editors assigned to your stage one Registered Report ("Warning "Don't spread" vs. "Don't be a spreader" to prevent the COVID-19 pandemic") have now received comments from reviewers.

We would like you to revise your paper in accordance with the referee and editors suggestions which can be found below (not including confidential reports to the Editor). Please note this decision does not guarantee eventual acceptance.

When submitting your revised manuscript, you must respond to the comments made by the referees and upload a file "Response to Referees" in "Section 2 - File Upload". Please use this to document how you have responded to the comments, and the adjustments you have made. In

order to expedite the processing of the revised manuscript, please be as specific as possible in your response.

Once again, thank you for submitting your manuscript to Royal Society Open Science and we look forward to receiving your revision. If you have any questions at all, please do not hesitate to get in touch.

Kind regards,
Lianne Parkhouse
Editorial Coordinator
Royal Society Open Science
openscience@royalsociety.org

on behalf of Professor Chris Chambers (Registered Reports Editor, Royal Society Open Science)
openscience@royalsociety.org

Associate Editor Comments to Author (Professor Chris Chambers):

Five reviewers with a range of expertise (from field specialists to methodologists) have now assessed the manuscript, and let me begin by offering my deepest thanks to the reviewers for providing such high quality assessments on such an extraordinary timescale. All of the reviewers find merit in the proposal while also raising concerns that span the full range of Stage 1 review criteria. Some of the most significant weaknesses to address include the strength of the theoretical rationale, the precision of hypotheses (and mapping of hypotheses to specific analyses), the sampling plan (including the definition of the smallest effect size of interest), the validity of the measures, and appropriateness of the statistical analysis plan, and the overall level of methodological detail.

Substantial improvements in all these areas (and others, as identified in the reviews) will be needed to achieve Stage 1 in-principle acceptance, yet I am convinced by the overall enthusiasm and constructive nature of the reviews that the proposal is sufficiently promising to offer the authors this opportunity. A major revision is therefore invited.

Reviewer Comments to Author:
Reviewer: 1
Comments to the Author(s)

In this manuscript and in the context of COVID-19 sanitary crisis, the authors want to compare the efficacy of prevention messages making self-identity salient (“don’t be a spreader”), compared to messages making self-identity less salient (“don’t spread”). When it comes to this crisis, I do think that it is a worthy goal for the field to conduct psychological research appropriately. I think that this research program is the kind of program that we need in these times, it is a straightforward intervention that could easily be adopted. However, even if the hypothesis the authors make sounds plausible, the current amount of evidence in their literature review is too light to consider testing this intervention right now. Overall, I have concerns that it is too soon to accept this paper, but if enough evidence is found to test this hypothesis, I think the results could be very interesting. Besides the literature review, the procedure seems fair regarding the test of the hypothesis authors want to test. It also has to be noted that the manuscript is well written and is easy to understand. Below, I describe several suggestions that, I think, could improve this program.

In their introduction, the authors discuss work by Bryan, Adams, and Monin (2013) to evoke the possibility that framing a prevention message in a way that makes identity salient should be more effective compared to a message that does not. Unfortunately, Bryan et al.’s (2013) empirical paper is the only work cited as the rationale for this study. Authors should mention whether further research has followed Bryan et al.’s (2013) work and they should also discuss the kind of

effect size one can expect from such intervention. Depending on the kind of effect sizes that we can expect from this literature, it is an occasion to build a case for their research program. In the current version of the manuscript, the readers would understand why it is important to investigate processes making people more likely to adopt behavior limiting the spread of the pandemic, but I am afraid they might still wonder why focusing on this intervention and not another one. I think that this manuscript would gain in quality if authors conduct a more in-depth review of the literature, as it would help the readers (including policymakers) to know what they can expect from it.

Besides the literature review, I think that authors could improve some part of their protocol. In their experiment, authors want to compare the two messages described above (“don’t be a spreader” vs. “don’t spread”), along with a control condition, in an experiment with a longitudinal design (wave 1: exposure to the message; wave 2: measures, one week later). Authors will recruit their participants using their email address, so they can send a reminder for the second wave. Given the sensitive nature of information related to the adoption of behaviors limiting the spread of COVID-19, it is critical that the authors do everything they can, so the identifying information is not linked to the data on COVID-19. Authors should be explicit about it.

Regarding the data collected during the experiment, we don’t know whether the data set collected for this experiment will be publicly available. Authors should mention it in their manuscript.

Authors conduct an a priori power analysis to decide what kind of sample size they need. They initially decide to compute the sample size needed based on Bryan, Adams, & Monin (2013) and other considerations, but end up doubling this sample size (for valid reasons). I think the reader would appreciate a sensitivity analysis on top of the information given. Given the sample size authors decided to use, what is the effect size authors have 80% statistical power to detect?

Regarding the analysis of their data, I think some parts could be improved. Authors will use a longitudinal design where they expect participants to come back 1-week after a first wave. However, we don’t know if the authors will adopt exclusion criteria depending on when participants decide to complete the second part of the study. It seems especially important because we know that things can move quickly in the current time, and the time when participants decide to take part in the second part of the study could carry a lot of variance). Authors should address this point in their preregistration.

Page 6, authors note that if they end up recruiting more than 1380 participants, they would use only the data from the first 1380 participants (based on timestamp), hence excluding some participants. From an ethical point of view, I would argue against the exclusion of the participants. Indeed, removing the participants from the analysis would 1) decrease the statistical power of the analysis, 2) waste participants’ time and the resources engaged in this study. I suggest adopting a more inclusive criterion.

Authors suggest using an ANCOVA (instead of an ANOVA) to test their hypothesis while controlling by the perceived vulnerability to disease. However, it is not entirely clear why it is important to do so, as authors are not controlling for a confound (the key independent variable is experimentally manipulated) and the experiment seems appropriately powered. Moreover, regarding the statistical power of the preregistered analysis, authors are planning on testing the omnibus effect related to their experimental manipulation and follow up with a Tukey’s multiple comparison test. Tukey’s test is a correction of the alpha rate that is needed when one tests multiple non-orthogonal hypotheses to control for the type I error. If authors want to test their hypotheses while controlling for an alpha rate of 5% and without being too conservative, they should consider using contrasts to test orthogonal hypotheses (see Rosnow & Rosenthal’s work). A contrast that seems appropriate here would be the treatment contrast, as it could answer both a pragmatic question (i.e., is a prevention message making self-identity salient

better than nothing?) and a theoretical one (i.e., is a prevention message making self-identity salient better than one that does not; but see also, Helmert contrasts).

Reviewer: 2

Comments to the Author(s)

See attached document, Review for Registered Report RSOS - COVID messaging.docx

Reviewer: 3

Comments to the Author(s)

This is a useful extension of a well-known effect in social ethics communication that has also been applied successfully in environmental attitudes work. To my knowledge, it has not been applied explicitly to infectious disease communication, and thus it has a good amount of applied potential.

From a basic science perspective, the main theoretical merit is knowing whether these effects from ethics and environmental research generalize to infectious disease communication. On this note, the contribution doesn't offer much in the way of explanation of why it wouldn't generalize to infectious disease, or measures that will help to explain why it didn't generalize if it doesn't. Thus, a primary weakness of the present report is that it doesn't really offer a "back-up" plan if they don't find the results they expect. All we will know is that, in this case, the previous results don't generalize. I don't have a great solution to this, but to the extent the authors could think of ways to anticipate what it might mean theoretically about infectious disease communication if their predictions don't hold, it could be helpful and might steer them toward additional variables to measure. For example, there might be differences in perceived agency or inevitability when it comes to infectious diseases that might reduce the effect of such a manipulation. Ethics and environmental stewardship are much more agent oriented domains, whereas infectious disease tends to lack the same level of controllability. Perceived control can have a large impact on behavioral intentions and attitude-behavior alignment.

No psychometric criteria are described for averaging over the questions in Table 1 or Table 2. I assume this is the intention as the statistical methods describes using difference scores for these tables as a whole as dependent variables. It would be useful to look at whether they do measure consistent psychological constructs by looking at item analyses and alphas. If not, whether the items will be looked at individually (instead of means) should be discussed and whether there are plans to correct for multiple comparisons if they move to an item-specific strategy. For example, what if people treat being in a low ventilation area as fundamentally different from going out to buy necessities? Or wearing ventilators as different from not speaking loudly? It seems these scales have the possibility of measuring multiple different constructs.

I don't understand why the authors need to set an upper bound on sample size. If they go over planned sample size, they could theoretically keep to the original planned t criteria (if they are worried about effect sizes that are too low being accepted as significant), for example, but the added precision would always seemingly be worthwhile.

Reviewer: 4

Comments to the Author(s)

The authors set out to determine whether different messages (the spreading, the spreader vs. control) would prevent high-risk behaviors during the pandemic. They use a mixed design where the dependent variable will be measured in a within-subjects manner, and the manipulation will be between-subjects (if I understood correctly).

I like the proposal for its simplicity and applicability, and I think it will be a timely contribution. But the authors should clarify some points before I recommend it for stage 1 acceptance.

If I understood correctly, the authors will first measure preventive behavioral intentions, and then in Wave 2, they will use the same participants and before measuring their preventive behavioral intentions again, the participants will be randomly assigned to three different conditions (the spreading, the spreader, or the control). They also measure PVS as a covariate and will look at the difference between a two-time frame due to the saliency of the pandemic. However, In P6, L47-59, there is no reference to experimental manipulations. The authors even explicitly state that the participants will be presented with “no reminder in all conditions” in Wave 2. So I think I’m misunderstanding the design. I think a separate design section should help the authors clarify their points.

There are also some inconsistencies. For example, in P6, L22, the authors are saying that they will take the participants’ email only in the first Wave but in the following paragraph, they report that they will measure several other variables including COVID-19 IP. This inconsistency should be solved. Relatedly, the hypotheses should be clearer.

If I were you, in Wave 1, I would measure the dependent variable and the covariates together with the demographics without experimental manipulation (i.e., baseline). And in Wave 2, the same participants would be randomly assigned to three conditions and then I would measure the same DV and covariate again. And to analyze this, I would conduct 3 (manipulation: spreading, spreader, control) by 2 (time: pretest, posttest) mixed ANOVA (where the latter factor was within-subjects) and look at whether there is any significant interaction. If this is the design the authors proposed, it should be better clarified. If not, I would like to quickly review a revised version of the manuscript.

A minor point is when COVID-19 IP abbreviation first appeared in the hypotheses section, the unabridged version should be reported in parenthesis.

I always sign my reviews,
Onurcan Yilmaz

Reviewer: 5
Comments to the Author(s)

Please find the review of your submitted report in the attached PDF document.

Author's Response to Decision Letter for (RSOS-200793.R0)

See Appendix C.

RSOS-200793.R1 (Revision)

Review form: Reviewer 1

Do you have any ethical concerns with this paper?

No

Recommendation?

Accept with minor revision

Comments to the Author(s)

See attached file (Appendix D).

Review form: Reviewer 2 (Jackie Thompson)**Do you have any ethical concerns with this paper?**

No

Recommendation?

Accept with minor revision

Comments to the Author(s)

See the fully formatted version of my comments in the word doc (Appendix E). - COVID messaging.docx"

Plain text is pasted below, but is missing some formatting.

Firstly, I am pleased and impressed to see how much work the authors have put into improving the Stage 1 manuscript (study protocol/preregistration) in such a short amount of time.

...

Below I note the changes I think they still need to make before the manuscript should be accepted:

Hypotheses:

- The 2 hypotheses are now much clearer that they are measuring the change from wave 1 to wave 2.
- However, I still believe the hypotheses are not quite what the authors intend, based on my reading of the rest of the manuscript. It seems to me the authors do not care whether the spreading condition will differ from control (although they expect based on past results that it will not differ- either way, it would not affect their conclusions about the spreader condition, i.e., the condition they are interested in). Despite this, the authors have added into H1 and H2 the prediction that change scores will not differ between the spreading and control conditions. I don't think that is a main element of what they are interested in, and makes the hypotheses too complex. I think instead they should remove the last sentence of hypotheses 1 and 2, and just change the null hypothesis to state that the spreader condition will not differ from either the spreading or control condition. It could be interesting to test whether the two conditions of non-interest (control and spreading) differ, but I would either put that as a separate, secondary hypothesis, or just include as an exploratory analysis.
- The "side hypothesis" about PVD should be listed in the hypotheses section and clearly indicated it is a secondary / less important / not main hypothesis (instead of only mentioning it in the Measures section). I apologize that my previous note on this from Round 1 (point 2-29 in the authors' response) was unclear - I meant the authors should include just the hypothesis itself in the Hypotheses section, and then detail all the information about the measures themselves in a separate section within the methods, which they have now done.

Measures

- I am pleased to see there is now more clarity about what each scale is attempting to measure, and the reasons for using these unvalidated scales (due to their tie to the government guidelines). However, a few remaining concerns below:
- I would highly recommend rephrasing the behavior scale items to ask about behaviors themselves rather than avoidance of behaviors. If you ask about "how often did you avoid x

behavior” it is difficult for the participant to answer – it is much harder to think of what you have NOT done, versus what you did. So I would instead ask, e.g., “how often did you go out” and then reverse-code these items.

- Are you sure you want to have a Likert response scale of “never” to “often” for the reported behavior measure? I actually thought that asking participants to count the number of times they did an action was more objective. It is hard to know what going out “often” means. However I can see arguments for and against either option. I can foresee that reporting raw numbers could increase the variability in responses and give problems with non-normality (right skew), so that is one argument against it (although these issues can be attenuated by testing for skew and applying a transformation if there is skew). Another possible argument against participants entering raw numbers is if you want to be able to directly compare the behavior intention and reported behavior scales – but even then, the response scales are qualitatively different (agreement versus reporting frequency) even though they have the same number of scale points. I would think carefully about why you want to use a Likert scale versus reporting actual numbers of behaviors. If you care more about whether the manipulation will change total numbers of behaviors across a population, I would go for reporting raw numbers. If you care more about how many individuals the manipulation will convince to change their behavior, go for the Likert scale.
- Some items on the reported behavior scale will need additional instructions, or include a “not-applicable” option, particularly number 2 and number 7. For instance, “how often did you go out wearing a mask” will depend on how often the participant went out, so it may be misleading (for instance a participant who did not go out at all would answer “never,” and therefore reduce their score – even though they followed guidelines by not going out!) Similarly, if participants did not cough or sneeze, they will get a low score on item 7, even though they are following guidelines. I would suggest giving participants a “not applicable” option for those 2 items, but if you cannot do that, you need to acknowledge that this might introduce noise to your measure and think of ways to address that (for instance, you might plan to re-run the analysis both with and without these 2 items).
- Related to the above note – item 2 is ambiguous and should be reworded to something like, “when you went out, how often did you wear a mask?” You care about the percentage of time people wore a mask in public, not the total amount of time (otherwise you would not know about how much of the time they went out in public NOT wearing a mask).
- Despite the authors’ arguments why they don’t expect a ceiling effect, I think they still have to be prepared for the very real possibility of a ceiling or floor effect on both scales. Please consider how you will test for a ceiling/floor effect, and what you will do if you find one.
- CRUCIAL: I unfortunately do not have enough knowledge about principle component scores to know whether they are appropriate to use as a dependent variable, but I have doubts. These should be checked by an expert. I gather using the principal component scores is an attempt to weight the items according to how well they cohere with the overall (forced single-factor) scale? I am not sure about that. – I would report an analysis that uses mean scores on each scale, regardless of whether you also use a principal components score. It is easier to understand and does not make “invisible” assumptions about how to weight the various items on the scale. Also, I am not sure whether it is even appropriate to calculate a linear change score between 2 different principal component scores. That doesn’t make sense to me how this will capture change in participants’ behavior – instead won’t it capture change in the relative weighting of different items on the scale, because the factor loadings will be idiosyncratic to each PCA? If everyone’s 2nd-Wave scores remained exactly the same as 1st-Wave except that they all went up by 1 Likert scale point, then wouldn’t the principal component scores remain the same both times, even though behavior changed considerably? (But I may be misunderstanding principal component scores, I am not an expert!)
- Related to the above I think it’s good to do a PCA to see how well the items on your scales cohere into a single measure, but since you have an a priori reason to use those items (they are the behavior guidelines the government wants to influence) I’m not sure it’s appropriate to drop any of them simply because they don’t fit in a single-factor model. You stated before that you don’t care whether these items all are affected using the same mechanism (i.e., you don’t have hypotheses about approach versus avoidance behaviors) and I think the same principle

applies here—you include the items because of the practical consideration of wanting to measure those specific behaviors, so you should include them all somehow. That could be forcing them to be all in the same scale regardless of fit, or separating them into separate factors based on the PCA.

Attention Check questions

- I like the new attention check questions, and it is good that they are text-based. However, since the participants are now answering everything on a Likert scale, they will still stand out if participants have to enter text! I would give them a Likert scale answer (unless you change the reported behavior scale back to number-entry).
- I am not convinced the instructional manipulation check will be effective. It is merely another attention check, and does not in any way guarantee the participant read the message. You must include a test that can tell whether or not the participant understood the message – I would highly recommend asking them to type it back on the next page (or you could ask them to choose it from a list of options – in this case I would not include both spreader and spreading message in the options, I would include other options that don't mention "spread" in any way, such as "be safe" or "don't be reckless"). Are you concerned with external validity, because in real life people are not asked to type back messages, or with demand characteristics (that participants might figure out the study is about the message)? I would not be concerned with these. I would be more concerned that the message gets lost and participants do not see it.

Small points:

- The section on ACQs still says "We will insert a simple math problem as an attention check question" (page 37 line 32)
- In the reply to point 2-21, the authors state that they will ask participants to answer both scales based on the week before participation in each wave. Shouldn't the behavior intention scale be based on the upcoming week (the week after each wave) instead? Behavioral intention is usually measured regarding the future. Have I misunderstood? Figure 1 is generally helpful but it does not differentiate between the two scale measures.

Review form: Reviewer 4

Do you have any ethical concerns with this paper?

No

Recommendation?

Accept with minor revision

Comments to the Author(s)

In general, the authors have followed most of the recommendations and this revision addresses all of my concerns.

I've two minor suggestions for improvement.

- 1) In the abstract, the authors use the word "actual behavior" but they will measure behavioral intentions in both IP-intention and IP-behavior scales. Both measures are based on participants' self-reported intentions. To measure actual preventive behaviors, the authors should not use a self-report measure; rather, they can use location data such as <https://www.unacast.com/>.
- 2) If a null-finding is hypothesized, a Bayes Factor should be calculated (i.e., "The change from the 1st wave (baseline) to the 2nd wave will not significantly differ between the spreading and control conditions.)."

Onurcan Yilmaz

Review form: Reviewer 5

Do you have any ethical concerns with this paper?

No

Recommendation?

Major revision

Comments to the Author(s)

Please see the attached PDF for my review (Appendix F).

Decision letter (RSOS-200793.R1)

Dear Dr Yonemitsu,

On behalf of the Editors, I am pleased to inform you that your Manuscript RSOS-200793.R1 entitled "Warning "Don't spread" vs. "Don't be a spreader" to prevent the COVID-19 pandemic" has been accepted in principle for publication in Royal Society Open Science subject to minor revision in accordance with the referee and editor suggestions. Please find their comments at the end of this email.

The reviewers and handling editors have recommended publication, but also suggest some minor revisions to your manuscript. Therefore, I invite you to respond to the comments and revise your manuscript.

Please you submit the revised version of your manuscript within 7 days (i.e. by the 10-Jun-2020). If you do not think you will be able to meet this date please let me know immediately.

When submitting your revised manuscript, you will be able to respond to the comments made by the referees and you should upload a file "Response to Referees". You can use this to document any changes you make to the original manuscript. In order to expedite the processing of the revised manuscript, please be as specific as possible in your response to the referees.

Full author guidelines can be found here <https://royalsocietypublishing.org/rsos/registered-reports>.

Kind regards,
Lianne Parkhouse
Editorial Coordinator
Royal Society Open Science
openscience@royalsociety.org

on behalf of Professor Chris Chambers (Associate Editor) and Chris Chambers (Subject Editor,
Royal Society Open Science)
openscience@royalsociety.org

Associate Editor Comments to Author (Professor Chris Chambers):

Four of the five original reviewers have now assessed the revised manuscript, and broadly agree that the proposal is significantly closer to meeting the Stage 1 criteria. However, there remain some significant issues to address before IPA can be awarded, primarily in terms of the conceptual framing of the study, appropriate use of statistical techniques (including methods for drawing conclusions from null results), and strengthening of the manipulation/quality checks. Please attend carefully to these points in a revision.

Reviewer comments to Author:

Reviewer: 1
Comments to the Author(s)

See attached file.

Reviewer: 2
Comments to the Author(s)

See the fully formatted version of my comments in the word doc, "Review RSOS Stage 1 Round 2 - COVID messaging.docx"

Plain text is pasted below, but is missing some formatting.

Firstly, I am pleased and impressed to see how much work the authors have put into improving the Stage 1 manuscript (study protocol/preregistration) in such a short amount of time.

...

Below I note the changes I think they still need to make before the manuscript should be accepted:

Hypotheses:

- The 2 hypotheses are now much clearer that they are measuring the change from wave 1 to wave 2.
- However, I still believe the hypotheses are not quite what the authors intend, based on my reading of the rest of the manuscript. It seems to me the authors do not care whether the spreading condition will differ from control (although they expect based on past results that it will not differ- either way, it would not affect their conclusions about the spreader condition, i.e., the condition they are interested in). Despite this, the authors have added into H1 and H2 the prediction that change scores will not differ between the spreading and control conditions. I don't

think that is a main element of what they are interested in, and makes the hypotheses too complex. I think instead they should remove the last sentence of hypotheses 1 and 2, and just change the null hypothesis to state that the spreader condition will not differ from either the spreading or control condition. It could be interesting to test whether the two conditions of non-interest (control and spreading) differ, but I would either put that as a separate, secondary hypothesis, or just include as an exploratory analysis.

- The “side hypothesis” about PVD should be listed in the hypotheses section and clearly indicated it is a secondary / less important / not main hypothesis (instead of only mentioning it in the Measures section). I apologize that my previous note on this from Round 1 (point 2-29 in the authors’ response) was unclear – I meant the authors should include just the hypothesis itself in the Hypotheses section, and then detail all the information about the measures themselves in a separate section within the methods, which they have now done.

Measures

- I am pleased to see there is now more clarity about what each scale is attempting to measure, and the reasons for using these unvalidated scales (due to their tie to the government guidelines). However, a few remaining concerns below:

- I would highly recommend rephrasing the behavior scale items to ask about behaviors themselves rather than avoidance of behaviors. If you ask about “how often did you avoid x behavior” it is difficult for the participant to answer – it is much harder to think of what you have NOT done, versus what you did. So I would instead ask, e.g., “how often did you go out” and then reverse-code these items.

- Are you sure you want to have a Likert response scale of “never” to “often” for the reported behavior measure? I actually thought that asking participants to count the number of times they did an action was more objective. It is hard to know what going out “often” means. However I can see arguments for and against either option. I can foresee that reporting raw numbers could increase the variability in responses and give problems with non-normality (right skew), so that is one argument against it (although these issues can be attenuated by testing for skew and applying a transformation if there is skew). Another possible argument against participants entering raw numbers is if you want to be able to directly compare the behavior intention and reported behavior scales – but even then, the response scales are qualitatively different (agreement versus reporting frequency) even though they have the same number of scale points. I would think carefully about why you want to use a Likert scale versus reporting actual numbers of behaviors. If you care more about whether the manipulation will change total numbers of behaviors across a population, I would go for reporting raw numbers. If you care more about how many individuals the manipulation will convince to change their behavior, go for the Likert scale.

- Some items on the reported behavior scale will need additional instructions, or include a “not-applicable” option, particularly number 2 and number 7. For instance, “how often did you go out wearing a mask” will depend on how often the participant went out, so it may be misleading (for instance a participant who did not go out at all would answer “never,” and therefore reduce their score – even though they followed guidelines by not going out!) Similarly, if participants did not cough or sneeze, they will get a low score on item 7, even though they are following guidelines. I would suggest giving participants a “not applicable” option for those 2 items, but if you cannot do that, you need to acknowledge that this might introduce noise to your measure and think of ways to address that (for instance, you might plan to re-run the analysis both with and without these 2 items).

- Related to the above note – item 2 is ambiguous and should be reworded to something like, “when you went out, how often did you wear a mask?” You care about the percentage of time people wore a mask in public, not the total amount of time (otherwise you would not know about how much of the time they went out in public NOT wearing a mask).

- Despite the authors’ arguments why they don’t expect a ceiling effect, I think they still have to be prepared for the very real possibility of a ceiling or floor effect on both scales. Please consider how you will test for a ceiling/floor effect, and what you will do if you find one.

- CRUCIAL: I unfortunately do not have enough knowledge about principle component scores to know whether they are appropriate to use as a dependent variable, but I have doubts. These

should be checked by an expert. I gather using the principal component scores is an attempt to weight the items according to how well they cohere with the overall (forced single-factor) scale? I am not sure about that. – I would report an analysis that uses mean scores on each scale, regardless of whether you also use a principal components score. It is easier to understand and does not make “invisible” assumptions about how to weight the various items on the scale. Also, I am not sure whether it is even appropriate to calculate a linear change score between 2 different principal component scores. That doesn’t make sense to me how this will capture change in participants’ behavior – instead won’t it capture change in the relative weighting of different items on the scale, because the factor loadings will be idiosyncratic to each PCA? If everyone’s 2nd-Wave scores remained exactly the same as 1st-Wave except that they all went up by 1 Likert scale point, then wouldn’t the principal component scores remain the same both times, even though behavior changed considerably? (But I may be misunderstanding principal component scores, I am not an expert!)

- Related to the above I think it’s good to do a PCA to see how well the items on your scales cohere into a single measure, but since you have an a priori reason to use those items (they are the behavior guidelines the government wants to influence) I’m not sure it’s appropriate to drop any of them simply because they don’t fit in a single-factor model. You stated before that you don’t care whether these items all are affected using the same mechanism (i.e., you don’t have hypotheses about approach versus avoidance behaviors) and I think the same principle applies here – you include the items because of the practical consideration of wanting to measure those specific behaviors, so you should include them all somehow. That could be forcing them to be all in the same scale regardless of fit, or separating them into separate factors based on the PCA.

Attention Check questions

- I like the new attention check questions, and it is good that they are text-based. However, since the participants are now answering everything on a Likert scale, they will still stand out if participants have to enter text! I would give them a Likert scale answer (unless you change the reported behavior scale back to number-entry).

- I am not convinced the instructional manipulation check will be effective. It is merely another attention check, and does not in any way guarantee the participant read the message. You must include a test that can tell whether or not the participant understood the message – I would highly recommend asking them to type it back on the next page (or you could ask them to choose it from a list of options – in this case I would not include both spreader and spreading message in the options, I would include other options that don’t mention “spread” in any way, such as “be safe” or “don’t be reckless”). Are you concerned with external validity, because in real life people are not asked to type back messages, or with demand characteristics (that participants might figure out the study is about the message)? I would not be concerned with these. I would be more concerned that the message gets lost and participants do not see it.

Small points:

- The section on ACQs still says “We will insert a simple math problem as an attention check question” (page 37 line 32)
- In the reply to point 2-21, the authors state that they will ask participants to answer both scales based on the week before participation in each wave. Shouldn’t the behavior intention scale be based on the upcoming week (the week after each wave) instead? Behavioral intention is usually measured regarding the future. Have I misunderstood? Figure 1 is generally helpful but it does not differentiate between the two scale measures.

Reviewer: 4

Comments to the Author(s)

In general, the authors have followed most of the recommendations and this revision addresses all of my concerns.

I’ve two minor suggestions for improvement.

- 1) In the abstract, the authors use the word "actual behavior" but they will measure behavioral intentions in both IP-intention and IP-behavior scales. Both measures are based on participants' self-reported intentions. To measure actual preventive behaviors, the authors should not use a self-report measure; rather, they can use location data such as <https://www.unacast.com/>.
- 2) If a null-finding is hypothesized, a Bayes Factor should be calculated (i.e., "The change from the 1st wave (baseline) to the 2nd wave will not significantly differ between the spreading and control conditions)."

Onurcan Yilmaz

Reviewer: 5
Comments to the Author(s)

Please see the attached PDF for my review.

Author's Response to Decision Letter for (RSOS-200793.R1)

See Appendix G.

Decision letter (RSOS-200793.R2)

Dear Dr Yonemitsu

On behalf of the Editor, I am pleased to inform you that your Manuscript RSOS-200793.R2 entitled "Warning 'Don't spread' vs. 'Don't be a spreader' to prevent the COVID-19 pandemic" has been accepted in principle for publication in Royal Society Open Science. You may now progress to Stage 2 and complete the study as approved.

Please read the following email carefully

Your accepted Stage 1 manuscript has been publicly registered at:
<https://doi.org/10.17605/OSF.IO/KZ5Y4>

Following completion of your study, we invite you to resubmit your paper for peer review as a Stage 2 Registered Report. Please note that your manuscript can still be rejected for publication at Stage 2 if the editors consider any of the following conditions to be met:

- The results were unable to test the authors' proposed hypotheses by failing to meet the approved outcome-neutral criteria.
- The authors altered the Introduction, rationale, or hypotheses, as approved in the Stage 1 submission.

- The authors failed to adhere closely to the registered experimental procedures. Please note that any deviations from the approved experimental procedures must be communicated to the editor immediately for approval, and prior to the completion of data collection. Failure to do so can result in revocation of in-principle acceptance and rejection at Stage 2 (see complete guidelines for further information).
- Any post-hoc (unregistered) analyses were either unjustified, insufficiently caveated, or overly dominant in shaping the authors' conclusions.
- The authors' conclusions were not justified given the data obtained.

We encourage you to read the complete guidelines for authors concerning Stage 2 submissions at <https://royalsocietypublishing.org/rsos/registered-reports#ReviewerGuideRegRep>. Please especially note the requirements for data sharing, reporting the URL of the independently registered protocol, and that withdrawing your manuscript will result in publication of a Withdrawn Registration.

Once again, thank you for submitting your manuscript to Royal Society Open Science and we look forward to receiving your Stage 2 submission. If you have any questions at all, please do not hesitate to get in touch. We look forward to hearing from you shortly with the anticipated submission date for your stage two manuscript.

on behalf of Professor Chris Chambers (Registered Reports Editor, Royal Society Open Science)
 openscience@royalsociety.org

Author's Response to Decision Letter for (RSOS-200793.R2)

See Appendix H.

RSOS-200793.R3 (Revision)

Review form: Reviewer 2 (Jackie Thompson)

Is the manuscript scientifically sound in its present form?

No

Are the interpretations and conclusions justified by the results?

No

Is the language acceptable?

No

Do you have any ethical concerns with this paper?

No

Have you any concerns about statistical analyses in this paper?

Yes

Recommendation?

Major revision

Comments to the Author(s)

See attached file (Appendix I). I have also pasted the unformatted version below (I recommend reading the file).

Review RSOS Stage 2

Overview:

As in my round 1 reviews, my main concern about the study is the lack of interpretability of the null results, given that I do not believe the manipulation check is sufficient to show that participants read and internalized the public health message (i.e. the manipulation – “don’t be a spreader” versus “don’t spread” or no message). As I pointed out in Stage 1, the manipulation check the authors used was more of an attention check and did not assess whether participants had actually read and understood the manipulation message. Additionally, as I flagged in Stage 1, the fact that the measurement scales were not piloted or validated means that the resulting ceiling effects (or other possible noise in the data) may have affected the interpretability of the results. However, I acknowledge that the editor accepted the manuscript in principle, including the manipulation check and measurement scales, so this cannot be held against the authors at this stage since they carried out their study as approved.

As detailed below, I believe the other most important issues to address are the way the main results are interpreted (need to make it clear that the results are not conclusive either way) and to make the discussion section more clear, understandable, and relevant.

RR Specific review questions:

Whether the data are able to test the authors’ proposed hypotheses by passing the approved outcome-neutral criteria (such as absence of floor and ceiling effects or success of positive controls)

As detailed in Stage 1 review, I do not believe the manipulation check was sufficient, and the authors also report a possible ceiling effect in one of their scales. These need to be acknowledged in the discussion at least, as unfortunately they weaken the evidential value of the study in my view.

Whether the Introduction, rationale and stated hypotheses are the same as the approved Stage 1 submission

I did not have enough time to read through the whole Stage 1 manuscript (and I also had to hunt to find the Stage 1 ms itself!), but a quick glance suggested it was similar at least. The hypotheses were the same.

Whether the authors adhered precisely to the registered experimental procedures

Everything I checked had adhered to the registered procedures, but in some cases the registration was under-specified (for instance, the equivalence test did not specify a SESOI, smallest effect size of interest to test against).

Where applicable, whether any unregistered exploratory statistical analyses are justified, methodologically sound, and informative

The only exploratory analyses were the extra equivalence tests for the comparisons between additional pairs of conditions – I would say these were justified and informative but as noted below they need to report more information on the tests and explain the interpretation of the tests.

Whether the authors' conclusions are justified given the data

The authors need to reframe their conclusions, given the equivalence tests they ran – the data are not conclusive either way, and the authors' conclusions should reflect that.

Data:

I was able to find and download the data and codebooks from the OSF, and at face value they look ok. However I did not actually try re-running the analyses on the data, due to the short turnaround of this review.

Registration:

There is no registration for this manuscript on the OSF link provided. There is a registration-style document on the OSF project from 4 May 2020 (around the time of the initial Stage 1 submission, I believe), and appears to be based on the original first draft of the manuscript before Stage 1 revisions. It is in a question and answer format (presumably one of the OSF preregistration formats) rather than in the format of a Stage 1 manuscript, despite the document name ("RSOS_COVID_RRStage 1_MS.pdf"). This lack of a formal registration (or at least an upload of the full accepted Stage 1 manuscript) makes it impossible for me (or other readers) to check consistency of the Stage 2 manuscript with the in-principle-accepted Stage 1 manuscript, which is one of my main tasks as a reviewer of a Registered Report. For the purposes of this review I am assuming that the manuscript submitted 6 June 2020 is the in-principle-accepted Stage 1 manuscript, and am comparing it to that. However, I recommend the authors to upload the in-principle-accepted Stage 1 manuscript to the OSF project as well.

Results:

Results: PVD – please report the means, standard deviations, and sample sizes for each sample you compared for each measurement. This could be in a table or the text.

Figure 3 – it would be helpful to label the "before" and "after" labels by the dates collected, e.g. ("Before COVID-19 onset: 1 January 2018") and ("After COVID-19 onset: 5 June 2020") or whatever the actual dates were.

On page 15 and page 16 when you discuss the PVD results, it would help to rephrase the results to keep the order of the timepoints constant, e.g., say score A was higher pre-pandemic compared to mid-pandemic, and score B was lower pre-pandemic compared to mid-pandemic. A fast or not-careful reader might misinterpret the results to indicate that both scales were higher in the pre-pandemic timepoint (that was my first impression on a quick reading) because they are just scanning for the words "higher" or "lower". I was especially confused by this in the wording on page 16, 2nd paragraph of discussion.

Page 15, lines 6-8 – please phrase the results of the equivalence tests as suggested in the TOSTER paper, and report the smallest effect size of interest that you tested for (i.e., the lower and upper bounds of equivalence that the TOST tested against) for each test. The TOST statistics are not meaningful without reporting that information. I would also request that you interpret/explain what these results mean for your claims, for readers who are not familiar with equivalence tests – in this case, if both the experimental test and the equivalence test have $p > .05$, then it suggests your data cannot conclusively say either that there is or is no meaningful effect, either way.

Discussion:

Page 16, Discussion (first paragraph) –

– Instead of "reduce" intentions/behaviors, perhaps "affect" would be better as it is more general (you expected the intervention might reduce some intentions/behaviors and increase others)

- The second sentence is confusing and not quite right. Firstly, you tested changes in behavior or intention scores, not the scores themselves. Secondly, the wording makes it sound like you were testing between behavior and intention scores, rather than each separately.
- In sentence 3, after you mention there is no statistical support for Hyp 1 and 2; to fully report your results, you should mention that there is also no statistical support AGAINST Hyp 1 and 2 (as per the non-significant equivalence tests)

Page 16, paragraph 3 – instead of launching straight into possible theoretical explanations for lack of conclusive results, I would first outline or emphasize the limitations of evidence in this study – that you did not find conclusive evidence in either direction, and so the effect size may be smaller than you guessed, and/or the data may be noisier than you had hoped. I think it is important to acknowledge this first, before speculating about theoretical or methodological concerns.

I really struggled to understand the 4th paragraph of the Discussion, page 16-17.

- The paragraph seems to argue two opposite conclusions – that according to PVD results people might have reduced infectious behavior (fear of infection) but also increased it (lower perceived susceptibility). Are you suggesting both could be true at the same time? I don't see how that could be the case. And why would that explain why you got the results you did? The first argument (people already reduced high-risk infectious behaviors, as suggested by germ aversion on the PVD) is consistent with the point that you might have ceiling effects. But shouldn't the second argument (that people might underestimate potential to be a spreader) be working against any ceiling effects of anti-spreading behavior/intentions?
- You wrote "The present study's data of IP-intention score may tend to be close to a ceiling effect, and therefore germ aversion may influence IP intention rather than behaviour." Why should a ceiling effect on intention scores indicate that germ aversion had an effect on it? And why should it indicate that germ aversion did NOT affect behavior? This seems like pure speculation. You do not have data for IP intentions or behavior from the previous PVD data collection, so you don't know how changes in intentions or behavior might be related to germ aversion PVD levels.
- I found the second half of the paragraph hard to follow as well – I don't understand how the logic leads you to your conclusions.
- Perhaps it would be better to just outline a simpler explanation/speculation in this paragraph --that the ceiling effects left little room for differences in intention, and that the levels of attitudes around disease (as measured by PVD) might have an unknown effect on the results (and any further interpretation would be speculation).

Page 17, paragraph 2 – I'm not sure I understand the first sentence, "Moreover, the results of IP may reflect the degree of effect of highlighting self identity." Do you mean that you may have gotten inconclusive results because the true effect size is very small? If so, I would just state that!

Page 18, paragraph 2 (limitations)

- please rethink/rephrase your conclusions about the study's findings. You wrote "However, this indicates that at least the procedures in our study could not change IP-intentions and behaviour." However, the results of the equivalence tests combined with the hypothesis tests indicate that you cannot rule out either outcome – the data are inconclusive. You could instead state that your data could not conclusively show that the procedures changed intentions or behavior. (Or perhaps this sentence could simply be omitted and the paragraph focus solely on the limitations.)
- instead of just considering internal validity (what methodological changes might result in a conclusive effect), what about also considering external validity – how well do the study stimuli reflect the way citizens might encounter such messaging "in real life" if a government or other entity decided to adopt the self identity messaging? The current study design could potentially be considered a merit rather than a limitation in some ways, as perhaps it reflects the passive messaging and sometimes long time lags between message and behavior that you might find in a modest messaging campaign. Likewise, regarding the potential effects you discuss of increasing the frequency of exposure to messaging – instead of just speculating that it might

increase likelihood of observing effects, it would be more interesting to me as a reader to consider how it relates to real world situations (e.g., the lab experiment you describe might provide information on how a well-funded, centralized and co-ordinated messaging campaign might affect responses.)

- in the limitations section, I would also suggest highlighting other ways the study methodology might be improved. By "improved," I do not mean better chances of finding a significant effect, but rather a better reflection of whether the effect exists in the real world. What changes might allow you to make more definitive claims about the hypotheses you are testing? For instance, validating and refining the measures might have helped make the data less noisy, or having a better manipulation check might have allowed you to test whether people actually absorbed the message.

- Lastly, I would suggest that the authors also discuss the limitation that the 'manipulation check' cannot actually ascertain whether participants read the message, and the possibility that many participants simply did not read it.

Page 18, last paragraph – again, I do not follow the logic of this paragraph or understand the argument it is attempting to make. Could this be phrased more clearly and directly? For instance, I wasn't sure if the low case numbers were meant as an argument that people were already following safe behaviors, or that low numbers meant people perceived low risk and therefore the messaging simply didn't work at all. I also am unclear what is meant by "If the threat of COVID-19 spread increases, people can easily follow the reminders." However these were not the only issues with the paragraph -- I'm sorry, I simply didn't understand any of the points or how they follow on from each other.

Page 19, paragraph 1 – again, I am having trouble following the argument of this paragraph. Is the argument that self identity effects have been found in many cultures? Or that they have failed to be found in many cultures? I don't follow the logic of the last several sentences either. Could this be rewritten more clearly and simply, perhaps just stating what evidence there already is about whether self- identity effects are cross cultural.

Page 19, paragraph 2 – again, the data do not quite support this conclusion, that the manipulation "has little effect on changing IP intention and behaviour." Instead, the data simply do not provide support for (or against) the hypotheses

Minor points:

I recommend that, in the codebooks (description of data text files), the authors should add in information on the allowable range of responses (1-7 I believe?) and indicate what those numerical responses corresponded to (e.g., 1=strongly disagree, 7 = strongly agree). This would be very helpful but is not strictly required.

Page 6, top paragraph – the wording is a bit unclear. I suggest something like "As a secondary hypothesis for this research, we additionally predicted that the scores ... [etc.]"

Page 12, line 29 (Main Analysis) – could the first sentence clarify that the 2 mean scores for each measure are the before and after (or time 1 and time 2) measures? I did not understand this at first and had to read it a few times before realizing what it must mean.

Page 12, line 39 – the last word of the paragraph should be variables, plural. You have written variable (singular), which makes it sound like they have been combined – but I gather from the paragraph below that you actually did not combine, but ran separate tests for each.

Page 14, line 17-18 – I don't understand what this sentence means: "However, the total number of the participants excluded by each criterion did not match the total number of excluded participants." Could you briefly explain more what you meant, and why this is important?

Page 14, line 27 - I think the PVD results should be listed last in this section (after the intentions and behaviors results) as it is a secondary hypothesis. Put the most important, central results first.

When referring to the PVD in text (e.g., page 17 line 47, but elsewhere as well), instead of "current situation" it may be useful to say "during pandemic" or "several months into the COVID-19 pandemic", or even "June 2020" as it is unclear when exactly "current situation" refers to (and will be even harder for readers in future years to pinpoint).

Review form: Reviewer 3

Is the manuscript scientifically sound in its present form?

No

Are the interpretations and conclusions justified by the results?

No

Is the language acceptable?

No

Do you have any ethical concerns with this paper?

No

Have you any concerns about statistical analyses in this paper?

No

Recommendation?

Accept with minor revision

Comments to the Author(s)

-Whether the data are able to test the authors' proposed hypotheses by passing the approved outcome-neutral criteria (such as absence of floor and ceiling effects or success of positive controls)

The check of comparing pre-PVD to current PVD was done. There may be ceiling effects, as the authors discuss at length. I am not sure there are ceiling effects as opposed to the manipulation just not being effective, however.

-Whether the Introduction, rationale and stated hypotheses are the same as the approved Stage 1 submission

They appear to be consistent

-Whether the authors adhered precisely to the registered experimental procedures

They appear to have adhered well

-Where applicable, whether any unregistered exploratory statistical analyses are justified, methodologically sound, and informative

This appears consistent with the rest of the manuscript

-Whether the authors' conclusions are justified given the data

When I reviewed this manuscript as a pre-registered report, the authors insisted that pre-registered reports should only test the direct hypotheses, and thus they should not include additional measures to try to answer why the primary manipulation failed. Now the primary manipulation failed, and the authors are trying to piece together why. However, I am not sure that much of the speculation is warranted given the paper was only designed to answer the primary hypothesis. The discussion is a bit long, accordingly.

There are a few paragraphs where lack of measurement of relevant variables seem to play a major role in the speculation - "In our speculation, cultural differences may cause this gap between findings"

"The present study has several limitations on generalizing the findings"

I think most of these caveats in this section are fine and reasonable. Maybe the other speculations should be listed as limitations if they are unanswerable given the current data.

As a side note, while I missed a revision of the pre-registered part of the report, it seems that basically nothing was done to assess the psychometrics of the scale, and whether it makes sense to take means across all of the items in these scales.

There are many writing issues in the new text (discussion, results, etc). I am not going to list them all, but the writing should be clearer prior to publishing.

Abstract-

"Practise"

Is misspelled

Discussion-

"the reminder hardly changes IP-intention"

Informal, and hardly is not really the result. The result is does not change IP-intention or behavior

"Given the small nature of the effect size, it is quite likely that the effect of highlighting self-identity was not observed depending on experimental settings such as the present "

Difficult to parse

"performed meta-analysis"

Performed a

"These findings show that it is not only Japanese people who experience little effect of highlighting self-identity. "

This sentence seems to say the opposite of what is intended, as it suggests that Japanese people and other groups are unlikely to show an effect of highlighting self-identity. I think it means to say there is evidence that they do show such an effect.

Review form: Reviewer 5

Is the manuscript scientifically sound in its present form?

Yes

Are the interpretations and conclusions justified by the results?

Yes

Is the language acceptable?

Yes

Do you have any ethical concerns with this paper?

No

Have you any concerns about statistical analyses in this paper?

No

Recommendation?

Major revision

Comments to the Author(s)

I have read the stage 1 version of this registered report and was very curious to read about the outcome of this research project and the authors' conclusions. Overall, regarding all relevant aspects, the authors did comply to the registered procedures and I also could not detect deviations from the introduction or the at stage 1 registered hypotheses.

In general, I would recommend this paper for publication, however, there are several aspects which in my opinion would benefit from revision.

I will highlight these aspects in the following in more detail:

TOST-Approach: I appreciate that the authors use this approach for being able to draw conclusions about the absence of an effect because that's usually not possible within a frequentist framework. However, as I think TOST generally is not a standard analysis, I would appreciate it if the authors could explain their approach here in more detail. Especially, as far as I have understood it, in the TOST it is necessary to define effect sizes as boundaries so that it's possible to reject effects outside this boundary. I wonder which effect sizes the authors used? Also, I don't know what the phrase "there was no significant equivalence" exactly means. What are the implications of this result? Again, I value that the authors used this approach, but I think the authors need to add more information about what these findings actually imply.

PVD-Scale: The two subscales of the PVD-Scale point in different directions (although it should be highlighted that the effect of the germ aversion scale is really small, $d < 0.1$). In their hypothesis concerning the PVD-scale, it read to me as the authors would assume coherent effects for the two subscales, so it seems a rather unusual result that they diverge. To me the conclusion of these findings is as follows: Participants felt more susceptible to infections before the outbreak of COVID-19, but now they experience more discomfort in situations with potentially high germ-exposure (although again: to a very small extent). I'm aware that the authors attempt to find explanations for this oddity on p. 16/17, as they argue (if I understand it correctly) that because people experience more germ aversion, they avoid certain events and thus they feel less susceptible to infections disease as before. That's valid reasoning, but I think it does not explain why the effect size of perceived infectability is larger than germ aversion. I think there might be some more processes involved, and I think it would be valuable to further elaborate on that. I'd also be curious about the exact statistics of the two scales (means, SDs, inter-items correlations) as well as the exact wording of the items in the appendix since I think that warrants a better assessment of the results.

p. 17: I don't really understand the reasoning about the role of effect sizes in prior studies. It is about effect sizes or context effects? I agree that it's important to consider not only p-values but effect sizes, but what exactly is the link between effect sizes in prior studies and the results of the present study?

I think a very likely explanation for the results on the IP intention is the existence of ceiling effects, which the authors also adequately discuss. Irrespective of that the authors could not detect effects of highlighting self-identity, I find the results worth publishing as they can provide important insights for future studies. Especially because the study was conducted in Japan and cultural factors most likely influence health-related behavior, the study can make an important contribution to the understanding of health-related behavior.

Decision letter (RSOS-200793.R3)

Dear Dr Yonemitsu,

The editors assigned to your paper ("Warning 'Don't spread' vs. 'Don't be a spreader' to prevent the COVID-19 pandemic") has now received comments from reviewers. We would like you to revise your paper in accordance with the referee and Subject Editor suggestions which can be found below (not including confidential reports to the Editor).

Please submit a copy of your revised paper within three weeks (i.e. by the 08-Sep-2020).

- Data accessibility

It is a condition of publication that all supporting data are made available either as supplementary information or preferably in a suitable permanent repository. The data accessibility section should state where the article's supporting data can be accessed. This section

should also include details, where possible of where to access other relevant research materials such as statistical tools, protocols, software etc can be accessed. If the data has been deposited in an external repository this section should list the database, accession number and link to the DOI for all data from the article that has been made publicly available. Data sets that have been deposited in an external repository and have a DOI should also be appropriately cited in the manuscript and included in the reference list.

If you wish to submit your supporting data or code to Dryad (<http://datadryad.org/>), or modify your current submission to dryad, please use the following link:
<http://datadryad.org/submit?journalID=RSOS&manu=RSOS-200793.R3>

- **Competing interests**

- **Authors' contributions**

- **Acknowledgements**

- **Funding statement**

Kind regards,
 Royal Society Open Science Editorial Office
 Royal Society Open Science

on behalf of Chris Chambers
 Subject Editor, Royal Society Open Science
openscience@royalsociety.org

Associate Editor's comments (Professor Chris Chambers):

Associate Editor: 1

Comments to the Author:

Three of the original Stage 1 reviewers were available to assess the Stage 2 manuscript. In general, the reviewers find that the Stage 2 criteria are nearly met, but with some work especially required

to improve the clarity of reporting and the Discussion especially (including ensuring that the conclusions are justified by the evidence, that the arguments are logically coherent, and highlighting key limitations such as the observed ceiling effect). Concerning the point raised by Reviewer 2 about the unavailability of the accepted Stage 1 protocol, the URL to the Stage 1 protocol was stated as required at the end of the Abstract (and I have discussed this issue already with the reviewer), but I agree that more could be done to flag it for readers. Therefore, please also include in the Method section the sentence in the Abstract that lists the URL to the Stage 1 protocol.

Comments to Author:

Reviewers' Comments to Author:

Reviewer: 2

Comments to the Author(s)

See attached file. I have also pasted the unformatted version below (I recommend reading the file).

Review RSOS Stage 2

Overview:

As in my round 1 reviews, my main concern about the study is the lack of interpretability of the null results, given that I do not believe the manipulation check is sufficient to show that participants read and internalized the public health message (i.e. the manipulation – “don’t be a spreader” versus “don’t spread” or no message). As I pointed out in Stage 1, the manipulation check the authors used was more of an attention check and did not assess whether participants had actually read and understood the manipulation message. Additionally, as I flagged in Stage 1, the fact that the measurement scales were not piloted or validated means that the resulting ceiling effects (or other possible noise in the data) may have affected the interpretability of the results. However, I acknowledge that the editor accepted the manuscript in principle, including the manipulation check and measurement scales, so this cannot be held against the authors at this stage since they carried out their study as approved.

As detailed below, I believe the other most important issues to address are the way the main results are interpreted (need to make it clear that the results are not conclusive either way) and to make the discussion section more clear, understandable, and relevant.

RR Specific review questions:

Whether the data are able to test the authors’ proposed hypotheses by passing the approved outcome-neutral criteria (such as absence of floor and ceiling effects or success of positive controls)

As detailed in Stage 1 review, I do not believe the manipulation check was sufficient, and the authors also report a possible ceiling effect in one of their scales. These need to be acknowledged in the discussion at least, as unfortunately they weaken the evidential value of the study in my view.

Whether the Introduction, rationale and stated hypotheses are the same as the approved Stage 1 submission

I did not have enough time to read through the whole Stage 1 manuscript (and I also had to hunt to find the Stage 1 ms itself!), but a quick glance suggested it was similar at least. The hypotheses were the same.

Whether the authors adhered precisely to the registered experimental procedures

Everything I checked had adhered to the registered procedures, but in some cases the registration was under-specified (for instance, the equivalence test did not specify a SESOI, smallest effect size of interest to test against).

Where applicable, whether any unregistered exploratory statistical analyses are justified, methodologically sound, and informative

The only exploratory analyses were the extra equivalence tests for the comparisons between additional pairs of conditions – I would say these were justified and informative but as noted below they need to report more information on the tests and explain the interpretation of the tests.

Whether the authors' conclusions are justified given the data

The authors need to reframe their conclusions, given the equivalence tests they ran – the data are not conclusive either way, and the authors' conclusions should reflect that.

Data:

I was able to find and download the data and codebooks from the OSF, and at face value they look ok. However I did not actually try re-running the analyses on the data, due to the short turnaround of this review.

Registration:

There is no registration for this manuscript on the OSF link provided. There is a registration-style document on the OSF project from 4 May 2020 (around the time of the initial Stage 1 submission, I believe), and appears to be based on the original first draft of the manuscript before Stage 1 revisions. It is in a question and answer format (presumably one of the OSF preregistration formats) rather than in the format of a Stage 1 manuscript, despite the document name ("RSOS_COVID_RRStage 1_MS.pdf"). This lack of a formal registration (or at least an upload of the full accepted Stage 1 manuscript) makes it impossible for me (or other readers) to check consistency of the Stage 2 manuscript with the in-principle-accepted Stage 1 manuscript, which is one of my main tasks as a reviewer of a Registered Report. For the purposes of this review I am assuming that the manuscript submitted 6 June 2020 is the in-principle-accepted Stage 1 manuscript, and am comparing it to that. However, I recommend the authors to upload the in-principle-accepted Stage 1 manuscript to the OSF project as well.

Results:

Results: PVD – please report the means, standard deviations, and sample sizes for each sample you compared for each measurement. This could be in a table or the text.

Figure 3 – it would be helpful to label the "before" and "after" labels by the dates collected, e.g. ("Before COVID-19 onset: 1 January 2018") and ("After COVID-19 onset: 5 June 2020") or whatever the actual dates were.

On page 15 and page 16 when you discuss the PVD results, it would help to rephrase the results to keep the order of the timepoints constant, e.g., say score A was higher pre-pandemic compared to mid-pandemic, and score B was lower pre-pandemic compared to mid-pandemic. A fast or not-careful reader might misinterpret the results to indicate that both scales were higher in the pre-pandemic timepoint (that was my first impression on a quick reading) because they are just scanning for the words "higher" or "lower". I was especially confused by this in the wording on page 16, 2nd paragraph of discussion.

Page 15, lines 6-8 – please phrase the results of the equivalence tests as suggested in the TOSTER paper, and report the smallest effect size of interest that you tested for (i.e., the lower and upper bounds of equivalence that the TOST tested against) for each test. The TOST statistics are not meaningful without reporting that information. I would also request that you interpret/explain what these results mean for your claims, for readers who are not familiar with equivalence tests – in this case, if both the experimental test and the equivalence test have $p > .05$, then it suggests your data cannot conclusively say either that there is or is no meaningful effect, either way.

Discussion:

Page 16, Discussion (first paragraph) –

- Instead of “reduce” intentions/behaviors, perhaps “affect” would be better as it is more general (you expected the intervention might reduce some intentions/behaviors and increase others)
- The second sentence is confusing and not quite right. Firstly, you tested changes in behavior or intention scores, not the scores themselves. Secondly, the wording makes it sound like you were testing between behavior and intention scores, rather than each separately.
- In sentence 3, after you mention there is no statistical support for Hyp 1 and 2; to fully report your results, you should mention that there is also no statistical support AGAINST Hyp 1 and 2 (as per the non-significant equivalence tests)

Page 16, paragraph 3 – instead of launching straight into possible theoretical explanations for lack of conclusive results, I would first outline or emphasize the limitations of evidence in this study – that you did not find conclusive evidence in either direction, and so the effect size may be smaller than you guessed, and/or the data may be noisier than you had hoped. I think it is important to acknowledge this first, before speculating about theoretical or methodological concerns.

I really struggled to understand the 4th paragraph of the Discussion, page 16-17.

- The paragraph seems to argue two opposite conclusions – that according to PVD results people might have reduced infectious behavior (fear of infection) but also increased it (lower perceived susceptibility). Are you suggesting both could be true at the same time? I don’t see how that could be the case. And why would that explain why you got the results you did? The first argument (people already reduced high-risk infectious behaviors, as suggested by germ aversion on the PVD) is consistent with the point that you might have ceiling effects. But shouldn’t the second argument (that people might underestimate potential to be a spreader) be working against any ceiling effects of anti-spreading behavior/intentions?
- You wrote “The present study’s data of IP-intention score may tend to be close to a ceiling effect, and therefore germ aversion may influence IP intention rather than behaviour.” Why should a ceiling effect on intention scores indicate that germ aversion had an effect on it? And why should it indicate that germ aversion did NOT affect behavior? This seems like pure speculation. You do not have data for IP intentions or behavior from the previous PVD data collection, so you don’t know how changes in intentions or behavior might be related to germ aversion PVD levels.
- I found the second half of the paragraph hard to follow as well – I don’t understand how the logic leads you to your conclusions.
- Perhaps it would be better to just outline a simpler explanation/speculation in this paragraph -- that the ceiling effects left little room for differences in intention, and that the levels of attitudes around disease (as measured by PVD) might have an unknown effect on the results (and any further interpretation would be speculation).

Page 17, paragraph 2 – I’m not sure I understand the first sentence, “Moreover, the results of IP may reflect the degree of effect of highlighting self identity.” Do you mean that you may have gotten inconclusive results because the true effect size is very small? If so, I would just state that!

Page 18, paragraph 2 (limitations)

- please rethink/rephrase your conclusions about the study’s findings. You wrote “However, this indicates that at least the procedures in our study could not change IP-intentions and behaviour.” However, the results of the equivalence tests combined with the hypothesis tests indicate that you cannot rule out either outcome – the data are inconclusive. You could instead state that your data could not conclusively show that the procedures changed intentions or behavior. (Or perhaps this sentence could simply be omitted and the paragraph focus solely on the limitations.)
- instead of just considering internal validity (what methodological changes might result in a conclusive effect), what about also considering external validity – how well do the study stimuli reflect the way citizens might encounter such messaging “in real life” if a government or other

entity decided to adopt the self identity messaging? The current study design could potentially be considered a merit rather than a limitation in some ways, as perhaps it reflects the passive messaging and sometimes long time lags between message and behavior that you might find in a modest messaging campaign. Likewise, regarding the potential effects you discuss of increasing the frequency of exposure to messaging – instead of just speculating that it might increase likelihood of observing effects, it would be more interesting to me as a reader to consider how it relates to real world situations (e.g., the lab experiment you describe might provide information on how a well-funded, centralized and co-ordinated messaging campaign might affect responses.)

– in the limitations section, I would also suggest highlighting other ways the study methodology might be improved. By “improved,” I do not mean better chances of finding a significant effect, but rather a better reflection of whether the effect exists in the real world. What changes might allow you to make more definitive claims about the hypotheses you are testing? For instance, validating and refining the measures might have helped make the data less noisy, or having a better manipulation check might have allowed you to test whether people actually absorbed the message.

– Lastly, I would suggest that the authors also discuss the limitation that the ‘manipulation check’ cannot actually ascertain whether participants read the message, and the possibility that many participants simply did not read it.

Page 18, last paragraph – again, I do not follow the logic of this paragraph or understand the argument it is attempting to make. Could this be phrased more clearly and directly? For instance, I wasn’t sure if the low case numbers were meant as an argument that people were already following safe behaviors, or that low numbers meant people perceived low risk and therefore the messaging simply didn’t work at all. I also am unclear what is meant by “If the threat of COVID-19 spread increases, people can easily follow the reminders.” However these were not the only issues with the paragraph -- I’m sorry, I simply didn’t understand any of the points or how they follow on from each other.

Page 19, paragraph 1 – again, I am having trouble following the argument of this paragraph. Is the argument that self identity effects have been found in many cultures? Or that they have failed to be found in many cultures? I don’t follow the logic of the last several sentences either. Could this be rewritten more clearly and simply, perhaps just stating what evidence there already is about whether self-identity effects are cross cultural.

Page 19, paragraph 2 – again, the data do not quite support this conclusion, that the manipulation “has little effect on changing IP intention and behaviour.” Instead, the data simply do not provide support for (or against) the hypotheses

Minor points:

I recommend that, in the codebooks (description of data text files), the authors should add in information on the allowable range of responses (1-7 I believe?) and indicate what those numerical responses corresponded to (e.g., 1=strongly disagree, 7 = strongly agree). This would be very helpful but is not strictly required.

Page 6, top paragraph – the wording is a bit unclear. I suggest something like “As a secondary hypothesis for this research, we additionally predicted that the scores ... [etc.]”

Page 12, line 29 (Main Analysis) – could the first sentence clarify that the 2 mean scores for each measure are the before and after (or time 1 and time 2) measures? I did not understand this at first and had to read it a few times before realizing what it must mean.

Page 12, line 39 – the last word of the paragraph should be variables, plural. You have written variable (singular), which makes it sound like they have been combined – but I gather from the paragraph below that you actually did not combine, but ran separate tests for each.

Page 14, line 17-18 - I don't understand what this sentence means: "However, the total number of the participants excluded by each criterion did not match the total number of excluded participants." Could you briefly explain more what you meant, and why this is important?

Page 14, line 27 - I think the PVD results should be listed last in this section (after the intentions and behaviors results) as it is a secondary hypothesis. Put the most important, central results first.

When referring to the PVD in text (e.g., page 17 line 47, but elsewhere as well), instead of "current situation" it may be useful to say "during pandemic" or "several months into the COVID-19 pandemic", or even "June 2020" as it is unclear when exactly "current situation" refers to (and will be even harder for readers in future years to pinpoint).

Reviewer: 3

Comments to the Author(s)

-Whether the data are able to test the authors' proposed hypotheses by passing the approved outcome-neutral criteria (such as absence of floor and ceiling effects or success of positive controls)

The check of comparing pre-PVD to current PVD was done. There may be ceiling effects, as the authors discuss at length. I am not sure there are ceiling effects as opposed to the manipulation just not being effective, however.

-Whether the Introduction, rationale and stated hypotheses are the same as the approved Stage 1 submission

They appear to be consistent

-Whether the authors adhered precisely to the registered experimental procedures

They appear to have adhered well

-Where applicable, whether any unregistered exploratory statistical analyses are justified, methodologically sound, and informative

This appears consistent with the rest of the manuscript

-Whether the authors' conclusions are justified given the data

When I reviewed this manuscript as a pre-registered report, the authors insisted that pre-registered reports should only test the direct hypotheses, and thus they should not include additional measures to try to answer why the primary manipulation failed. Now the primary manipulation failed, and the authors are trying to piece together why. However, I am not sure that much of the speculation is warranted given the paper was only designed to answer the primary hypothesis. The discussion is a bit long, accordingly.

There are a few paragraphs where lack of measurement of relevant variables seem to play a major role in the speculation - "In our speculation, cultural differences may cause this gap between findings"

"The present study has several limitations on generalizing the findings"

I think most of these caveats in this section are fine and reasonable. Maybe the other speculations should be listed as limitations if they are unanswerable given the current data.

As a side note, while I missed a revision of the pre-registered part of the report, it seems that basically nothing was done to assess the psychometrics of the scale, and whether it makes sense to take means across all of the items in these scales.

There are many writing issues in the new text (discussion, results, etc). I am not going to list them all, but the writing should be clearer prior to publishing.

Abstract-

"Practise"

Is misspelled

Discussion-

"the reminder hardly changes IP-intention"

Informal, and hardly is not really the result. The result is does not change IP-intention or behavior

"Given the small nature of the effect size, it is quite likely that the effect of highlighting self-identity was not observed depending on experimental settings such as the present "

Difficult to parse

"performed meta-analysis"

Performed a

"These findings show that it is not only Japanese people who experience little effect of highlighting self-identity. "

This sentence seems to say the opposite of what is intended, as it suggests that Japanese people and other groups are unlikely to show an effect of highlighting self-identity. I think it means to say there is evidence that they do show such an effect.

Reviewer: 5

Comments to the Author(s)

I have read the stage 1 version of this registered report and was very curious to read about the outcome of this research project and the authors' conclusions. Overall, regarding all relevant aspects, the authors did comply to the registered procedures and I also could not detect deviations from the introduction or the at stage 1 registered hypotheses.

In general, I would recommend this paper for publication, however, there are several aspects which in my opinion would benefit from revision.

I will highlight these aspects in the following in more detail:

TOST-Approach: I appreciate that the authors use this approach for being able to draw conclusions about the absence of an effect because that's usually not possible within a frequentist framework. However, as I think TOST generally is not a standard analysis, I would appreciate it

if the authors could explain their approach here in more detail. Especially, as far as I have understood it, in the TOST it is necessary to define effect sizes as boundaries so that it's possible to reject effects outside this boundary. I wonder which effect sizes the authors used? Also, I don't know what the phrase "there was no significant equivalence" exactly means. What are the implications of this result? Again, I value that the authors used this approach, but I think the authors need to add more information about what these findings actually imply.

PVD-Scale: The two subscales of the PVD-Scale point in different directions (although it should be highlighted that the effect of the germ aversion scale is really small, $d < 0.1$). In their hypothesis concerning the PVD-scale, it read to me as the authors would assume coherent effects for the two subscales, so it seems a rather unusual result that they diverge. To me the conclusion of these findings is as follows: Participants felt more susceptible to infections before the outbreak of COVID-19, but now they experience more discomfort in situations with potentially high germ-exposure (although again: to a very small extent). I'm aware that the authors attempt to find explanations for this oddity on p. 16/17, as they argue (if I understand it correctly) that because people experience more germ aversion, they avoid certain events and thus they feel less susceptible to infections disease as before. That's valid reasoning, but I think it does not explain why the effect size of perceived infectability is larger than germ aversion. I think there might be some more processes involved, and I think it would be valuable to further elaborate on that. I'd also be curious about the exact statistics of the two scales (means, SDs, inter-items correlations) as well as the exact wording of the items in the appendix since I think that warrants a better assessment of the results.

p. 17: I don't really understand the reasoning about the role of effect sizes in prior studies. It is about effect sizes or context effects? I agree that it's important to consider not only p-values but effect sizes, but what exactly is the link between effect sizes in prior studies and the results of the present study?

I think a very likely explanation for the results on the IP intention is the existence of ceiling effects, which the authors also adequately discuss. Irrespective of that the authors could not detect effects of highlighting self-identity, I find the results worth publishing as they can provide important insights for future studies. Especially because the study was conducted in Japan and cultural factors most likely influence health-related behavior, the study can make an important contribution to the understanding of health-related behavior.

Author's Response to Decision Letter for (RSOS-200793.R3)

See Appendix J.

Decision letter (RSOS-200793.R4)

Dear Dr Yonemitsu:

It is a pleasure to accept your Stage 2 Registered Report entitled "Warning 'Don't spread' vs. 'Don't be a spreader' to prevent the COVID-19 pandemic" in its current form for publication in Royal Society Open Science.

COVID-19 rapid publication process:

We are taking steps to expedite the publication of research relevant to the pandemic. If you wish, you can opt to have your paper published as soon as it is ready, rather than waiting for it to be published the scheduled Wednesday.

This means your paper will not be included in the weekly media round-up which the Society sends to journalists ahead of publication. However, it will still appear in the COVID-19 Publishing Collection which journalists will be directed to each week (<https://royalsocietypublishing.org/topic/special-collections/novel-coronavirus-outbreak>).

If you wish to have your paper considered for immediate publication, or to discuss further, please notify openscience_proofs@royalsociety.org and press@royalsociety.org when you respond to this email.

Best regards,

on behalf of Professor Chris Chambers (Subject Editor)
openscience@royalsociety.org

Reviewer comments to Author:

Appendix A

Review for Registered Report entitled: Warning “Don’t spread” vs. “Don’t be a spreader” to prevent the COVID-19 pandemic (RSOS-200793)

I have suggested resubmitting with major revisions, because I believe the overall premise of the paper is (probably) valid and could be a useful contribution to the literature and to public health. However, there are many important changes that need to be made, and these could add up to a fair amount of work. I have given suggestions for how to implement these whenever possible. I have given the label “crucial” to all those changes I think must be implemented for the study to be methodologically sound (and by extension, for the paper to be publishable in RSOS or another reputable journal).

Below please find my comments about:

The scientific validity of the research question(s)

As far as I am aware, this seems like a scientifically valid research question. I do not have particular expertise in health messaging or theories about how self-identity affects behaviour, but the general principle of testing how behavioral intentions are affected by different messaging strategies (chosen based on underlying theory) seems like a sound basis for research. In addition to potentially helping improve public health messaging, this study would be a conceptual replication testing the generalizability of the study the authors seem to have adapted (Bryan, Adams, & Monin, 2013). That study appeared to be very underpowered and yet proposed a striking behavioural change based on a very simple manipulation, so it seems especially worthwhile to attempt a replication of any sort, including this conceptual replication. Experts in this field might be able to advise whether extensive, credible (e.g., preregistered) replication efforts have already been made—if so, then this study would be less important. But even if there is convincing evidence in the literature for this theory one way or another, I would suspect much of the research will have been done in WEIRD contexts (e.g., Western countries with undergraduates) so could be valuable to investigate it in Japan and in the context of a pandemic.

The logic, rationale, and plausibility of the proposed hypotheses

Again, I am not an expert in this content area specifically, but looking at the highly underpowered original paper (Bryan, Adams, & Monin, 2013) I am not particularly convinced of the previous evidence behind this general theory. In a quick literature search I found a couple of papers testing this theory in other contexts, all underpowered (and not preregistered as far as I could see). However, the hypothesis intuitively seems at least a little bit plausible, and I can see the rationale for testing different public health messages.

Crucial: However, the hypotheses themselves need to be more specific, and make sure they match with the proposed analyses to test them. H1 predicts that a scale measure will be higher in one condition as compared to two others, but the proposed analysis is of a change score between time 1 and time 2. This needs to be reflected in the wording of the hypothesis – that the *change from baseline after getting a reminder* will be larger for the spreader condition than the spreading or control condition. Same principle applies for H2. (These issues are repeated in Table 3). Similarly, the null hypotheses (H0) for hypotheses H1

and H2 need to be more specific. At the moment they assume that no conditions will differ from each other, but according to the alternative hypotheses, the null hypotheses need to be changed to specify only that the spreader condition will not differ from the spreading or control condition.

Crucial: At the moment, these hypotheses are also double-barrelled, i.e. you predict the spreader condition will differ from both spreading and control conditions. If one of the post-hoc comparisons you mention in the hypothesis is significant, but the other isn't, will you consider that evidence for your hypothesis? If so, they should be separated. If not, and you require both post-hoc comparisons to be significant, you should make that clear also (and then you may actually also want to increase your alpha level, since Tukey's test aims to keep the alpha level for each individual test at the set level, e.g., .05, whereas you are really testing the combination of two tests, and want the alpha level of those combined tests to be .05 – but it is probably ok not to bother with this, it will just be more conservative of a test.)

Crucial: Also, I would suggest additionally hypothesizing and testing whether the spreading condition differs from the control condition. If you find that it doesn't, and the spreader condition also does not differ from control, then that would be an indication that your manipulation perhaps did not work as you intended (i.e., that a simple reminder does not affect reporting of health behaviors and intentions). Even if you don't include this as a main hypothesis, you should at least plan to test whether spreading and control conditions differ as a positive control of sorts. (This test is also potentially useful to know for public health applications.)

I would also suggest that the authors make it clear that H3 is a secondary hypothesis (e.g. by calling it "secondary hypothesis" or putting in its own section) – it seems unrelated to the main point of this paper. I think it might be a useful thing to measure and report, but I'm not sure how it contributes to the aim of this paper. We know from actual behavior and many other sources of evidence that people are more concerned with disease. To be honest I would be completely shocked if scores on the PVD did not increase significantly from previous testing, and would question the validity of the measure if they did not. In fact I would treat the findings from any comparison of PVD pre- and post-pandemic as a very strong validity test of using this scale as a measure to compare population means (versus using it as an individual differences measure, which I suspect is the intention it was probably developed and validated for.)

The soundness and feasibility of the methodology and analysis pipeline (including statistical power analysis where applicable)

I have several major concerns with the study design and analysis.

- Why use the PVD covariate only if scores differ from pre-COVID-19 levels? It seems to me that either this covariate will be applicable and relevant, or not, regardless of overall levels – it is meant to essentially partial out individual differences. Why should use of it depend on whether *overall* levels of PVD have increased since the pandemic?
- And, why use the PVD covariate at all? You should make the case why you think perceived vulnerability to disease is noise that needs to be controlled for in this potential effect.

- The authors calculated power by choosing an effect size using the convention of halving the original effect. That is probably ok, but having read the original study I think there's a good chance the real effect might be even less than half the original. A better way would be to calculate power based on the smallest effect size of interest – that is, the smallest effect size that you think would make a practical or theoretical difference. Is there a way to assess practical/clinical significance here?
- I would not recommend using arithmetic problems as an attention check, for several reasons: it is an entirely different domain to what you want to measure (reading/comprehension), it will bias the sample against people who struggle with numeracy, which is a bigger population than you might think, and it will probably serve to catch attention since the numbers may stand out amongst text. Also, it sounds like this method has been used a lot previously, and online survey-takers evolve to spot previous attention check question styles, so it is good to use different types of ACQs wherever possible. Ideally ACQs have the same format as the questions around them, but ask a question with only one right answer. So, for instance you could ask something like “how many times have you...” and then finish with something impossible, then exclude everyone who does not answer 0.
- Crucial: The demographic information should be asked at the end of the survey – at least age and sex. This is important, as asking these at the beginning could affect the results by making e.g., participants' own age more salient to them and thereby influencing their perceived vulnerability to disease or behavioral intentions, as it is well known that age (and to some extent sex) is linked to health outcomes with COVID-19.
- Crucial: I have concerns about the validity of the proposed scale measures, Infection-Prevention and Outing Frequency. I would advise only running this study after the authors have done a pilot for these measures and ironed out the potential problems. See the responses to this tweet for a good collection of resources for validating measurement scales: <https://twitter.com/NicoleBarbaro/status/1258514761429352448> The items in the IP and OFI seem to have face validity but the authors have not provided any information on how they will check for internal reliability, nor sensitivity or any other form of measurement quality. A few specific issues jump out at me, but these are not the only issues:
 - 1) Why is the rating scale from “not applicable” to “highly applicable”? I realize this may be an issue of translation and perhaps this makes sense for Japanese respondents, but it does not sound like a balanced scale, nor one suited for the items. Usually I would expect behavioral intentions to be rated on a scale that measures agreement “strongly disagree” to “strongly agree,” for instance, or something equally balanced.
 - 2) How were the items for the Outing Frequency scale chosen? Many of them seem to overlap with the Infection Prevention scale, but not all. I can understand wanting to ask about both behavioral intentions and reported behavior, but I would expect you to ask about the same behaviors in both scales unless you give a reason why not. There may be a good reason, but the authors need to share it.
 - 3) Item 3 in the IP is a triple-barreled item – I would advise on putting each in its own question, as people may have different answers for each situation.

- 4) Some of the IP items are vague – what does “often” mean for hand-washing? I can’t immediately think of a way to improve that specific example, unfortunately.
- 5) I am concerned that the IP will have ceiling effects as people will want to present themselves in a socially desirable way. If a pilot study shows that there are ceiling effects, you may need to change the rating scale, re-word items to be less clearly socially desirable, and/or change the instructions to participants to emphasize honesty and anonymity.
- 6) The IP scale contains both approach- and avoid- contexts – that is, both active and passive behaviors (e.g., active = hand-washing, passive = avoiding crowds). It’s not clear to me that these would be affected equally through the proposed mechanism. Furthermore, the OF scale contains only the passive/avoidance behaviors. I would think that if you consider the active behaviors such as hand-washing to be important, they should be on both scales—or at least better justify why this discrepancy.
- Related to the above point: in Table 3, the first line in the “Question” column mentions inhibiting risky behaviors, but the related hypothesis concerns the IP which measures intentions for both inhibiting risky behaviors and activating/promoting risk-mitigating behaviors.

Whether the clarity and degree of methodological detail would be sufficient to replicate exactly the proposed experimental procedures and analysis pipeline

(NB: Many of these apply to the next criterion (researcher degrees of freedom/flexibility) as well)

- How will you test/check that participants understand Japanese well? This does not necessarily have to be a complicated or time-consuming check, but since you mention it as a criterion please explain how you will ascertain that.
- Define Tokyo exclusion criteria – will you advise potential participants on which areas “count” as Tokyo or is that already clear? E.g., will participants living in Tokyo suburbs be allowed to take part? Is there a clear way to demarcate (e.g., by listing the relevant postcodes or boundaries all participants will understand correctly?)
- On page 6 you state “It is important to set a limit on where participants live because this will allow us to ensure the validity of the dependent variable, the outing frequency index.” Please expand on this and explain how location can help you ensure the validity of the OFI.
- Crucial: Will assignment of conditions be randomized? If so, how? This is crucial. I would highly recommend randomizing because of course the inferential ability of the study could be drastically reduced if not.
- Crucial: Please give (a lot) more detail as to how the experimental manipulation (i.e., the behavior message/reminder) will be presented. You mentioned only, “After completing the first wave, we will present one of the infection-prevention reminders to the participants in the spreading and spreader conditions, whereas the participants in the control condition are given no reminder.” What will be the exact text of these messages, and how will the message be presented (e.g., audio or visual? If audio, what kind of voice? If visual, what size and color font, for how long?) How will you be sure participants have seen the message?

- Crucial: Please report how you will aggregate the scale measures – will you take the mean of all items? Or something else?
- Please report how you will validate the scale measures (at least the IP and OFI measures presented in tables 1 and 2)? E.g., will you check for internal reliability (for instance, by calculating Cronbach’s alpha and excluding the lowest-performing item until alpha reaches a certain acceptable threshold?)
- Crucial: Please provide the introductory text / instructions to participants for how they should answer the scale measures IP and OF. For instance, what period of time are these questions asking about? This text matters to how participants answer.

Whether the authors provide a sufficiently clear and detailed description of the methods to prevent undisclosed flexibility in the experimental procedures or analysis pipeline

- Good specification of sample size and how sampling plan will deal with exclusion rules

Whether the authors have considered sufficient outcome-neutral conditions (e.g. positive controls) for ensuring that the results obtained are able to test the stated hypotheses

Crucial: In their “Quality Checks” section, the authors state “it is not possible to set positive controls for any pandemic effect.” However, the main hypothesis and analysis in their study is not testing a pandemic effect, but rather a controlled trial (they do not state whether randomized) of an experimental manipulation on reported behavior and intentions.

Therefore it is necessary to include a test to check that the experimental manipulation has “worked.” They need to avoid the potential situation where critics could object to any null effects by arguing that maybe the participants didn’t actually see/read the message (or otherwise, more broadly, that it was not presented in an appropriate or externally valid way).

Crucial: To answer the first of those objections, you need a manipulation check to be sure participants have seen (and ideally also absorbed) the message. I would suggest asking them to type it back into the survey system after they read (or hear?) it. You could present it on one page for a minimum of a certain length of time, (say 5 seconds) then when participants click to the next page, ask them to repeat the text. This will also serve not only to test attention, but to (hopefully) make the message more salient.

Crucial: There is another issue related to positive controls, in the “Interpretation given different outcomes” column of Table 3. Here the authors raise the possibility that a null outcome would either signal a true null effect, or (if I understand them correctly), a ceiling effect where perceived vulnerability to disease is so high that it cannot be increased. I am confused about this interpretation because 1) the outcome measure is IP and OF, not PVD (and the authors have not made it clear how they expect PVD to be related to the main outcome measures, nor have they indicated there is any evidence for it)—so why would they expect null results to be related to ceiling effects in PVD?; 2) if they indeed expect a ceiling effect of any of the measures, then why haven’t they included any plan to mitigate this possibility?

A plan to detect and mitigate potential ceiling effects in the outcome measures will be necessary to making sure the results are interpretable, in either case. As I suggested above, a pilot test (a single cross-sectional survey) of the outcome measures will show whether the IP and OF have ceiling effects. It does not matter a lot if the PVD has ceiling effects, as this

was only intended as a covariate in any case; ceiling effects would just make it a not very useful covariate but (I think) shouldn't compromise the integrity of the experiment.

Lastly, as I mentioned in the section on Hypotheses, above, the analysis plan should include a test between the spreading and control conditions (for each of the outcome measures). This will help determine whether any kind of message, presented in the way the study does, can have an effect on health behaviors and intentions, and in turn this can help interpret the results if no evidence is found to support the main hypotheses.

Minor points:

(Note that when I refer to page numbers it is the numbering from the original document next to the running head, not the numbers imposed by the PDF creation.)

Page 3, line 53: you appear to be claiming your findings already show something, but you have not run the study yet. Please rephrase – for instance, you could state that your study “will aim to show whether it may be possible...”

At the top of page 6, you mention an exclusion “criterion detailed in (7).” I didn't know where to find this – could you list the section it is in, or make it clear whether you are referring to page number, etc.

Please move the sentences explaining the details about the PVD from the Hypotheses section to Materials and Procedure (or create a Measures section and explain all measures there).

In the IP scale, item 4, I am not sure what “I will ventilate regularly” means. I understand this is translated from Japanese, so I assume it will make more sense to participants!

Appendix B

Warning "Don't spread" vs. "Don't be a spreader" to prevent the COVID-19 pandemic

Review Stage 1 Registered Report

In the submitted proposal, the authors suggest that verbally emphasizing self-identity might play a role in motivating individuals to comply with the government's guidelines regarding the containment of COVID-19 cases. To investigate their assumption, they propose to implement an intervention and randomly assign individuals into one out of three groups: In one group, they are presented with the reminder "don't be a spreader" (self-identity related), in the second group, they are presented with the reminder "don't spread" (self-identity unrelated) and the third group serves as a control group without any reminder. Effectiveness of this intervention is planned to be examined by assessing individuals' agreement with the specific guidelines provided by the Japanese Ministry of Health, Labor and Welfare (COVID-19 IP scale), which is assessed at two timepoints (before and after the intervention).

In general, I think the proposed research study could make a valuable contribution with regard to the questions how individuals can be motivated to comply with guidelines and recommendations by political and scientific institutions to contain the spread of COVID-19. Even though the expected effect-sizes might be small: Regarding the current global health crisis every measure that even in the slightest can contribute to lowering infection rates and thus prevent deaths, is worth being taken seriously.

However, without doubting the potential value this research might have, I think that the submitted manuscript would benefit from a comprehensive revision and that the authors might also want to reconsider the statistical approach they suggest. In the following I will highlight my concerns in more detail:

Majors:

Design/Statistical Approach:

As far as I understand it, the proposed design entails a within-subjects factor:

The main dependent variable (COVID-19 IP Scale) is measured in the first wave as well as in the second wave one week later. However, the authors address their design as a between-subject design. They propose to calculate a COVID-19 IP index by subtracting the IP Scale Scores of the first wave from those of the second wave and to perform an ANOVA/ANCOVA with the COVID-19 IP index as a dependent variable.

In my opinion, this approach has major drawbacks and I would strongly recommend to apply analyses that are more appropriate for within-subject designs, such as a one-way repeated measures ANOVA or mixed models.

Between-subject designs have substantial power deficits compared to within-subjects designs, which is why this aspect should already be taken into account for power/sample size considerations. In my opinion, this is actually one of the major benefits of choosing a repeated-measures design, and I'm irritated that the authors would choose this approach and nevertheless analyze their data with a simple one-way ANOVA.

Theoretical Background:

The introduction part is extremely short and, in my opinion, lacks a proper theoretical foundation. The authors introduce the concept of "self-identity" as a basis for their intervention, but don't explain what this concept actually entails, nor do they refer to relevant literature in this field. They provide one reference from a study that applied a similar intervention in a different context, but don't elaborate on the psychological underpinnings that drive this process. The basic assumption that people strive for a positive self-image would be

at least worth mentioning in my view. To sum it up, in my opinion the current content of the introduction makes no strong claim for the hypotheses; it's difficult to evaluate their plausibility if there barely is a theoretical foundation to refer to.

Minors:

ANCOVA:

What is the authors' intention in including the "perceived vulnerability to disease" (PVD) as a covariate? How do they expect this variable to influence their depended variable? I suspect it's about reducing error variance, since the PVD most likely will influence individuals' behavior and their compliance with guidelines. But why is it only included as a covariate if the PVD scores have changed during the pandemic compared to before? I would appreciate if the authors would elaborate in more detail why they choose this approach.

Confidentiality:

How is participants' data linked? To me it reads like email addresses are used to link the data, which would not be an appropriate approach since it clearly contains identifiable information. Please state if an anonymous code is used to link the data.

Scales:

On what scale is the Outing frequency scale measured?

Plus: Why does the number of scale point vary between the different scales? In case there's no strong reason for it, I always find it unnecessarily confusing for participants.

Hypotheses:

"H0: If the COVID-19 pandemic does not affect perceived vulnerability to disease, there will be no significant differences in the PVD scales scores during and before the COVID-19 pandemic"

There's an assumption in this hypothesis that does not belong in there, since it's correlational data. It still might be that there are significant differences in the PVD that is not related to COVID-19. I would recommend to stick with the wording from H1.

Hypotheses 1 refers to the COVID-19 IP Index, but at this point this measure has not been introduced yet.

Summary

To address the following points required by the journal and recapitulate the above-mentioned aspects:

1. *The scientific validity of the research question(s)*

Generally valid and might make a valuable contribution in the current situation; but please note my remark regarding lack of theoretical foundation, which makes it difficult to evaluate this aspect.

2. *The logic, rationale, and plausibility of the proposed hypotheses*

They basically are derived from the outcome of one similar prior study in a different context. The authors don't elaborate on other existing research and literature, and thus

it's difficult to evaluate the hypotheses' plausibility since in my view the manuscript lacks a proper theoretical foundation.

3. *The soundness and feasibility of the methodology and analysis pipeline*

Please see my statement on design issues above.

4. *Whether the clarity and degree of methodological detail would be sufficient to replicate exactly the proposed experimental procedures and analysis pipeline.*

The authors do provide their materials and replication should generally be feasible. However, based on the methods section, the exact procedure of the study seems a bit hard to reproduce, hence e.g. a Flow Chart might simplify that.

5. *Whether the authors provide a sufficiently clear and detailed description of the methods to prevent undisclosed flexibility in the experimental procedures or analysis pipeline*

In my opinion the methods section would profit from depicting the procedure in a more chronological order and in more detail.

6. *Whether the authors have considered sufficient outcome-neutral conditions (e.g. positive controls) for ensuring that the results obtained are able to test the stated hypotheses*

In my opinion the authors meet this criterion since they include three different groups plus pre- and post-measures.

Appendix C

27th May, 2020

Dear Dr. Chambers:

We sincerely appreciate the reviewers' ultrafast, valuable comments on our manuscript entitled 'Don't spread' vs. "Don't be a spreader" to prevent the COVID-19 pandemic' (RSOS-200793) which is being considered for proceeding to Stage 2 Registered Reports in *Royal Society Open Science*. Based on the comments, we have substantially revised the manuscript and have provided an amended version. All edited portions of the manuscript are highlighted in blue. Our individual responses to each of the comments by you and the reviewers are listed as follows.

Responses to Associate Editor

Comments & Replies

1-1. Five reviewers with a range of expertise (from field specialists to methodologists) have now assessed the manuscript, and let me begin by offering my deepest thanks to the reviewers for providing such high quality assessments on such an extraordinary timescale. All of the reviewers find merit in the proposal while also raising concerns that span the full range of Stage 1 review criteria. Some of the most significant weaknesses to address include the strength of the theoretical rationale, the precision of hypotheses (and mapping of hypotheses to specific analyses), the sampling plan (including the definition of the smallest effect size of interest), the validity of the measures, and appropriateness of the statistical analysis plan, and the overall level of methodological detail.

Substantial improvements in all these areas (and others, as identified in the reviews) will be needed to achieve Stage 1 in-principle acceptance, yet I am convinced by the overall enthusiasm and constructive nature of the reviews that the proposal is sufficiently promising to offer the authors this opportunity. A major revision is therefore invited.

Reply: We thank the Associate Editor for giving us this opportunity to revise our paper. Moreover, we would like to express our sincere gratitude to all the reviewers. Reviewer 2, for example, gave us a great deal of constructive advice in this extremely short time, and the comments of the other reviewers were truly helpful, as each of them brilliantly identified fatal problems with the first manuscript that we were unaware of. We have replied to the comments carefully and substantially revised the manuscript based on the comments. The following is a summary of some of the most significant changes:

1. Theoretical rationale: This was mainly based on the comments of Reviewer 1 and Reviewer 5. The introduction in the previous version was based on a relatively narrow range of literature and had a tenuous theoretical position. Therefore, we have clarified the position of the present study by dealing more broadly with previous discussions of the persuasion effect of messages, as well as the subsequent work of Bryan et al. (2013).
2. Validity of the measures: This was mainly pointed out by Reviewer 2. In the present study, we will use COVID-19 recognition and behaviour scales to measure infectious spreading attitudes and behaviours. Our aim here is to determine how to enhance the efficacy of the instructional methods provided by the Ministry of Health, Labour and Welfare of Japan to encourage the acceptance of administrative instructions. Therefore, it is quite natural and valid to use scale items (i.e. the COVID-19 IP scale) in accordance with the administrative guidance of the Ministry of Health, Labour and Welfare. Therefore, we modified the configuration of the scale items so that a complete correspondence between attitude and behaviour could be made. In particular, we clearly stated that we will use two scales to measure behavioural intentions (the IP-intention scale) and reported behaviours (the IP-behaviour scale) of infection prevention in the revised manuscript, and eliminated unnecessary and overlapped items (e.g. the outing frequency scales in the previous version of the manuscript) according to Reviewer 2's suggestions. Moreover, we will use a Likert scale for both scales to ensure statistical consistency with other items.
3. More precisely, the hypotheses are as follows: Hypothesis 1: If highlighting self-relevance corrects our intention, the change of the IP-intention scale for behavioural intentions from the 1st wave (baseline) to the 2nd wave will be significantly larger in the spreader condition than in the spreading and control conditions. The change from the 1st wave (baseline) to the 2nd wave will not significantly differ between the spreading and control conditions.
H0: If highlighting self-relevance does not influence our intention, there will be no significant differences in the IP-intention scale between the conditions.
Hypothesis 2: If highlighting self-relevance corrects our behaviour, the change of the IP-behaviour scale from the 1st wave (baseline) to the 2nd wave will be significantly larger in the spreader condition than the spreading and control conditions. The change from the 1st wave (baseline) to the 2nd wave will not significantly differ between the spreading and control conditions.
H0: If highlighting self-relevance does not influence our behaviour, there will be no significant differences in the IP-behaviour scale between the conditions.
4. Sampling plan: This was mainly pointed out by Reviewers 1 and 2. In the previous version, the statistical power was set a little lower (but the effect size was originally set much lower), but in the modified version, the power is raised to the maximum possible

(i.e., $1-\beta = .99$). In addition, we have removed the stopping rule. This has resulted in greatly improved statistical reliability.

5. Appropriateness of the statistical analysis plan: This was mainly pointed out by Reviewers 2 and 3. After receiving a number of convincing arguments and taking them into consideration, we decided not to submit the PVD as a covariate to ANOVA. In other words, we have decided not to conduct ANCOVA. This is also related to the theoretical rationale issue in 1, as it was less reasonable to treat PVD as a covariate.

Responses to Reviewer #1

Comments & Replies

- 1-1. In this manuscript and in the context of COVID-19 sanitary crisis, the authors want to compare the efficacy of prevention messages making self-identity salient (“don’t be a spreader”), compared to messages making self-identity less salient (“don’t spread”). When it comes to this crisis, I do think that it is a worthy goal for the field to conduct psychological research appropriately. I think that this research program is the kind of program that we need in these times, it is a straightforward intervention that could easily be adopted. However, even if the hypothesis the authors make sounds plausible, the current amount of evidence in their literature review is too light to consider testing this intervention right now. Overall, I have concerns that it is too soon to accept this paper, but if enough evidence is found to test this hypothesis, I think the results could be very interesting. Besides the literature review, the procedure seems fair regarding the test of the hypothesis authors want to test. It also has to be noted that the manuscript is well written and is easy to understand. Below, I describe several suggestions that, I think, could improve this program.

In their introduction, the authors discuss work by Bryan, Adams, and Monin (2013) to evoke the possibility that framing a prevention message in a way that makes identity salient should be more effective compared to a message that does not. Unfortunately, Bryan et al.’s (2013) empirical paper is the only work cited as the rationale for this study. Authors should mention whether further research has followed Bryan et al.’s (2013) work and they should also discuss the kind of effect size one can expect from such intervention. Depending on the kind of effect sizes that we can expect from this literature, it is an occasion to build a case for their research program. In the current version of the manuscript, the readers would understand why it is important to investigate processes making people more likely to adopt behavior limiting the spread of the pandemic, but I am afraid they might still wonder why focusing on this intervention and not another one. I think that this manuscript would gain in quality if authors conduct a more in-depth review of the literature, as it would help the readers (including policymakers) to know what they can expect from it.

Reply: We thank Reviewer 1 so much for his/her very quick review during these difficult times. We also deeply appreciate Reviewer #1 for providing positive comments on our study and important suggestions. One reason we focused on how to highlight self-identity is the simplicity and general versatility of simply manipulating the wording of instructions. According to a recent study, manipulating the expressions of messages to appeal to citizens' emotions is effective in promoting social distancing behaviour (Heffner et al., 2020). However, it is important to find a more useful method, given the circumstances in different countries around the world, such as cultural, linguistic, and legal differences. Thus, making self-identity salient in messages is a very simple way to intervene with large numbers of people. In addition, the effectiveness of this intervention has been repeatedly verified in other studies (Bryan, Master, & Walton, 2014; Savir & Gamliel, 2019). In particular, Savir and Gamliel (2019) could succeed in replicating the study of Bryan et al. (2013). Before Bryan et al. (2013), Bryan and colleagues investigated the effect of highlighting self-identity reminders on voting behaviours (voting vs. being a voter), finding that the reminder 'be a voter' elevated actual voter turnout (Bryan, Walton, Rogers, & Dweck, 2011). On the other hand, another intervention is to evoke mortality in messages based on terror management theory (Greenberg et al., 1994). However, recent studies (Klein et al., 2019; Sætrevik & Sjøstad, 2019) have failed to replicate this effect. Accordingly, in the present study investigating infection-prevention reminders, we avoided the use of mortality salience as the target manipulation as dangerous. Therefore, we considered highlighting self-identity reminders as an obviously effective way to promote behavioural change that can also be applied in the COVID-19 pandemic. Therefore, we considered highlighting self-identity reminders to be an obviously effective way to promote behavioural change that can also be applied in the COVID-19 pandemic. We added the above explanation and further literature such as Bryan et al. (2013) to the introduction section in our revised manuscript. We believe that these points make it easier for a wide range of readers to understand.

1-2. Besides the literature review, I think that authors could improve some part of their protocol. In their experiment, authors want to compare the two messages described above ("don't be a spreader" vs. "don't spread"), along with a control condition, in an experiment with a longitudinal design (wave 1: exposure to the message; wave 2: measures, one week later). Authors will recruit their participants using their email address, so they can send a reminder for the second wave. Given the sensitive nature of information related to the adoption of behaviors limiting the spread of COVID-19, it is critical that the authors do everything they can, so the identifying information is not linked to the data on COVID-19. Authors should be explicit about it. Regarding the data collected during the experiment, we don't know whether the data set collected for this experiment will be publicly available. Authors should mention it in their manuscript.

Reply: As Reviewer 1 pointed out, we should show that identifying information is ethically sensitive in this survey. Certainly, our description could be misleading and have insufficient information as if the identifying information is linked to data on the COVID-19 and made public. However, we will conduct our survey to ensure informed consent and specify that participants' email addresses will be collected anonymously before participation in the survey. In addition, we will explain that participants' email addresses will be used only to contact them for the 2nd wave survey, not for data purposes. Therefore, identifying information will not be linked to the data on COVID-19 and will not be included in our open data. We have clearly stated these points in the Participants section.

1-3. Authors conduct an a priori power analysis to decide what kind of sample size they need. They initially decide to compute the sample size needed based on Bryan, Adams, & Monin (2013) and other considerations, but end up doubling this sample size (for valid reasons). I think the reader would appreciate a sensitivity analysis on top of the information given. Given the sample size authors decided to use, what is the effect size authors have 80% statistical power to detect?

Reply: Based on the sample size we determined, we performed a sensitivity analysis ($\alpha = 0.05$, $1-\beta = 0.80$, $N = 1890$, number of groups = 3), yielding an effect size of $f = 0.071$ with 80% statistical power. Following this suggestion, we have added this information to the Participants subsection.

1-4. Regarding the analysis of their data, I think some parts could be improved. Authors will use a longitudinal design where they expect participants to come back 1-week after a first wave. However, we don't know if the authors will adopt exclusion criteria depending on when participants decide to complete the second part of the study. It seems especially important because we know that things can move quickly in the current time, and the time when participants decide to take part in the second part of the study could carry a lot of variance). Authors should address this point in their preregistration.

Reply: Our description of the criteria for data exclusion was definitely insufficient. Based on these comments, we have established the criterion that participants who did not respond to the second wave survey within 24 hours of requesting it would be excluded from the data. We added this information in the exclusion criteria section. This point is important to maintain the transparency of data exclusion in our registered study.

1-5. Page 6, authors note that if they end up recruiting more than 1380 participants, they would use only the data from the first 1380 participants (based on timestamp), hence excluding some participants. From an ethical point of view, I would argue against the exclusion of the participants. Indeed, removing the participants from the analysis would 1) decrease the

statistical power of the analysis, 2) waste participants' time and the resources engage in this study. I suggest adopting a more inclusive criterion.

Reply: Just as Reviewer 1 said, we certainly should not remove the excess data above the required sample size with regard to statistical power and participants' cost. Thus, we have decided to release the stopping rule and data on participants in excess of 1890 (considering another reviewer's suggestion, we have changed the maximum sample size from the previous version of the manuscript). We removed the statement that only the data for the first 1890 people will be used (based on timestamps) from the revised manuscript.

1-6. Authors suggest using an ANCOVA (instead of an ANOVA) to test their hypothesis while controlling by the perceived vulnerability to disease. However, it is not entirely clear why it is important to do so, as authors are not controlling for a confound (the key independent variable is experimentally manipulated) and the experiment seems appropriately powered.

Reply: We thank Reviewer 1 for this important opinion on the analysis. As Reviewer 1 pointed out, it is unnecessary to include the PVD scores because our experimental design has sufficient power. Given his/her opinion and Reviewer 2's suggestion, we have decided not to use the PVD scale scores as a covariate (Please see our reply to Reviewer #2's comments no. 2-3 for details). Therefore, we will analyse the data using only one-way ANOVAs instead of ANCOVAs.

1-7. Moreover, regarding the statistical power of the preregistered analysis, authors are planning on testing the omnibus effect related to their experimental manipulation and follow up with a Tukey's multiple comparison test. Tukey's test is a correction of the alpha rate that is needed when one test multiple non-orthogonal hypotheses to control for the type I error. If authors want to test their hypotheses while controlling for an alpha rate of 5% and without being too conservative, they should consider using contrasts to test orthogonal hypotheses (see Rosnow & Rosenthal's work). A contrast that seems appropriate here would be the treatment contrast, as it could answer both a pragmatic question (i.e., is a prevention message making self-identity salient better than nothing?) and a theoretical one (i.e., is a prevention message making self-identity salient better than one that does not; but see also, Helmert contrasts).

Reply: We thank this reviewer for this significant suggestion for multiple comparison tests. Accordingly, we will use the test of orthogonal contrasts as a multiple comparison test. Among the orthogonal contrasts test, Scheffé's *F* test is more appropriate to contrast all the conditions. We have therefore adopted Scheffé's *F* test and added this information to the *Main Analysis* subsection.

References

- Bryan, C. J., Master, A., & Walton, G. M. (2014). “Helping” versus “being a helper”: invoking the self to increase helping in young children. *Child Development*, 85(5), 1836–1842.
- Bryan, C. J., Walton, G. M., Rogers, T., & Dweck, C. S. (2011). Motivating voter turnout by invoking the self. *Proceedings of the National Academy of Sciences of the United States of America*, 108(31), 12653–12656.
- Greenberg, J., Pyszczynski, T., Solomon, S., Simon, L., & Breus, M. (1994). Role of consciousness and accessibility of death-related thoughts in mortality salience effects. *Journal of Personality and Social Psychology*, 67(4), 627–637.
- Heffner, J., Vives, M. L., & FeldmanHall, O. (2020). Emotional responses to prosocial messages increase willingness to self-isolate during the COVID-19 pandemic. *PsyArXiv*, <https://doi.org/10.31234/osf.io/qkxvb>
- Klein, R. A., Cook, C. L., Ebersole, C. R., Vitiello, C. A., Nosek, B. A., Chartier, C. R., Christopherson, C. D., Clay, S., Collisson, B., Crawford, J., & al., E. (2019). Many Labs 4: Failure to replicate Mortality salience effect with and without original author involvement. *PsyArXiv*, <https://doi.org/10.31234/osf.io/vef2c>
- Savir, T., & Gamliel, E. (2019). To be an honest person or not to be a cheater: Replicating the effect of messages relating to the self on unethical behaviour. *International Journal of Psychology: Journal International de Psychologie*, 54(5), 650–658.
- Sætrevik, B., & Sjøstad, H. (2019). Failed pre-registered replication of mortality salience effects in traditional and novel measures. *PsyArXiv*, <https://doi.org/10.31234/osf.io/dkg53>

Responses to Reviewer #2

Comments & Replies

- 2-1. I have suggested resubmitting with major revisions, because I believe the overall premise of the paper is (probably) valid and could be a useful contribution to the literature and to public health. However, there are many important changes that need to be made, and these could add up to a fair amount of work. I have given suggestions for how to implement these whenever possible. I have given the label “crucial” to all those changes I think must be implemented for the study to be methodologically sound (and by extension, for the paper to be publishable in RSOS or another reputable journal).

Below please find my comments about:

The scientific validity of the research question(s)

As far as I am aware, this seems like a scientifically valid research question. I do not have particular expertise in health messaging or theories about how self-identity affects behaviour, but the general principle of testing how behavioral intentions are affected by different messaging strategies (chosen based on underlying theory) seems like a sound basis for

research. In addition to potentially helping improve public health messaging, this study would be a conceptual replication testing the generalizability of the study the authors seem to have adapted (Bryan, Adams, & Monin, 2013). That study appeared to be very underpowered and yet proposed a striking behavioural change based on a very simple manipulation, so it seems especially worthwhile to attempt a replication of any sort, including this conceptual replication. Experts in this field might be able to advise whether extensive, credible (e.g., preregistered) replication efforts have already been made—if so, then this study would be less important. But even if there is convincing evidence in the literature for this theory one way or another, I would suspect much of the research will have been done in WEIRD contexts (e.g., Western countries with undergraduates) so could be valuable to investigate it in Japan and in the context of a pandemic.

The logic, rationale, and plausibility of the proposed hypotheses

Again, I am not an expert in this content area specifically, but looking at the highly underpowered original paper (Bryan, Adams, & Monin, 2013) I am not particularly convinced of the previous evidence behind this general theory. In a quick literature search I found a couple of papers testing this theory in other contexts, all underpowered (and not preregistered as far as I could see). However, the hypothesis intuitively seems at least a little bit plausible, and I can see the rationale for testing different public health messages.

Reply: First, we would like to express our gratitude to Reviewer 2 for this very fast review in this difficult time. In addition, we also greatly appreciate his/her positive feedback, valuable comments, and suggestions. Based on what he/she pointed out, our manuscript has been greatly improved.

2-2. Crucial: However, the hypotheses themselves need to be more specific, and make sure they match with the proposed analyses to test them. H1 predicts that a scale measure will be higher in one condition as compared to two others, but the proposed analysis is of a change score between time 1 and time 2. This needs to be reflected in the wording of the hypothesis – that the change from baseline after getting a reminder will be larger for the spreader condition than the spreading or control condition. Same principle applies for H2. (These issues are repeated in Table 3). Similarly, the null hypotheses (H0) for hypotheses H1 and H2 need to be more specific. At the moment they assume that no conditions will differ from each other, but according to the alternative hypotheses, the null hypotheses need to be changed to specify only that the spreader condition will not differ from the spreading or control condition.

Crucial: At the moment, these hypotheses are also double-barrelled, i.e. you predict the spreader condition will differ from both spreading and control conditions. If one of the posthoc comparisons you mention in the hypothesis is significant, but the other isn't, will you consider that evidence for your hypothesis? If so, they should be separated. If not, and you require both post-hoc comparisons to be significant, you should make that clear also (and then you may

actually also want to increase your alpha level, since Tukey's test aims to keep the alpha level for each individual test at the set level, e.g., .05, whereas you are really testing the combination of two tests, and want the alpha level of those combined tests to be .05 – but it is probably ok not to bother with this, it will just be more conservative of a test.)

Crucial: Also, I would suggest additionally hypothesizing and testing whether the spreading condition differs from the control condition. If you find that it doesn't, and the spreader condition also does not differ from control, then that would be an indication that your manipulation perhaps did not work as you intended (i.e., that a simple reminder does not affect reporting of health behaviors and intentions). Even if you don't include this as a main hypothesis, you should at least plan to test whether spreading and control conditions differ as a positive control of sorts. (This test is also potentially useful to know for public health applications.)

Reply: According to Reviewer 2's important comments and suggestion, we have made H1 and H2 more specific: The changes of the IP-intention scale and the IP-behaviour scale from the 1st wave (baseline) to the 2nd wave will be larger in the spreader condition than in the spreading and control conditions. Moreover, we have clarified the null hypotheses (H0) for H1 and H2: The change from the 1st wave (baseline) to the 2nd wave will not significantly differ between the spreading and control conditions. This is because previous studies (e.g., Bryan et al., 2013) have demonstrated that the self-related reminder suppressed unethical behaviours in comparison with the less-self-related reminders including the control condition. Thus, we will perform post-hoc tests for all the pairs, for which we must avoid alpha inflation. Following Reviewer 1's suggestion, we will adopt Scheffé's *F* test, which is more appropriate to contrast all the conditions, instead of Tukey's test. We have also clarified these points in the revised manuscript. These points are mentioned in our reply to his/her comments No. 1-7 and 2-26.

2-3. I would also suggest that the authors make it clear that H3 is a secondary hypothesis (e.g. by calling it "secondary hypothesis" or putting in its own section) – it seems unrelated to the main point of this paper. I think it might be a useful thing to measure and report, but I'm not sure how it contributes to the aim of this paper. We know from actual behavior and many other sources of evidence that people are more concerned with disease. To be honest I would be completely shocked if scores on the PVD did not increase significantly from previous testing, and would question the validity of the measure if they did not. In fact I would treat the findings from any comparison of PVD pre- and post-pandemic as a very strong validity test of using this scale as a measure to compare population means (versus using it as an individual differences measure, which I suspect is the intention it was probably developed and validated for.)

Reply: We thank Reviewer 2 very much for these valuable comments on Hypothesis 3. Certainly, Hypothesis 3 is a secondary hypothesis and PVD is not the main scale of this study, as

Reviewer 2 pointed out. It will be clearly stated in the revised manuscript that the variation in PVD before and after the beginning of the COVID-19 pandemic is a secondary hypothesis. In addition, we will not use the PVD as a covariate following his/her comments. The PVD will be only used for comparing the PVD scores between the current and non-pandemic situations, and we will discuss the effect of self-related reminders based on these comparisons (see our reply to 3-1 in more detail). Thanks to these significant suggestions, we are able to focus more on the purpose and key variates of the present study.

2-4. **The soundness and feasibility of the methodology and analysis pipeline (including statistical power analysis where applicable)**

I have several major concerns with the study design and analysis.

- Why use the PVD covariate only if scores differ from pre-COVID-19 levels? It seems to me that either this covariate will be applicable and relevant, or not, regardless of overall levels – it is meant to essentially partial out individual differences. Why should use of it depend on whether overall levels of PVD have increased since the pandemic?

- And, why use the PVD covariate at all? You should make the case why you think perceived vulnerability to disease is noise that needs to be controlled for in this potential effect.

Reply: According to these valuable comments on the analysis design, we have decided not to use the PVD scale as a covariate, as we replied in 2-3. This comment reminded us that investigating the effects of PVD is not the main theme of our study. We will only use PVD scale to compare the data before the COVID-19 pandemic.

2-5. The authors calculated power by choosing an effect size using the convention of halving the original effect. That is probably ok, but having read the original study I think there's a good chance the real effect might be even less than half the original. A better way would be to calculate power based on the smallest effect size of interest – that is, the smallest effect size that you think would make a practical or theoretical difference. Is there a way to assess practical/clinical significance here?

Reply: Exactly, the smallest effect size of interest (SESOI) is often discussed (e.g. Lakens, 2014). There is still no common understanding of how to calculate it, particularly in case of ANOVA (e.g. Lakens, 2014; Perezgonzalez, 2017). Here, we consider half the original effect (i.e. Cohen's $f = .151$) as the practical SESOI, which is sufficiently small. Perhaps Reviewer 2 is concerned that our sample size would be too small to detect the effect. If so, we agree with this concern and thus increased the statistical power (i.e. $1-\beta = .99$) instead of adjusting the effect size. As a result, in the revised manuscript we set $N = 942$ as the required sample size, which is much larger than in the previous version. Because we adopt a high statistical power (i.e. $1 - \beta = .99$) for the power analysis, we believe that the required sample size is sufficient to detect the effect of interest.

2-6. I would not recommend using arithmetic problems as an attention check, for several reasons: it is an entirely different domain to what you want to measure (reading/comprehension), it will bias the sample against people who struggle with numeracy, which is a bigger population than you might think, and it will probably serve to catch attention since the numbers may stand out amongst text. Also, it sounds like this method has been used a lot previously, and online survey-takers evolve to spot previous attention check question styles, so it is good to use different types of ACQs wherever possible. Ideally ACQs have the same format as the questions around them, but ask a question with only one right answer. So, for instance you could ask something like “how many times have you...” and then finish with something impossible, then exclude everyone who does not answer 0.

Reply: According to this proposal about ACQs, we have decided not to use arithmetic problems in ACQs in order to avoid catching attention and instead set a sentence type of ACQ that must be answered with 0. In particular, we ask participants in the 1st wave, ‘how many times have you had questions about his/her blood type so far?’ In the 2nd wave, we ask participants, ‘how many times have you been to the Mars?’ We will exclude participants who do not answer 0 for these questions. This information has been added to the *Data exclusion* subsection in the revised manuscript.

2-7. Crucial: The demographic information should be asked at the end of the survey – at least age and sex. This is important, as asking these at the beginning could affect the results by making e.g., participants’ own age more salient to them and thereby influencing their perceived vulnerability to disease or behavioral intentions, as it is well known that age (and to some extent sex) is linked to health outcomes with COVID-19.

Reply: According to these comments, the demographic information in the 1st wave will be presented just before the reminders. In the 2nd wave, the demographic information will be presented after all the surveys are completed.

2-8. Crucial: I have concerns about the validity of the proposed scale measures, Infection Prevention and Outing Frequency. I would advise only running this study after the authors have done a pilot for these measures and ironed out the potential problems. See the responses to this tweet for a good collection of resources for validating measurement scales: <https://twitter.com/NicoleBarbaro/status/1258514761429352448>

The items in the IP and OFI seem to have face validity but the authors have not provided any information on how they will check for internal reliability, nor sensitivity or any other form of measurement quality. A few specific issues jump out at me, but these are not the only issues: 1) Why is the rating scale from “not applicable” to “highly applicable”? I realize this may be an issue of translation and perhaps this makes sense for Japanese respondents, but it does not sound like a balanced scale, nor one suited for the items. Usually I would expect behavioral

intentions to be rated on a scale that measures agreement “strongly disagree” to “strongly agree,” for instance, or something equally balanced.

Reply: We deeply thank Reviewer 2 for these careful considerations. We will not conduct a pilot study. Let us explain the reason. Our research aims to explore teaching methods that encourage acceptance of administrative instruction, and we thus use it for the scale as-is. The administrative instructions to prevent the spread of infection have been developed by professionals of public health and are not published as validated psychometric scales. Thus, our scales are very well aligned with our research aims. Additionally, thanks to the Reviewers’ comments (of course, including Reviewer 2), the methods of our survey have been made much more sophisticated. In particular, as we stated in 2-10, we have also increased the statistical power according to Reviewer 2’s valuable comments. We are thus now able to iron out potential problems without a pilot study. Indeed, postponing the main survey should be avoided because the COVID-19 pandemic is converging in Japan. For these reasons, we will not conduct any pilot study of the psychological validity of those scales.

As Reviewer 2 pointed out, we should use a rating scale that is balanced and suited to the items. We have changed behavioural intentions so as to be rated on a scale that measures agreement *strongly disagree* to *strongly agree*. We thank Reviewer 2 for the valuable comment on the rating scale.

2-9. 2) How were the items for the Outing Frequency scale chosen? Many of them seem to overlap with the Infection Prevention scale, but not all. I can understand wanting to ask about both behavioral intentions and reported behavior, but I would expect Reviewer 2 to ask about the same behaviors in both scales unless you give a reason why not. There may be a good reason, but the authors need to share it.

Reply: As Reviewer 2 mentioned, the items partially overlap between the Outing Frequency and the Infection Prevention scales, which poses a potential problem. Therefore, we eliminated the Outing Frequency scale and made the IP-behaviour scale related to reported behaviours similar to the Infection Prevention scale (see Table 2 in the revised manuscript). Accordingly, we have changed the rating scale of reported behaviour to a 7-point Likert scale (from 1 = *not at all* to 7 = *very often*) for the items related to the reported behaviour.

2-10. 3) Item 3 in the IP is a triple-barreled item – I would advise on putting each in its own question, as people may have different answers for each situation.

Reply: As we replied in 2-13, the items in the COVID-19 IP scale were made in accordance with the guidelines given by the Japanese Ministry of Health, Labour and Welfare. This is also true for Item 3 (i.e. ‘I will avoid the “Three-Cs”’), which means to avoid the places satisfying any of these three conditions: (1) closed spaces with poor ventilation, (2) crowded places with many people nearby, and (3) close-contact settings such as close-range conversations. Therefore,

Item 3 in the IP is not a triple-barrelled item and we can conduct our surveys without difficulties or problems because ‘Three-Cs’ is a term widely understood by public relations activities of the Japanese government.

2-11. 4) Some of the IP items are vague – what does “often” mean for handwashing? I can’t immediately think of a way to improve that specific example, unfortunately.

Reply: The items regarding the frequency of infection-prevention behaviour ask for the frequency of actions related to the outing in the government’s guidelines. Therefore, ‘often’ in this item refers to the frequency of hand washing.

2-12. 5) I am concerned that the IP will have ceiling effects as people will want to present themselves in a socially desirable way. If a pilot study shows that there are ceiling effects, you may need to change the rating scale, re-word items to be less clearly socially desirable, and/or change the instructions to participants to emphasize honesty and anonymity.

Reply: We would like to express our appreciation of Reviewer 2’s careful consideration. As Reviewer 1 pointed out as well, we will emphasize in the instructions to participants that anonymity is ensured and the data obtained from the survey on personal information are not linked. In addition, the possibility of ceiling effects is not necessarily high because Japan has sustained its economic activities and has not imposed lockdowns.

2-13. 6) The IP scale contains both approach- and avoid- contexts – that is, both active and passive behaviors (e.g., active = hand-washing, passive = avoiding crowds). It’s not clear to me that these would be affected equally through the proposed mechanism. Furthermore, the OF scale contains only the passive/avoidance behaviors. I would think that if you consider the active behaviors such as hand-washing to be important, they should be on both scales—or at least better justify why this discrepancy.

Related to the above point: in Table 3, the first line in the “Question” column mentions inhibiting risky behaviors, but the related hypothesis concerns the IP which measures intentions for both inhibiting risky behaviors and activating/promoting risk-mitigating behaviors.

Reply: We would like to thank Reviewer 2 for these valuable comments. However, at this time, there is no evidence or reason urging us to separately address the active and passive behaviours in the present study. This means that we cannot develop a clear hypothesis about these issues. Thus, we do not plan to separately address active and passive behaviours in the present study. However, we can understand and are interested in Reviewer 2’s intuitive idea and some readers might have the same idea. For them, we will provide all the data at OSF and hope that they will perform exploratory examinations with our data. Moreover, we agree with Reviewer 2’s point about Table 3 and have modified this point in the revised manuscript.

2-14. **Whether the clarity and degree of methodological detail would be sufficient to replicate exactly the proposed experimental procedures and analysis pipeline**

(NB: Many of these apply to the next criterion (researcher degrees of freedom/flexibility) as well)

• How will you test/check that participants understand Japanese well? This does not necessarily have to be a complicated or time-consuming check, but since you mention it as a criterion please explain how you will ascertain that.

Reply: We thank Reviewer 2 for pointing that out. Since the recruitment information is written in Japanese, we consider participants who can understand them to have met the requirements. However, it is important to specify this inclusion qualification in the recruitment information. We will add the information that participants need to understand Japanese well to the recruitment requirements. In addition, we have added questions about participants' native language and nationality to the demographic information. We will exclude the participants whose answers for these questions are not Japanese and Japan, respectively.

2-15. • Define Tokyo exclusion criteria – will you advise potential participants on which areas “count” as Tokyo or is that already clear? E.g., will participants living in Tokyo suburbs be allowed to take part? Is there a clear way to demarcate (e.g., by listing the relevant postcodes or boundaries all participants will understand correctly?)

Reply: Yahoo! crowdsourcing can be set up so that survey requests will only appear for people who have registered their residence in Tokyo. We will use this setting to recruit participants who live in Tokyo and will also confirm their residence in the question form. We have added this point in the revised manuscript.

2-16. • On page 6 you state “It is important to set a limit on where participants live because this will allow us to ensure the validity of the dependent variable, the outing frequency index.” Please expand on this and explain how location can help you ensure the validity of the OFI.

Reply: The situation in Japan is gradually improving. However, Tokyo is still in a critical condition because it has a large population, and the spread of COVID-19 might thus become severe again without due care. Therefore, it should be in Tokyo that the reminders will work well at this time. Considering these differences in the situations among the cities in Japan, we will limit the participants to the persons who live in Tokyo.

2-17. • Crucial: Will assignment of conditions be randomized? If so, how? This is crucial. I would highly recommend randomizing because of course the inferential ability of the study could be drastically reduced if not.

Reply: We thank Reviewer 2 for this important point. We will conduct a randomized assignment to the three reminder conditions based on the participant's birthday: Participants whose birthday is the 1st–10th of the month will be assigned to the spreader condition, those with the 11th–20th to the spread condition, and those with the 21st–31st to the control condition.

2-18. •Crucial: Please give (a lot) more detail as to how the experimental manipulation (i.e., the behavior message/reminder) will be presented. You mentioned only, “After completing the first wave, we will present one of the infection-prevention reminders to the participants in the spreading and spreader conditions, whereas the participants in the control condition are given no reminder.” What will be the exact text of these messages, and how will the message be presented (e.g., audio or visual? If audio, what kind of voice? If visual, what size and color font, for how long?) How will you be sure participants have seen the message?

Reply: We will present the reminders as visual images with text. The colour of the text is white and the background colour is green. Because we will present images, the text size will depend on the execution environment, such as laptops, smartphones, or tablets, but the visibility of the reminders is well maintained in any environment. In order to make it easier for readers to image, we have added images of the reminders as actually presented as a figure (see Figure 2 in the revised manuscript) and provided further detail in the description of the experimental manipulation in our revised manuscript.

2-19. Crucial: Please report how you will aggregate the scale measures – will you take the mean of all items? Or something else?

Reply: Based on other reviewers' comments, we decided to perform principal component analysis for the COVID-19 IP scale about behavioural intentions and the COVID-19 IP scale about reported behaviour. We will then calculate the principal component scores and use the differences in the scores between the 1st wave and the 2nd wave for the statistical analyses (i.e. subtracting the scores of the 1st wave from those of 2nd wave). For more information on the main analysis, we discuss it in reply 3-2 to Reviewer 3. I would appreciate it if Reviewer 2 could check it. Moreover, we have added these points to the revised manuscript.

2-20. •Please report how you will validate the scale measures (at least the IP and OFI measures presented in tables 1 and 2)? E.g., will you check for internal reliability (for instance, by calculating Cronbach's alpha and excluding the lowest-performing item until alpha reaches a certain acceptable threshold?)

Reply: As mentioned above, our research aims to explore teaching methods encouraging acceptance of administrative instruction, and we use it for the scale as-is. Therefore, we will not conduct a pilot study of the psychological validity of those scales. However, this point is reasonable

because we use the total scale score for the analysis, and thus we will perform principal component analysis on the scales (for the details, see reply 3-2 to Reviewer 3).

2-21. **Crucial:** Please provide the introductory text / instructions to participants for how they should answer the scale measures IP and OF. For instance, what period of time are these questions asking about? This text matters to how participants answer.

Reply: We will ask the participants to respond with the IP questionnaires based on their behavioural intentions and actual behaviours for the week until the previous day of participation in each wave. To make it easier to understand, we have added a timeline of the survey schedule as a figure in the revised manuscript.

2-22. **Whether the authors provide a sufficiently clear and detailed description of the methods to prevent undisclosed flexibility in the experimental procedures or analysis pipeline**
Good specification of sample size and how sampling plan will deal with exclusion rules

Reply: We are very glad to receive Reviewer 2's positive comments.

2-23. **Whether the authors have considered sufficient outcome-neutral conditions (e.g. positive controls) for ensuring that the results obtained are able to test the stated hypotheses**

Crucial: In their "Quality Checks" section, the authors state "it is not possible to set positive controls for any pandemic effect." However, the main hypothesis and analysis in their study is not testing a pandemic effect, but rather a controlled trial (they do not state whether randomized) of an experimental manipulation on reported behavior and intentions. Therefore it is necessary to include a test to check that the experimental manipulation has "worked." They need to avoid the potential situation where critics could object to any null effects by arguing that maybe the participants didn't actually see/read the message (or otherwise, more broadly, that it was not presented in an appropriate or externally valid way).

Crucial: To answer the first of those objections, you need a manipulation check to be sure participants have seen (and ideally also absorbed) the message. I would suggest asking them to type it back into the survey system after they read (or hear?) it. You could present it on one page for a minimum of a certain length of time, (say 5 seconds) then when participants click to the next page, ask them to repeat the text. This will also serve not only to test attention, but to (hopefully) make the message more salient.

Reply: We would like to thank Reviewer 2 for pointing out this instruction. Although presenting the reminder for a short duration and asking the participants to type it might be a good idea, it is likely that many participants will miss the reminder. Moreover, it is possible to set such a task in the control condition; in this case, the number of the tasks becomes uneven among the conditions. Alternatively, set up instructional manipulation checks (IMC) to exclude the participants who did not actually see/read the gratitude messages including the reminder

message in the spreader and spreading conditions. The IMC will be placed on the same page as the gratitude messages and will read, 'Do you like the font used in the above message? Be sure to answer N/A'. Non-N/A respondents will be excluded as not having read the message.

2-24. Crucial: There is another issue related to positive controls, in the "Interpretation given different outcomes" column of Table 3. Here the authors raise the possibility that a null outcome would either signal a true null effect, or (if I understand them correctly), a ceiling effect where perceived vulnerability to disease is so high that it cannot be increased. I am confused about this interpretation because 1) the outcome measure is IP and OF, not PVD (and the authors have not made it clear how they expect PVD to be related to the main outcome measures, nor have they indicated there is any evidence for it)—so why would they expect null results to be related to ceiling effects in PVD?; 2) if they indeed expect a ceiling effect of any of the measures, then why haven't they included any plan to mitigate this possibility?

Reply: As mentioned above, we have decided not to use the PVD score as a covariate. Thanks to Reviewer 2's valuable comments, our hypotheses and predictions have been clarified. Moreover, as we replied in No. 2-12, we consider the possibility of a ceiling effect as not necessarily high because Japan has sustained its economic activities and has not conducted lockdown. In addition, we will emphasize in the instructions to participants that their anonymity is ensured and the data obtained from the survey of personal information are not linked.

2-25. A plan to detect and mitigate potential ceiling effects in the outcome measures will be necessary to making sure the results are interpretable, in either case. As I suggested above, a pilot test (a single cross-sectional survey) of the outcome measures will show whether the IP and OF have ceiling effects. It does not matter a lot if the PVD has ceiling effects, as this was only intended as a covariate in any case; ceiling effects would just make it a not very useful covariate but (I think) shouldn't compromise the integrity of the experiment.

Reply: Thanks to these comments, the methods of our survey have been drastically improved. In particular, following Reviewer 2's helpful comments, we will emphasize in the instructions to participants that anonymity is ensured and the data obtained from the survey personal information are not linked. Moreover, it is less likely that a ceiling effect will occur. Indeed, postponing the main survey should be avoided because the COVID-19 pandemic is converging in Japan. For these reasons, we will not conduct any pilot study of the psychological validity of those scales. Moreover, we will not use the PVD as a covariate, as we mentioned in reply 2-3.

2-26. Lastly, as I mentioned in the section on Hypotheses, above, the analysis plan should include a test between the spreading and control conditions (for each of the outcome measures). This

will help determine whether any kind of message, presented in the way the study does, can have an effect on health behaviors and intentions, and in turn this can help interpret the results if no evidence is found to support the main hypotheses.

Reply: As we replied in 2-2, based on the findings in previous studies (e.g., Bryan et al., 2013), we predict that the change from the 1st wave to the 2nd wave will be higher in the spreader condition than the spreading and control conditions and there will be no difference between the spreading and control conditions.

2-27. Minor points: (Note that when I refer to page numbers it is the numbering from the original document next to the running head, not the numbers imposed by the PDF creation.)

•Page 3, line 53: you appear to be claiming your findings already show something, but you have not run the study yet. Please rephrase – for instance, you could state that your study “will aim to show whether it may be possible...”

Reply: We deeply thank Reviewer 2 for pointing out the wording. Regarding the description of expected findings, we have replaced the affirmative expression with a weaker expression, as you suggested.

2-28. •At the top of page 6, you mention an exclusion “criterion detailed in (7).” I didn’t know where to find this – could you list the section it is in, or make it clear whether you are referring to page number, etc.

Reply: This was a careless mistake. We have amended our manuscript as follows: *criterion detailed in Data exclusion criteria.*

2-29. Please move the sentences explaining the details about the PVD from the Hypotheses section to Materials and Procedure (or create a Measures section and explain all measures there).

Reply: We have decided to exclude this hypothesis about PVD from the Hypotheses section. In addition, based on your suggestion, we have mentioned the detailed explanation of PVD at Materials and Procedure section in our revised manuscript. We would sincerely thank you for checking our research plan in detail.

2-30. In the IP scale, item 4, I am not sure what “I will ventilate regularly” means. I understand this is translated from Japanese, so I assume it will make more sense to participants!

Reply: In this study, we constructed the COVID-19 IP scale based on the guidelines provided by the Japanese Ministry of Health, Labour and Welfare as-is. It seems that the meaning of the sentence is hard to understand when it is translated into English. This question asks about the frequency of the act of supplying fresh air and eliminating of foul air (e.g., opening the windows and let some fresh air in the room). In Japan, ventilation is a familiar and common

practice in normal times as well in order to combat infectious diseases. Therefore, participants can understand the meaning of this question.

References

- Bryan, C. J., Adams, G. S., & Monin, B. (2013). When cheating would make you a cheater: Implicating the self prevents unethical behavior. *Journal of Experimental Psychology: General*, *142*, 1001–1005.
- Lakens, D. (2014). Performing high-powered studies efficiently with sequential analyses: Sequential analyses. *European Journal of Social Psychology*, *44*, 701–710.
- Perezgonzalez, J. D. (2017). Statistical Sensitiveness for the Behavioral Sciences. *PsyArXiv*, <https://doi.org/10.31234/osf.io/qd3gu>

Responses to Reviewer #3

- 3-1. This is a useful extension of a well-known effect in social ethics communication that has also been applied successfully in environmental attitudes work. To my knowledge, it has not been applied explicitly to infectious disease communication, and thus it has a good amount of applied potential.

From a basic science perspective, the main theoretical merit is knowing whether these effects from ethics and environmental research generalize to infectious disease communication. On this note, the contribution doesn't offer much in the way of explanation of why it wouldn't generalize to infectious disease, or measures that will help to explain why it didn't generalize if it doesn't. Thus, a primary weakness of the present report is that it doesn't really offer a "back-up" plan if they don't find the results they expect. All we will know is that, in this case, the previous results don't generalize. I don't have a great solution to this, but to the extent the authors could think of ways to anticipate what it might mean theoretically about infectious disease communication if their predictions don't hold, it could be helpful and might steer them toward additional variables to measure. For example, there might be differences in perceived agency or inevitability when it comes to infectious diseases that might reduce the effect of such a manipulation. Ethics and environmental stewardship are much more agent oriented domains, whereas infectious disease tends to lack the same level of controllability. Perceived control can have a large impact on behavioral intentions and attitude-behavior alignment.

- Reply:** Even though you are probably in a difficult situation, we cordially thank you for the extremely quick review, positive comments, and valuable suggestions. As you point out, if the hypothesis is supported and more effective teaching methods to prevent the spread of infection are identified, they could be generalized to infectious disease communication. On the other hand, if the hypothesis is not supported, this would suggest that self-relevance reminders do not have

remarkable effects in a serious context like the COVID-19 pandemic. Therefore, our research will not have no-contribution results even if we fail to find our expected results. This is just our humble opinion, but we think registered reports and pre-registration study do not have theoretical significance, but serve to increase the strictness and transparency of the respective experiment. If we add several variables to seek theoretical significance, the strictness and transparency will inevitably be reduced. In fact, a previous study reports that almost all registered report papers deviate from their protocols (Claesen et al., 2019). Therefore, we believe that the simpler, the better in the case of registered report protocols. Accordingly, we will not add any variables. Fortunately, because we can make comparisons of the PVD scores between the current and non-pandemic situations, we will discuss the effect of self-related reminders based on these comparisons.

3-2. No psychometric criteria are described for averaging over the questions in Table 1 or Table 2. I assume this is the intention as the statistical methods describes using difference scores for these tables as a whole as dependent variables. It would be useful to look at whether they do measure consistent psychological constructs by looking at item analyses and alphas. If not, whether the items will be looked at individually (instead of means) should be discussed and whether there are plans to correct for multiple comparisons if they move to an item-specific strategy. For example, what if people treat being in a low ventilation area as fundamentally different from going out to buy necessities? Or wearing ventilators as different from not speaking loudly? It seems these scales have the possibility of measuring multiple different constructs.

Reply: Thank you for your important comments and suggestions on the analysis. First, for the questions in Table 1, we will perform principal component analysis to aggregate data along a single dimension. As a criterion, we exclude items with a loading of less than 0.35 for the first principal component. We then repeat this process until the first principal component loading for all items is greater than or equal to 0.35. Next, we calculate the principal component score for the first principal component. When testing the hypothesis, we will use the principal component score as the dependent variable instead of the mean. The same process will be performed for Table 2.

3-3. I don't understand why the authors need to set an upper bound on sample size. If they go over planned sample size, they could theoretically keep to the original planned t criteria (if they are worried about effect sizes that are too low being accepted as significant), for example, but the added precision would always seemingly be worthwhile.

Reply: As you pointed out, we should not set an upper limit on the sample size and exclude excess data from the point of view of statistical power and participants' cost. A similar point was made by Reviewer 1 too. We thus do not set an upper limit on data collection.

References

Claesen, A., Gomes, S. L. B. T., Tuerlinckx, F., & Vanpaemel, W. (2019). Preregistration: Comparing Dream to Reality. *PsyArXiv*. <https://doi.org/10.31234/osf.io/d8wex>

Responses to Reviewer #4

4-1. The authors set out to determine whether different messages (the spreading, the spreader vs. control) would prevent high-risk behaviors during the pandemic. They use a mixed design where the dependent variable will be measured in a within-subjects manner, and the manipulation will be between-subjects (if I understood correctly).

I like the proposal for its simplicity and applicability, and I think it will be a timely contribution. But the authors should clarify some points before I recommend it for stage 1 acceptance.

If I understood correctly, the authors will first measure preventive behavioral intentions, and then in Wave 2, they will use the same participants and before measuring their preventive behavioral intentions again, the participants will be randomly assigned to three different conditions (the spreading, the spreader, or the control). They also measure PVS as a covariate and will look at the difference between a two-time frame due to the saliency of the pandemic. However, in P6, L47-59, there is no reference to experimental manipulations. The authors even explicitly state that the participants will be presented with “no reminder in all conditions” in Wave 2. So I think I’m misunderstanding the design. I think a separate design section should help the authors clarify their points.

Reply: Even though you are probably in a difficult situation, we thank you so much for the extremely quick review. All of your comments are very important, and we will deal with them in good faith. Frankly speaking, you have misunderstood our design because our description may be ambiguous. In the 1st wave, we plan to assign the participants into three groups based on the kinds of reminders (‘Don’t be a spreader’, ‘Don’t spread’, and no reminder). We will then examine the behavioural changes in each group for a week after they read the reminders. Because our description involved misleading expressions, we have amended and added the timeline schedule of the survey so that it is easy to understand.

4-2. There are also some inconsistencies. For example, in P6, L22, the authors are saying that they will take the participants’ email only in the first Wave but in the following paragraph, they report that they will measure several other variables including COVID-19 IP. This inconsistency should be solved. Relatedly, the hypotheses should be clearer.

If I were you, in Wave 1, I would measure the dependent variable and the covariates together with the demographics without experimental manipulation (i.e., baseline). And in Wave 2, the

same participants would be randomly assigned to three conditions and then I would measure the same DV and covariate again. And to analyze this, I would conduct 3 (manipulation: spreading, spreader, control) by 2 (time: pretest, posttest) mixed ANOVA (where the latter factor was within-subjects) and look at whether there is any significant interaction. If this is the design the authors proposed, it should be better clarified. If not, I would like to quickly review a revised version of the manuscript.

Reply: Our description of methods is not immediately obvious and as a result it seems to be confusing. We thus here provide further explanation of the procedure. We will use a popular Japanese crowdsourcing site (Yahoo! Crowdsourcing) to encourage widespread participation in the 1st wave survey. At the end of the 1st wave survey, we will ask and collect participants' email addresses to send them a web link to the 2nd wave survey because it is not possible to use crowdsourcing to invite the same participants to participate in the 2nd wave survey due to anonymity. Next, participants will complete the COVID-19 IP scale about behavioural intentions, the COVID-19 IP scale about reported behaviour, the PVD scale, and a questionnaire about demographic information. At the end of the survey, one of the three reminders will be presented ('Don't spread', or 'Don't be a spreader', or no reminder). The purpose of this study is to investigate whether the differences in the reminders influence participants' behaviour in the week before the 2nd-wave survey. Therefore, the reminders will be presented only at the end of wave 1, and they will not be presented during the 2nd wave. In the 2nd wave, we only ask participants to answer demographic information and complete the COVID-19 IP scale about behavioural intentions, COVID-19 IP scale about reported behaviour, and PVD scale. We repeatedly note that participants will not answer from their email address in the 2nd wave. In addition, although you did not seem to find it particularly problematic, the analysis with PVD as a covariate was dropped based on the comments of other reviewers. We think the factor design has become much simpler.

4-3. A minor point is when COVID-19 IP abbreviation first appeared in the hypotheses section, the unabridged version should be reported in parenthesis.

Reply: We amended the COVID-19 IP abbreviation where it first appeared in the hypotheses section to provide the full term in parenthesis.

Responses to Reviewer #5

5-1. In the submitted proposal, the authors suggest that verbally emphasizing self-identity might play a role in motivating individuals to comply with the government's guidelines regarding the containment of COVID-19 cases. To investigate their assumption, they propose to implement an intervention and randomly assign individuals into one out of three groups: In

one group, they are presented with the reminder "don't be a spreader" (self-identity related), in the second group, they are presented with the reminder "don't spread" (self-identity unrelated) and the third group serves as a control group without any reminder. Effectiveness of this intervention is planned to be examined by assessing individuals' agreement with the specific guidelines provided by the Japanese Ministry of Health, Labor and Welfare (COVID-19 IP scale), which is assessed at two timepoints (before and after the intervention). In general, I think the proposed research study could make a valuable contribution with regard to the questions how individuals can be motivated to comply with guidelines and recommendations by political and scientific institutions to contain the spread of COVID-19. Even though the expected effect-sizes might be small: Regarding the current global health crisis every measure that even in the slightest can contribute to lowering infection rates and thus prevent deaths, is worth being taken seriously. However, without doubting the potential value this research might have, I think that the submitted manuscript would benefit from a comprehensive revision and that the authors might also want to reconsider the statistical approach they suggest. In the following I will highlight my concerns in more detail:

Reply: Even though you are probably in a difficult situation due to the COVID-19 pandemic, we sincerely appreciate your quick review and suggestive comments on our manuscript. We have replied to your comments as follows.

5-2. **Majors:**

Design/Statistical Approach: As far as I understand it, the proposed design entails a within-subjects factor: The main dependent variable (COVID-19 IP Scale) is measured in the first wave as well as in the second wave one week later. However, the authors address their design as a between-subject design. They propose to calculate a COVID-19 IP index by subtracting the IP Scale Scores of the first wave from those of the second wave and to perform an ANOVA/ANCOVA with the COVID-19 IP index as a dependent variable. In my opinion, this approach has major drawbacks and I would strongly recommend to apply analyses that are more appropriate for within-subject designs, such as a one-way repeated measures ANOVA or mixed models. Between-subject designs have substantial power deficits compared to within-subjects designs, which is why this aspect should already be taken into account for power/sample size considerations. In my opinion, this is actually one of the major benefits of choosing a repeated-measures design, and I'm irritated that the authors would choose this approach and nevertheless analyze their data with a simple one-way ANOVA.

Reply: Thank you for your important suggestion for statistical analysis. As we responded to other reviewers' similar comments, we have decided to use one-way ANOVAs instead of ANCOVAs. In addition, our research interest is whether the difference in infection-prevention reminders between participants can influence Japanese citizens to modify their own

behaviours, not intra-participant change of behaviour from 1st wave to 2nd wave. Therefore, we have decided not to analyse within-subject factor.

5-3. **Theoretical Background:** The introduction part is extremely short and, in my opinion, lacks a proper theoretical foundation. The authors introduce the concept of "self-identity" as a basis for their intervention, but don't explain what this concept actually entails, nor do they refer to relevant literature in this field. They provide one reference from a study that applied a similar intervention in a different context, but don't elaborate on the psychological underpinnings that drive this process. The basic assumption that people strive for a positive self-image would be at least worth mentioning in my view. To sum it up, in my opinion the current content of the introduction makes no strong claim for the hypotheses; it's difficult to evaluate their plausibility if there barely is a theoretical foundation to refer to.

Reply: Thank you for your important suggestions. As you pointed out, the description of introduction was insufficient to support our hypothesis. Therefore, we have added a detailed explanation of self-identity and its relevant literature. We also explained the psychological background of the promotion of behaviours to inhibit the spread of COVID-19 by verbal expressions.

5-4. **Minors:**

ANCOVA: What is the authors' intention in including the "perceived vulnerability to disease" (PVD) as a covariate? How do they expect this variable to influence their dependent variable? I suspect it's about reducing error variance, since the PVD most likely will influence individuals' behavior and their compliance with guidelines. But why is it only included as a covariate if the PVD scores have changed during the pandemic compared to before? I would appreciate if the authors would elaborate in more detail why they choose this approach.

Reply: We are grateful for your comments about the analysis. We have decided that the analysis with PVD as a covariate was dropped based on the comments of other reviewers.

5-5. **Confidentiality:** How is participants' data linked? To me it reads like email addresses are used to link the data, which would not be an appropriate approach since it clearly contains identifiable information. Please state if an anonymous code is used to link the data.

Reply: We appreciate your consideration of this point. As we referred to in our replies to other reviewers' comments, participants' email addresses collected in the 1st wave will only be used to send the Web link of the 2nd wave survey to participants. In addition, we will conduct our survey anonymously and COVID-19-related data and personal information such as e-mail address will not be linked. We have added this information to the Methods section.

5-6. Scales: On what scale is the Outing frequency scale measured? Plus: Why does the number of scale point vary between the different scales? In case there's no strong reason for it, I always find it unnecessarily confusing for participants.

Reply: Thank you for reviewing all the details of our study design. In accordance with your comment, we have added an explanation of what the COVID-19 IP scale about reported behaviour is. Moreover, following other reviewers' comments, we eliminated the Outing Frequency scale and constructed the IP-behaviour scale, which measures behaviours reported on the Infection Prevention scale (see Table 2 in the revised manuscript). Accordingly, we have changed the rating scale of reported behaviour to a 7-point Likert scale (from 1 = *not at all* to 7 = *very often*) for the items related to the reported behaviour. We have stated this information in the Measures part.

5-7. Hypotheses: "H0: If the COVID-19 pandemic does not affect perceived vulnerability to disease, there will be no significant differences in the PVD scales scores during and before the COVID-19 pandemic" There's an assumption in this hypothesis that does not belong in there, since it's correlational data. It still might be that there are significant differences in the PVD that is not related to COVID-19. I would recommend to stick with the wording from H1.

Reply: We have removed the hypothesis of an association between PVD and the COVID-19 from the key hypotheses in light of other reviewers' comments. In addition, we used the PVD data only to confirm whether the PVD scale scores during the COVID-19 pandemic is different from that before the pandemic. Accordingly, as you pointed out, we approve of your opinion that the association between PVD and the COVID-19 is correlational data, not causal. However, it would allow us to state at least that people's attitude to an infectious disease is different during and before the COVID-19 pandemic if there are significant differences in the PVD scale scores that might be not related to COVID-19.

5-8. Hypotheses 1 refers to the COVID-19 IP Index, but at this point this measure has not been introduced yet.

Reply: Thank you for your advice. We now state that the COVID-19 IP scale is based on the COVID-19 infection prevention policy of Japanese Ministry of Health, Labour and Welfare in the Hypotheses section.

5-9. **Summary**

To address the following points required by the journal and recapitulate the above-mentioned aspects:

1. *The scientific validity of the research question(s)*

Generally valid and might make a valuable contribution in the current situation; but please note my remark regarding lack of theoretical foundation, which makes it difficult to evaluate this aspect.

2. The logic, rationale, and plausibility of the proposed hypotheses

They basically are derived from the outcome of one similar prior study in a different context. The authors don't elaborate on other existing research and literature, and thus it's difficult to evaluate the hypotheses' plausibility since in my view the manuscript lacks a proper theoretical foundation.

Reply: We are grateful for these positive comments and careful considerations. As stated in our response to comment 5-3, we have described the theoretical background of our hypothesis, citing some relevant documents.

5-10. *3. The soundness and feasibility of the methodology and analysis pipeline*

Please see my statement on design issues above.

Reply: We have responded to this comment in replies 5-2 and 5-4.

5-11. *4. Whether the clarity and degree of methodological detail would be sufficient to replicate exactly the proposed experimental procedures and analysis pipeline.*

The authors do provide their materials and replication should generally be feasible. However, based on the methods section, the exact procedure of the study seems a bit hard to reproduce, hence e.g. a Flow Chart might simplify that.

5. Whether the authors provide a sufficiently clear and detailed description of the methods to prevent undisclosed flexibility in the experimental procedures or analysis pipeline

In my opinion the methods section would profit from depicting the procedure in a more chronological order and in more detail.

Reply: We thank Reviewer 2 for his/her sincere comments. According to his/her suggestion, we have added a diagram of the chronology of the survey and detailed descriptions of the experiment and its ethical considerations to the Method.

5-12. *6. Whether the authors have considered sufficient outcome-neutral conditions (e.g. positive controls) for ensuring that the results obtained are able to test the stated hypotheses*

In my opinion the authors meet this criterion since they include three different groups plus pre- and post-measures

Reply: We would like to deeply thank Reviewer 5 for his/her positive comment.

Again, we would like to express our sincere gratitude to the reviewers for all of their prompt, thoughtful,

constructive comments. We hope that our revised manuscript is now suitable for Stage 2 Registered Reports in *Royal Society Open Science*.

Sincerely,

Fumiya Yonemitsu

Kyushu University

744 Motoooka, Nishi-ku, Fukuoka 819-0395, Japan

Phone/Fax number: ++81-92-642-2418

E-mails: y.fumiya.0408@gmail.com

Appendix D

The authors resubmitted their manuscript intitled: Warning “Don’t spread” vs. “Don’t be a spreader” to prevent the COVID-19 pandemic, addressing reviewers’ comments. I think that this improved the overall quality of the manuscript. Hopefully, the authors agree. Among other, the authors updated their literature review, removed the PVS from their main analysis, and are way more specific about how the study will be conducted. My main concerns were related to the strength of the rationale, detailed related to the sampling plans, and the appropriateness of the analysis. In my opinion, the authors successfully addressed some of these comments (namely, the ones related to the sampling plans and the appropriateness of the analysis). However, I still think the authors should discuss (even if briefly) the mechanism involved in their intervention.

It would be kind of unfair from my position to simply ask to discuss the mechanisms without explaining why. Obviously, the present manuscript could have a very clear consequence from an applied perspective: if the expected results are found, the theoretical contribution could be minor, but the paper could still have a lot of value. However, I think that underlying the mechanisms that the authors think explains their effect could be of practical importance: Walton (2014) made the case for so-called wise intervention. By identifying the mechanisms involved, psychologists can design intervention targeting recursive processes, hence causing a lasting change. By doing so, we also have better chance to understand how an intervention could be context dependent. Right now, I have to say that I am not entirely clear on what would make this intervention work (the fact that it worked before is not a strong argument in my opinion). I do understand the time-sensitive nature of the situation, so I’ll let the editor weight what would be desirable for this manuscript. However, I strongly encourage authors discussing mechanisms involved in their intervention: Why making an identity salient would make participants adopt more desired behavior?

Major

- Authors should discuss what is the process through which identity salience should impact adoption of behavior related to the spreading of Covid-19.
- Authors should be careful when citing research that haven’t been peer-reviewed “recent research has found that manipulating the expressions of messages to appeal to citizens’ emotions is effective in promoting behaviour to self-isolate”. Right now, it is not entirely clear that this research is a preprint.
- The authors should be explicit about their intention to make the (anonymized) data set of this experiment publicly available.
- In the Quality checks for providing a fair test section, the authors say that “If there is no significant variation, we will interpret this [t -test] as indicating that the perceived concern for infection is essentially the same as before COVID-19 even under the influence of the COVID-19 pandemic.”. If the authors want to make such conclusion, they should conduct the appropriate tests (see Lakens et al., 2018)

Minor

- Right now, it is not crystal clear whether Klein et al. (2019) failed to replicate mortality salience effects (see Chatard et al.'s analysis, 2020; doi: 10.31234/osf.io/ejubn). However, if significant this effect seems to be way smaller than originally thought.
- Authors should report their intention to conduct **two** one-way ANOVAs in their participants section. Moreover, they should be explicit that the DVs will be the change in IP-intention and IP-behavior.
- Authors report wanted to test whether the reminder condition differ. They report their intention to conduct Scheffé's *F* test to do so. I still don't understand why the authors want to adopt a conservative test whereas they could have used a priori contrast (see below). However, given the sample size, it should not make a big difference.

Table 1. A priori contrast to test authors hypotheses. C1 tests whether the reminder condition differ, C2 if they outperform the control condition.

	Don't spread	Don't be a spreader	Control
C1			
Do the reminder condition differ?	1	-1	0
C2			
Are the reminder condition better than no reminder?	-1	-1	2

References

- Chatard, A., Hirschberger, G., & Pyszczynski, T. (2020). *A Word of Caution about Many Labs 4 : If You Fail to Follow Your Preregistered Plan, You May Fail to Find a Real Effect* [Preprint]. PsyArXiv. <https://doi.org/10.31234/osf.io/ejubn>
- Walton, G. M. (2014). The New Science of Wise Psychological Interventions. *Current Directions in Psychological Science*, 23(1), 73-82. <https://doi.org/10.1177/0963721413512856>
- Klein, R. A., Cook, C. L., Ebersole, C. R., Vitiello, C. A., Nosek, B. A., Chartier, C. R., Christopherson, C. D., Clay, S., Collisson, B., Crawford, J., Cromar, R., Dudley, D., Gardiner, G., Gosnell, C., Grahe, J. E., Hall, C., Joy-Gaba, J. A., Legg, A. M., Levitan, C., ... Ratliff, K. A. (2019). *Many Labs 4 : Failure to Replicate Mortality Salience Effect With and Without Original Author Involvement* [Preprint]. PsyArXiv. <https://doi.org/10.31234/osf.io/vef2c>

Lakens, D., McLatchie, N., Isager, P. M., Scheel, A. M., & Dienes, Z. (2018). Improving Inferences about Null Effects with Bayes Factors and Equivalence Tests. *The Journals of Gerontology: Series B*. <https://doi.org/10.1093/geronb/gby065>

Appendix E

Review (Stage 1, Round 2) for Registered Report entitled: Warning “Don’t spread” vs. “Don’t be a spreader” to prevent the COVID-19 pandemic (RSOS-200793)

NOTE TO EDITOR: I don’t know that much about Principal Components Analysis, so it is crucial that you get feedback from someone who can tell you whether it is appropriate to use principal component scores as a dependent measure that is then compared in an ANOVA.

Also, I would only accept the paper if the authors are willing to provide stronger positive controls tests – I am not convinced by their manipulation check, nor by their confidence that there will be no ceiling effects. If they get null results, I would not consider them informative unless they have a strong manipulation check and can show the results are not due to ceiling effects.

Firstly, I am pleased and impressed to see how much work the authors have put into improving the Stage 1 manuscript (study protocol/preregistration) in such a short amount of time.

...

Below I note the changes I think they still need to make before the manuscript should be accepted:

Hypotheses:

- The 2 hypotheses are now much clearer that they are measuring the change from wave 1 to wave 2.
- **However**, I still believe the hypotheses are not quite what the authors intend, based on my reading of the rest of the manuscript. It seems to me the authors do not care whether the spreading condition will differ from control (although they expect based on past results that it will not differ— either way, it would not affect their conclusions about the spreader condition, i.e., the condition they are interested in). Despite this, the authors have added into H1 and H2 the prediction that change scores will not differ between the spreading and control conditions. I don’t think that is a main element of what they are interested in, and makes the hypotheses too complex. **I think instead they should remove the last sentence of hypotheses 1 and 2, and just change the null hypothesis to state that the spreader condition will not differ from either the spreading or control condition.** It could be interesting to test whether the two conditions of non-interest (control and spreading) differ, but I would either put that as a separate, secondary hypothesis, or just include as an exploratory analysis.
- The “side hypothesis” about PVD should be listed in the hypotheses section and clearly indicated it is a secondary / less important / not main hypothesis (instead of only mentioning it in the Measures section). I apologize that my previous note on this from Round 1 (point 2-29 in the authors’ response) was unclear – I meant the authors should include just the hypothesis itself in the Hypotheses section, and then detail all the information about the measures themselves in a separate section within the methods, which they have now done.

Measures

- I am pleased to see there is now more clarity about what each scale is attempting to measure, and the reasons for using these unvalidated scales (due to their tie to the government guidelines). However, a few remaining concerns below:
- I would highly recommend rephrasing the behavior scale items to ask about behaviors themselves rather than avoidance of behaviors. If you ask about “how often did you avoid x behavior” it is difficult for the participant to answer – it is much harder to think of what you have NOT done, versus what you did. So I would instead ask, e.g., “how often did you go out” and then reverse-code these items.
- Are you sure you want to have a Likert response scale of “never” to “often” for the reported behavior measure? I actually thought that asking participants to count the number of times they did an action was more objective. It is hard to know what going out “often” means. However I can see arguments for and against either option. I can foresee that reporting raw numbers could increase the variability in responses and give problems with non-normality (right skew), so that is one argument against it (although these issues can be attenuated by testing for skew and applying a transformation if there is skew). Another possible argument against participants entering raw numbers is if you want to be able to directly compare the behavior intention and reported behavior scales—but even then, the response scales are qualitatively different (agreement versus reporting frequency) even though they have the same number of scale points. I would think carefully about why you want to use a Likert scale versus reporting actual numbers of behaviors. If you care more about whether the manipulation will change total numbers of behaviors across a population, I would go for reporting raw numbers. If you care more about how many individuals the manipulation will convince to change their behavior, go for the Likert scale.
- Some items on the reported behavior scale will need additional instructions, or include a “not-applicable” option, particularly number 2 and number 7. For instance, “how often did you go out wearing a mask” will depend on how often the participant went out, so it may be misleading (for instance a participant who did not go out at all would answer “never,” and therefore reduce their score – even though they followed guidelines by not going out!) Similarly, if participants did not cough or sneeze, they will get a low score on item 7, even though they are following guidelines. I would suggest giving participants a “not applicable” option for those 2 items, but if you cannot do that, you need to acknowledge that this might introduce noise to your measure and think of ways to address that (for instance, you might plan to re-run the analysis both with and without these 2 items).
- Related to the above note – item 2 is ambiguous and should be reworded to something like, “when you went out, how often did you wear a mask?” You care about the percentage of time people wore a mask in public, not the total amount of time (otherwise you would not know about how much of the time they went out in public NOT wearing a mask).
- Despite the authors’ arguments why they don’t expect a ceiling effect, I think they still have to be prepared for the very real possibility of a ceiling or floor effect on both scales. Please consider how you will test for a ceiling/floor effect, and what you will do if you find one.

- **CRUCIAL:** I unfortunately do not have enough knowledge about principle component scores to know whether they are appropriate to use as a dependent variable, but I have doubts. These should be checked by an expert. I gather using the principal component scores is an attempt to weight the items according to how well they cohere with the overall (forced single-factor) scale? I am not sure about that. – **I would report an analysis that uses mean scores on each scale, regardless of whether you also use a principal components score.** It is easier to understand and does not make “invisible” assumptions about how to weight the various items on the scale. Also, I am not sure whether it is even appropriate to calculate a linear change score between 2 different principal component scores. That doesn’t make sense to me how this will capture change in participants’ behavior – instead won’t it capture change in the relative weighting of different items on the scale, because the factor loadings will be idiosyncratic to each PCA? If everyone’s 2nd-Wave scores remained exactly the same as 1st-Wave except that they all went up by 1 Likert scale point, then wouldn’t the principal component scores remain the same both times, even though behavior changed considerably? (But I may be misunderstanding principal component scores, I am not an expert!)
- Related to the above I think it’s good to do a PCA to see how well the items on your scales cohere into a single measure, but since you have an a priori reason to use those items (they are the behavior guidelines the government wants to influence) I’m not sure it’s appropriate to drop any of them simply because they don’t fit in a single-factor model. You stated before that you don’t care whether these items all are affected using the same mechanism (i.e., you don’t have hypotheses about approach versus avoidance behaviors) and I think the same principle applies here— you include the items because of the practical consideration of wanting to measure those specific behaviors, so you should include them all somehow. That could be forcing them to be all in the same scale regardless of fit, or separating them into separate factors based on the PCA.

Attention Check questions

- I like the new attention check questions, and it is good that they are text-based. However, since the participants are now answering everything on a Likert scale, they will still stand out if participants have to enter text! I would give them a Likert scale answer (unless you change the reported behavior scale back to number-entry).
- I am not convinced the instructional manipulation check will be effective. It is merely another attention check, and does not in any way guarantee the participant read the message. You must include a test that can tell whether or not the participant understood the message – I would highly recommend asking them to type it back on the next page (or you could ask them to choose it from a list of options—in this case I would not include both spreader and spreading message in the options, I would include other options that don’t mention “spread” in any way, such as “be safe” or “don’t be reckless”). Are you concerned with external validity, because in real life people are not asked to type back messages, or with demand characteristics (that participants might figure out the study is about the message)? I would not be concerned with these. I would be more concerned that the message gets lost and participants do not see it.

Small points:

- The section on ACQs still says “We will insert a simple math problem as an attention check question” (page 37 line 32)
- In the reply to point 2-21, the authors state that they will ask participants to answer both scales based on the week before participation in each wave. Shouldn't the behavior intention scale be based on the upcoming week (the week after each wave) instead? Behavioral intention is usually measured regarding the future. Have I misunderstood? Figure 1 is generally helpful but it does not differentiate between the two scale measures.

Appendix F

Warning "Don't spread" vs. "Don't be a spreader" to prevent the COVID-19 pandemic

Review Stage 1 Registered Report – Revision

I have read a previous version of the manuscript and I appreciate the work the authors put into the revision of the paper and their effort to comply with the reviewers' and editor's requests. I still believe that the addressed research question could make a valuable contribution to the current situation and literature and is worth being investigated and published. However, I still noted some aspects in the revision of the manuscript which in my opinion need editing, which keeps me from recommending acceptance of the manuscript at this point. In the following, I will outline these critical aspects in more detail:

Introduction:

I appreciate that the authors put effort in revising the introduction part, they did add important information on the findings in similar studies. However, I still miss an integration of the research question into the general literature framework. The authors are listing similar research findings that support the assumption that their intervention might work, but they still don't refer to the psychological background and meaning of social identity. As far as I understand it from the journal's guidelines, after acceptance of the manuscript at stage 1 the introduction cannot be altered in stage 2. If, besides being able to provide recommendations for policy makers regarding the current situation, the submitted manuscript also is supposed to make a sustainable contribution to the field's literature (in the sense of a conceptual replication that might support the validity of other findings on social identity), I would appreciate a more exhaustive elaboration on the psychological construct of social identity.

I further do not understand the reference to Mortality Saliency in the introduction part, since to me it isn't clear how this relates to the topic of social identity:

On the other hand, another intervention is to evoke mortality in messages based on terror management theory (Greenberg et al., 1994).

Method:

The randomized assignment for the three reminders will be based on the participant's birthday: Participants whose birthday is the 1st–10th of the month will be assigned to the spreader condition, those with the 11th–20th to the spread condition, and those with the 21st–31st to the control condition.

Strictly speaking, this is not a randomized assignment since there is clear pattern in the way participants are assigned to the conditions. What if there actually is an external cause for participants born at the beginning of the month to behave differently than participants born at the end of the month? This might seriously threaten the validity of the authors' findings and I would strongly recommend to use a more valid approach of randomization (e.g. a random number generator, should be implemented in most survey software)

Data Analysis:

I understand that you aim at doing a PCA to eliminate items that are not consistent with the construct you want to measure. Therefore, you suggest to eliminate items with a low loading on the first principal component. As PCA is an explorative approach, there might come up more than just one relevant component that explain a substantial part of the variance in your data, which however you would just drop (is that necessary/useful?).

Further you suggest using PCA scores as a dependent variable in your ANOVA. I think that PCA Scores are sometimes used as a DV when dealing with correlated multivariate data, but

it seems a rather unusual approach to me in your case. However, I'm absolutely no expert on this and your approach here might be completely adequate. But I suspect this way of analyzing your data might strike many readers as rather unusual, which is why I would appreciate some more information why you specifically chose this approach.

I'm aware that due to the urgency of your research project, you want to avoid pre-testing. However, a pre-test on the two scales you plan to use might be assessed quickly and can help you understand the structure of the scales and drop items before administering them. Of course, you can also drop non-fitting items in your main study, but I think removing them beforehand holds the benefit that they won't influence participants' responses to the other items and create unnecessary variance.

Appendix G

6th June 2020

Dear Dr. Chambers,

We sincerely appreciate the reviewers' extremely prompt, valuable comments on our manuscript titled 'Don't spread' vs. 'Don't be a spreader' to prevent the (COVID-19) pandemic' (RSOS-200793), which is being considered at Stage 2 for publication as a Registered Report in *Royal Society Open Science*. Based on the comments, we have substantially revised the manuscript and have provided an amended version. All edited portions are in blue. Our individual responses to each comment posited are listed as follows.

Responses to Associate Editor

**Comments & Replies**

1-1 Four of the five original reviewers have now assessed the revised manuscript, and broadly agree that the proposal is significantly closer to meeting the Stage 1 criteria. However, there remain some significant issues to address before IPA can be awarded, primarily in terms of the conceptual framing of the study, appropriate use of statistical techniques (including methods for drawing conclusions from null results), and strengthening of the manipulation/quality checks. Please attend carefully to these points in a revision

Reply: We thank the Associate Editor for giving us the opportunity to revise our paper for IPA. We would also like to express our sincere gratitude to all the reviewers. Reviewer 2, in particular, reverted with much constructive advice in a short period of time. Moreover, comments by other reviewers, each of whom pointed out problems and concerns with the manuscript that we were unaware of, were very helpful. We have revised the manuscript based on this feedback and have replied to the comments to this effect. Some of the major changes are listed below.

1. Reference to a mechanism: Reviewer 1 had commented on this. As the introduction in the previous version did not mention the processes through which identity salience changes behaviour, we have now specified the intervention mechanism.

2. Calculation of the IP scale scores: Concerns were mainly raised by Reviewers 2 and 5. In the previous version of the manuscript and reply, we said that we would perform principal component analysis for each of the IP-intention and IP-behaviour scales and use the principal component score as the key dependent variable. However, considering the reviewers' concern about using the principal component score for ANOVA, we have decided to use mean scores of both the IP scales for ANOVA in the modified version. As both IP scales are scored on a Likert-scale, calculating mean is suitable.

3. Appropriateness of the statistical analysis plan: Concerns regarding this were mainly raised by Reviewers 1 and 4. We said, 'the change from the first wave (baseline) to the second wave would not differ significantly between the spreading and control conditions.' Furthermore, 'If there is no significant variation, we will interpret this [t-test] as indicating that the perceived concern for infection is essentially the same as before COVID-19, even under the influence of the COVID-19 pandemic.' In this regard, following the advice of Reviewer 1, we have decided to perform equivalence tests (Lakens et al., 2018) if there is no significant difference in the PVD scales between the current and non-pandemic situations, and if there is no significant variation between before COVID-19 and under the influence of the COVID-19 pandemic.

Responses to Reviewer 1

**Comments & Replies** (File name: 20200529-RSOSreview)

1-1 The authors resubmitted their manuscript intitled: Warning "Don't spread" vs. "Don't be a spreader" to prevent the COVID-19 pandemic, addressing reviewers' comments. I think that this improved the overall quality of the manuscript. Hopefully, the authors agree. Among other, the authors updated their literature review, removed the PVS from their main analysis, and are way more specific about how the study will be conducted. My main concerns were related to the strength of the rationale, detailed related to the sampling plans, and the appropriateness of the analysis. In my opinion, the authors successfully addressed some of these comments (namely, the ones related

to the sampling plans and the appropriateness of the analysis). However, I still think the authors should discuss (even if briefly) the mechanism involved in their intervention.

It would be kind of unfair from my position to simply ask to discuss the mechanisms without explaining why. Obviously, the present manuscript could have a very clear consequence from an applied perspective: if the expected results are found, the theoretical contribution could be minor, but the paper could still have a lot of value. However, I think that underlying the mechanisms that the authors think explains their effect could be of practical importance: Walton (2014) made the case for so-called wise intervention. By identifying the mechanisms involved, psychologists can design intervention targeting recursive processes, hence causing a lasting change. By doing so, we also have better chance to understand how an intervention could be context dependent. Right now, I have to say that I am not entirely clear on what would make this intervention work (the fact that it worked before is not a strong argument in my opinion). I do understand the time-sensitive nature of the situation, so I'll let the editor weight what would be desirable for this manuscript. However, I strongly encourage authors discussing mechanisms involved in their intervention: Why making an identity salient would make participants adopt more desired behavior?

Major

- Authors should discuss what is the process through which identity salience should impact adoption of behavior related to the spreading of Covid-19.

Reply: Initially, we would like to express our gratitude for Reviewer 1's extremely prompt review in this difficult time, in particular, the recommendation to discuss mechanisms in the manuscript. We consider the mechanism in which salient identity can influence and modify people's own behaviours as follows. Generally speaking, spreading infection is a negative and an undesirable behaviour. Nominalising the verb indicating such a behaviour with suffixes that represent agents ('-er') creates a strong link between an identity and negative self-image. In other words, people who read a reminder 'Don't be a spreader' come to consider infection spreader as part of their self-identity. As this threatens positive self-image, the avoidant motivation to circumvent such labels is driven to protect one's image, which makes ignoring the reminder difficult. As a result, people who read the reminder are more likely to

modify their behaviours to avoid spreading infection. This point has been stated in the revised Introduction section in the manuscript.

1-2 - Authors should be careful when citing research that haven't been peer-reviewed "recent research has found that manipulating the expressions of messages to appeal to citizens' emotions is effective in promoting behaviour to self-isolate". Right now, it is not entirely clear that this research is a preprint.

Reply: Certainly, our description might be too similar to cite preprint. Therefore, we have changed the description as follows: '*a recent paper reported that...*'.

1-3 - The authors should be explicit about their intention to make the (anonymized) data set of this experiment publicly available.

Reply: Implementing Reviewer 1's much appreciated advice, we will specifically state that the anonymized data set of the present study will be publicly available at <https://osf.io/dc7rs/>. This is reflected in the data analysis section of the revised manuscript.

1-4 - In the Quality checks for providing a fair test section, the authors say that "If there is no significant variation, we will interpret this [t-test] as indicating that the perceived concern for infection is essentially the same as before COVID-19 even under the influence of the COVID-19 pandemic.". If the authors want to make such conclusion, they should conduct the appropriate tests (see Lakens et al., 2018)

Reply: Following the advice of Reviewer 1, for which we are grateful, we will perform equivalence tests (Lakens et al., 2018) if there is no significant difference in the PVD scales between the current and pre-pandemic situations. In this regard, we have added a postscript in the quality checks for providing a fair test section of the manuscript.

1-5 **Minor**

- Right now, it is not crystal clear whether Klein et al. (2019) failed to replicate

mortality salience effects (see Chatard et al.'s analysis, 2020; doi: 10.31234/osf.io/ejubn). However, if significant this effect seems to be way smaller than originally thought.

Reply: We thank Reviewer 1's deep consideration and knowledge about mortality salience effects. Certainly, as the replication of mortality salience effects is not entirely clear, it is difficult to say with certainty that Klein et al. (2019) failed to replicate mortality salience effects. Therefore, we have modified this point in the revised manuscript as follows: 'studies (Klein et al., 2019; Sætrevik & Sjøstad, 2019) have also pointed out that mortality salience effects are small or not robust compared to the original study'.

1-6 - Authors should report their intention to conduct two one-way ANOVAs in their participants section. Moreover, they should be explicit that the DVs will be the change in IP-intention and IP-behavior.

Reply: We clearly stated the intention to conduct two one-way ANOVAs with different DVs (i.e., IP-intention and IP-behaviour scores) in the participants section.

1-7 - Authors report wanted to test whether the reminder condition differ. They report their intention to conduct Scheffé's F test to do so. I still don't understand why the authors want to adopt a conservative test whereas they could have used a priori contrast (see below). However, given the sample size, it should not make a big difference.

Table 1. A priori contrast to test authors hypotheses. C1 tests whether the reminder condition differ, C2 if they outperform the control condition.

	Don't spread	Don't be a spreader	Control
C1			
Do the reminder condition differ?	1	-1	0
C2			
Are the reminder condition better than no reminder?	-1	-1	2

Reply: Certainly, our main hypotheses focus on the comparison between the spreader and spreading conditions, or between the spreader and control conditions. However, as we stated in the manuscript, we want to test the main effect of the reminder conditions,

and then compare the three pairs to confirm which pairs would be significantly different. We believe that comparisons for all the pairs is important not only theoretically but also practically; these comparisons will shed light on which of the reminders is efficient based on the data—as also recommended by another reviewer. Therefore, to avoid inflation of the alpha level, we have decided to adopt Scheffe's F tests for adjusting the alpha level.

Responses to Reviewer 2

Comments & Replies

2-1 Firstly, I am pleased and impressed to see how much work the authors have put into improving the Stage 1 manuscript (study protocol/preregistration) in such a short amount of time.

...

Below I note the changes I think they still need to make before the manuscript should be accepted:

Hypotheses:

- The 2 hypotheses are now much clearer that they are measuring the change from wave 1 to wave 2.
- However, I still believe the hypotheses are not quite what the authors intend, based on my reading of the rest of the manuscript. It seems to me the authors do not care whether the spreading condition will differ from control (although they expect based on past results that it will not differ— either way, it would not affect their conclusions about the spreader condition, i.e., the condition they are interested in). Despite this, the authors have added into H1 and H2 the prediction that change scores will not differ between the spreading and control conditions. I don't think that is a main element of what they are interested in, and makes the hypotheses too complex. I think instead they should remove the last sentence of hypotheses 1 and 2, and just change the null hypothesis to state that the spreader condition will not differ from either the spreading or control condition. It could be interesting to test whether the two conditions of non-interest (control and spreading) differ, but I would either put that as

a separate, secondary hypothesis, or just include as an exploratory analysis.

Reply: We cordially thank Reviewer 2 for their prompt review and valuable comments at this unusual and difficult time. As pointed out, we have changed the null hypothesis to state that the spreader condition will not differ from either the spreading or the control condition.

2-2 • The “side hypothesis” about PVD should be listed in the hypotheses section and clearly indicated it is a secondary / less important / not main hypothesis (instead of only mentioning it in the Measures section). I apologize that my previous note on this from Round 1 (point 2-29 in the authors’ response) was unclear – I meant the authors should include just the hypothesis itself in the Hypotheses section, and then detail all the information about the measures themselves in a separate section within the methods, which they have now done.

Reply: We thank Reviewer 2 for deep consideration for our hypothesis. Following Reviewer 2’s suggestion, we have described main and secondary hypotheses in the Hypotheses section and detailed information about scales in the Measures section.

2-3 Measures

• I am pleased to see there is now more clarity about what each scale is attempting to measure, and the reasons for using these unvalidated scales (due to their tie to the government guidelines). However, a few remaining concerns below:

• I would highly recommend rephrasing the behavior scale items to ask about behaviors themselves rather than avoidance of behaviors. If you ask about “how often did you avoid x behavior” it is difficult for the participant to answer – it is much harder to think of what you have NOT done, versus what you did. So I would instead ask, e.g., “how often did you go out” and then reverse-code these items.

Reply: According to Reviewer 2’s important suggestion, we have changed the items asking about avoidance of behaviours to those asking about how often the participant did x behaviours. In addition, for a similar reason, the item ‘How often were you careful not to touch the mucous membranes of your eyes, nose, and mouth?’ is also changed as

follows: 'How often did you touch the mucous membranes of your eyes, nose, and mouth?' We will treat these items as reverse-code ones. Thanks to Reviewer 2's brilliant idea, we will conduct our investigation in a natural way without discomfort to the participants.

- 2-4** • Are you sure you want to have a Likert response scale of “never” to “often” for the reported behavior measure? I actually thought that asking participants to count the number of times they did an action was more objective. It is hard to know what going out “often” means. However I can see arguments for and against either option. I can foresee that reporting raw numbers could increase the variability in responses and give problems with non-normality (right skew), so that is one argument against it (although these issues can be attenuated by testing for skew and applying a transformation if there is skew). Another possible argument against participants entering raw numbers is if you want to be able to directly compare the behavior intention and reported behavior scales—but even then, the response scales are qualitatively different (agreement versus reporting frequency) even though they have the same number of scale points. I would think carefully about why you want to use a Likert scale versus reporting actual numbers of behaviors. If you care more about whether the manipulation will change total numbers of behaviors across a population, I would go for reporting raw numbers. If you care more about how many individuals the manipulation will convince to change their behavior, go for the Likert scale.

Reply: As Reviewer 2 said, we want to focus on the number of any individuals that will change their behaviour as a result of the manipulation. Moreover, as Reviewer 2 mentioned, the distribution of the number of times is likely to become a non-normality. Of course, we understand that some transformation might solve this problem, but it is difficult to expect what transformation is appropriated in advance. This means the possibility that we cannot avoid choosing the way of transformation after observing the data; in this case, pre-review becomes meaningless. Thus, to reduce the possibility of transforming the data as much as possible, we decided to use the Likert scale. We thank Reviewer 2 for giving us the opportunity to carefully consider using a Likert scale.

2-5 • Some items on the reported behaviour scale will need additional instructions, or include a “not-applicable” option, particularly number 2 and number 7. For instance, “how often did you go out wearing a mask” will depend on how often the participant went out, so it may be misleading (for instance a participant who did not go out at all would answer “never,” and therefore reduce their score – even though they followed guidelines by not going out!) Similarly, if participants did not cough or sneeze, they will get a low score on item 7, even though they are following guidelines. I would suggest giving participants a “not applicable” option for those 2 items, but if you cannot do that, you need to acknowledge that this might introduce noise to your measure and think of ways to address that (for instance, you might plan to re-run the analysis both with and without these 2 items).

Reply: We would like to thank Reviewer 2 for the important comments. Certainly, the items 2 and 7 assessing IP-behaviour do not correctly reflect answers from participants who followed guidelines and thus they are not appropriate. This could unnecessarily distort the data as Reviewer 2 pointed out. Therefore, we decided to remove the items 2 and 7 from the IP-behaviour scale.

2-6 • Related to the above note – item 2 is ambiguous and should be reworded to something like, “when you went out, how often did you wear a mask?” You care about the percentage of time people wore a mask in public, not the total amount of time (otherwise you would not know about how much of the time they went out in public NOT wearing a mask).

Reply: As we mentioned in comment 2-5, we decided to remove item 2 from the IP-behaviour scale.

2-7 • Despite the authors’ arguments why they don’t expect a ceiling effect, I think they still have to be prepared for the very real possibility of a ceiling or floor effect on both scales. Please consider how you will test for a ceiling/floor effect, and what you will do if you find one.

Reply: We are deeply grateful for Reviewer 2’s concern about a ceiling and floor effect. After

carefully discussing this point again, we believe a ceiling/floor effect is less likely to occur. Now, economic activity is returning in Japan and the Japanese government allows people to go outside and engage in social activities. Meanwhile, the Japanese government alerts the people about the second wave of COVID-19 particularly in Tokyo. Thus, under the current situation in Tokyo, people are free to engage in social activities but have to prevent the spread of COVID-19 in line with personal morality. Therefore, we never think that the distribution of scores is heavily biased in one direction (ceiling/floor). In addition, as we mentioned repeatedly, our main focus is to make a more practical contribution rather than a theoretical one and examine whether people come to obey the current published Japanese administrative guidelines as a means to promote public health via our reminder. Thus, changing the scale means that we cannot examine our interest.

However, we can also understand Reviewer 2's concern. Presumably, we can guess whether the ceiling/floor effect would have occurred from the raw data. Additionally, even if the ceiling/floor effect occurs, we will speculate attitudes of Japanese people to COVID-19, the Japanese administrative guidelines, and our reminders within a reasonable range.

- 2-8**
- **CRUCIAL:** I unfortunately do not have enough knowledge about principle component scores to know whether they are appropriate to use as a dependent variable, but I have doubts. These should be checked by an expert. I gather using the principal component scores is an attempt to weight the items according to how well they cohere with the overall (forced single-factor) scale? I am not sure about that. – I would report an analysis that uses mean scores on each scale, regardless of whether you also use a principal components score. It is easier to understand and does not make “invisible” assumptions about how to weight the various items on the scale. Also, I am not sure whether it is even appropriate to calculate a linear change score between 2 different principal component scores. That doesn't make sense to me how this will capture change in participants' behavior – instead won't it capture change in the relative weighting of different items on the scale, because the factor loadings will be idiosyncratic to each PCA? If everyone's 2nd-Wave scores remained exactly the same

as 1st-Wave except that they all went up by 1 Likert scale point, then wouldn't the principal component scores remain the same both times, even though behavior changed considerably? (But I may be misunderstanding principal component scores, I am not an expert!)

Reply: We appreciate Reviewer 2's important comments. As per Reviewer 2's and other reviewers' comments, performing PCA will cause a fatal problem in the present study. In the previous review, we followed Reviewer 3's suggestion about calculating scale scores. However, as Reviewer 2 pointed out, we would like to prevent the loss of IP items by the PCA from the perspective of practical implication. Therefore, given the suggestions and concerns of Reviewer 2 and other reviewers, we have decided not to conduct a PCA and we will report the mean scores for each scale as we planned at first.

2-9 • Related to the above I think it's good to do a PCA to see how well the items on your scales cohere into a single measure, but since you have an a priori reason to use those items (they are the behavior guidelines the government wants to influence) I'm not sure it's appropriate to drop any of them simply because they don't fit in a single-factor model. You stated before that you don't care whether these items all are affected using the same mechanism (i.e., you don't have hypotheses about approach versus avoidance behaviors) and I think the same principle applies here—you include the items because of the practical consideration of wanting to measure those specific behaviors, so you should include them all somehow. That could be forcing them to be all in the same scale regardless of fit, or separating them into separate factors based on the PCA.

Reply: As we responded to comment 2-8, we decided not to conduct a PCA.

2-10 Attention Check questions

• I like the new attention check questions, and it is good that they are text-based. However, since the participants are now answering everything on a Likert scale, they will still stand out if participants have to enter text! I would give them a Likert scale

answer (unless you change the reported behavior scale back to number-entry).

Reply: This suggestion is highly valuable. Therefore, we have changed text-based answer to Likert-scale answer in ACQs. In addition, we decided to inset ACQs in IP-behaviour scale. Accordingly, to be consistent with expressions of the other items, we have changed the description of ACQ as below: ‘In this survey, how often have you been asked about your blood type so far?’ for the first wave and ‘How often did you go to Mars?’ for the second wave. We will exclude participants who do not answer ‘1 (not at all)’ to these ACQs.

2-11 • I am not convinced the instructional manipulation check will be effective. It is merely another attention check, and does not in any way guarantee the participant read the message. You must include a test that can tell whether or not the participant understood the message – I would highly recommend asking them to type it back on the next page (or you could ask them to choose it from a list of options—in this case I would not include both spreader and spreading message in the options, I would include other options that don’t mention “spread” in any way, such as “be safe” or “don’t be reckless”). Are you concerned with external validity, because in real life people are not asked to type back messages, or with demand characteristics (that participants might figure out the study is about the message)? I would not be concerned with these. I would be more concerned that the message gets lost and participants do not see it.

Reply: From the conclusion, we will not change our instructional manipulation check (IMC). We can deeply understand Reviewer 2’s concern for the reminders getting missed. However, Reviewer 2’s suggestions for check items seem to be unsuitable for our experiment and there is weak evidence that they are better than our IMC. Firstly, both of Reviewer 2’s suggestions (i.e. typing and choice of options) cannot be applied for the control condition (the no reminder condition). This is a fatal problem. Next, while typing the reminder it is easy to produce minor errors (e.g. simple typographical errors) or orthographical variants. Reviewer 2 might argue that this issue can be addressed if we set a criteria for typographical errors. However, it is very hard to set objective criteria for what expressions to exclude as error. For example, the reminder

‘拡散者にならないでください’, which means ‘Don’t be a spreader’, can be mistyped as ‘拡散者に~~な~~まらないでください’. For another example, the reminder ‘拡散しないでください’, which means ‘Don’t spread’, can be mistyped as ‘拡散~~を~~しないでください’. These typo examples of the reminders are almost the same in meaning in Japanese as the original reminders but only the particles are different. Many other variations of expression are also possible in Japanese while maintaining the meaning of the original reminders. Thus, it is too difficult to set the quantitative criteria in advance and this generates the possibility that we cannot avoid setting or changing the criteria ‘after’ collecting and observing data; this means that pre-review is meaningless. Moreover, the aim of our research focuses on the practical aspect as we have repeatedly said. Forcing them to do something unnatural such as typing reminder would undermine the significance of our research because in real life no citizen has the opportunity to memorize administrative reminders by typing. Therefore, we cannot accept this kind of check item. Next, we also disagree with choosing the right reminder from a list of options. Reviewer 2 suggests that the messages ‘be safe’ or ‘don’t be reckless’ should be included as other options. However, the other options such as ‘be safe’ itself is highly possible to work as reminders. In order to avoid this fatal contamination, we cannot adopt this style of the check item.

On the other hand, our IMC can be set in the control condition as well. Our IMC can also ask the participants to read the reminders naturally and certainly because the image of the instruction is larger and more salient than the text of the IMC and thus will automatically capture visual attention of the participants (e.g., Kerzel & Schönhammer, 2013; Theeuwes, 2010). Additionally, it is impossible to correctly answer our IMC without observing the entire page carefully. Therefore, Reviewer 2’s suggestions are not quite suitable for our study and have many disadvantages compared to our IMC. We are convinced that our IMC is the best and the most effectively work for excluding participants who do not read the reminder.

Reference

Theeuwes, J. (2010). Top-down and bottom-up control of visual selection. *Acta psychologica*, 135(2), 77-99.

Kerzel, D., & Schönhammer, J. (2013). Salient stimuli capture attention and action. *Attention, Perception, & Psychophysics*, 75(8), 1633-1643.

2-12 Small points:

- The section on ACQs still says “We will insert a simple math problem as an attention check question” (page 37 line 32)

Reply: We appreciate Reviewer 2’s check for detail. We have amended this sentence in the revised manuscript.

2-13 • In the reply to point 2-21, the authors state that they will ask participants to answer both scales based on the week before participation in each wave. Shouldn’t the behavior intention scale be based on the upcoming week (the week after each wave) instead? Behavioral intention is usually measured regarding the future. Have I misunderstood? Figure 1 is generally helpful but it does not differentiate between the two scale measures.

Reply: We sincerely appreciate Reviewer 2’ important comments. As Reviewer 2 pointed out, behavioural intention should be measured regarding the future. Therefore, we will ask participants to answer about their intention of upcoming behaviours (regardless of when they behave) with the IP-intentions scale in the revised manuscript. According to this change, we have amended the relevant parts of manuscript and Figure 1.

Responses to Reviewer 4

Comments & Replies

4-1 In general, the authors have followed most of the recommendations and this revision addresses all of my concerns.

I've two minor suggestions for improvement.

1) In the abstract, the authors use the word "actual behavior" but they will measure behavioral intentions in both IP-intention and IP-behavior scales. Both measures are based on participants' self-reported intentions. To measure actual preventive behaviors, the authors should not use a self-report measure; rather, they can use location data such as <https://www.unacast.com/>.

Reply: We would like to thank Reviewer 4 for the important point and suggestion on our

measurement. The ‘actual behaviours’ in the abstract will be measured by the IP-behaviour scale (‘Behavioural intentions’ will be measured by the IP-intention scale). In the IP-behaviour scale, participants have to answer each item based on their own behavioural history during the past week. Thus, although this scale is a self-assessment, the IP-behaviour scale is reflected in the actual behaviour of the participants. For the above reasons, we will use the IP-behaviour scale to measure ‘actual behaviours’.

4-2 2) If a null-finding is hypothesized, a Bayes Factor should be calculated (i.e., "The change from the 1st wave (baseline) to the 2nd wave will not significantly differ between the spreading and control conditions.)."

Reply: We deeply thank Reviewer 4 for this important point. Performing only ANOVAs cannot confirm a null-finding even if they indicate that there is no significant difference. However, considering that the series of our analyses are null hypothesis significance testing, it seems inappropriate to use a Bayesian statistical approach only for confirming whether the null-finding is favoured. Alternatively, we will perform equivalence tests with TOSTER (e.g. Lakens et al., 2018) if there is no significant difference. We have clearly stated this point in the revised manuscript.

Responses to Reviewer 5

**Comments & Replies** (File name: RSOS-200793R1)

5-1 I have read a previous version of the manuscript and I appreciate the work the authors put into the revision of the paper and their effort to comply with the reviewers' and editor's requests. I still believe that the addressed research question could make a valuable contribution to the current situation and literature and is worth being investigated and published. However, I still noted some aspects in the revision of the manuscript which in my opinion need editing, which keeps me from recommending acceptance of the manuscript at this point. In the following, I will outline these critical aspects in more detail:

Reply: We sincerely appreciate Reviewer 5’s helpful suggestions to our revised manuscript. We were happy to reply to Reviewer 5’s comments as follows.

5-2 Introduction:

I appreciate that the authors put effort in revising the introduction part, they did add important information on the findings in similar studies. However, I still miss an integration of the research question into the general literature framework. The authors are listing similar research findings that support the assumption that their intervention might work, but they still don't refer to the psychological background and meaning of social identity. As far as I understand it from the journal's guidelines, after acceptance of the manuscript at stage 1 the introduction cannot be altered in stage 2. If, besides being able to provide recommendations for policy makers regarding the current situation, the submitted manuscript also is supposed to make a sustainable contribution to the field's literature (in the sense of a conceptual replication that might support the validity of other findings on social identity), I would appreciate a more exhaustive elaboration on the psychological construct of social identity.

I further do not understand the reference to Mortality Salience in the introduction part, since to me it isn't clear how this relates to the topic of social identity:

On the other hand, another intervention is to evoke mortality in messages based on terror management theory (Greenberg et al., 1994).

Reply: The main objective of this study is not to replicate the social identity conceptually, but to establish an applied approach from the psychological perspective to thwart the COVID-19 pandemic. Therefore, we added further detail in Introduction the mechanisms by which small changes in the teaching text inhibit behaviours that lead to the spread of infection. Moreover, we presented Terror management theory just as an example of intervention methods other than our methods (in other words, the way to highlighting the self-identity).

5-3 Method:

The randomized assignment for the three reminders will be based on the participant's birthday: Participants whose birthday is the 1st–10th of the month will be assigned to the spreader condition, those with the 11th–20th to the spread condition, and those with the 21st–31st to the control condition.

Strictly speaking, this is not a randomized assignment since there is clear pattern in

the way participants are assigned to the conditions. What if there actually is an external cause for participants born at the beginning of the month to behave differently than participants born at the end of the month? This might seriously threaten the validity of the authors' findings and I would strongly recommend to use a more valid approach of randomization (e.g. a random number generator, should be implemented in most survey software)

Reply: We thank Reviewer 5 for suggesting our survey method. After careful consideration, we did not use random number generator but still plan to adopt the way of randomisation based on birthdays. We have explained the reasons as follows:

1. Disadvantages caused by using random number generator

The crowdsourcing site, which we will use, does not have a random number generator, nor is it possible to incorporate a random number generator in the survey form. Therefore, if we adopt Reviewer 5's idea, it is necessary for the participants to access an external page of the random number generation tool apart from the survey page. In this case, we cannot verify whether the participants will correctly use the tool to generate the random numbers. We are also concerned about whether the participants will return to the survey after completing the random number generation in an external web site. Furthermore, adding transitions to an external tool to the survey may interfere with the concentration of the participants and produce technical errors (e.g. accidentally erasing the survey page).

2. Few possibilities of the impact of birthdays on behaviour

There is no scientific evidence that birthdays can be an external factor in determining behaviour. Thus, using birthdays for the assignment to the conditions is less likely to produce the fatal contamination. However, we understand the intuitive interest of Reviewer 5 and perhaps some readers have the same idea. For them, we will provide the dataset including birthdays of the participants at OSF and make it available for exploratory post-hoc analysis.

5-4 Data Analysis:

I understand that you aim at doing a PCA to eliminate items that are not consistent with the construct you want to measure. Therefore, you suggest to eliminate items

with a low loading on the first principal component. As PCA is an explorative approach, there might come up more than just one relevant component that explain a substantial part of the variance in your data, which however you would just drop (is that necessary/useful?).

Further you suggest using PCA scores as a dependent variable in your ANOVA. I think that PCA Scores are sometimes used as a DV when dealing with correlated multivariate data, but it seems a rather unusual approach to me in your case. However, I'm absolutely no expert on this and your approach here might be completely adequate. But I suspect this way of analyzing your data might strike many readers as rather unusual, which is why I would appreciate some more information why you specifically chose this approach.

Reply: We thank Reviewer 5 for the valuable comments. As Reviewer 5 pointed out, the PCA seems to have some problems in the present study. Indeed, the other reviewer also pointed out the problems of using the PCA in the present study (Please see our reply to Reviewer 2). Thus, considering these valuable suggestions, we have decided not to perform PCA in the present study.

5-5 I'm aware that due to the urgency of your research project, you want to avoid pre-testing. However, a pre-test on the two scales you plan to use might be assessed quickly and can help you understand the structure of the scales and drop items before administering them. Of course, you can also drop non-fitting items in your main study, but I think removing them beforehand holds the benefit that they won't influence participants' responses to the other items and create unnecessary variance.

Reply: We deeply thank Reviewer 5 for this comment. After carefully considering again, we decided not to conduct a pre-test and not to perform the PCA, and hence the scales' items will not be deleted. The reason for reaching such a decision is that we are aiming at the practical effect on the acceptance of administrative instruction in this study. From this perspective, we have prepared the scale in line with the administrative instruction. Thus, we realized that dropping any item is inappropriate for our main aim, and therefore, cancelled the plan to drop the items by performing the PCA. Now, our plan is suitable for our purpose.

Again, we would express our sincere gratitude to the reviewers for all of their prompt, thoughtful, and constructive comments. We hope that our revised manuscript is now suitable for proceeding to Stage 2 Registered Reports in *Royal Society Open Science*.

Sincerely,

Fumiya Yonemitsu

Kyushu University

744 Motooka, Nishi-ku, Fukuoka 819-0395, Japan

Phone/Fax number: ++81-92-642-2418

E-mails: y.fumiya.0408@gmail.com

Appendix H

3rd August 2020

Dear Editor of Royal Society Open Science,

Please find enclosed our manuscript titled, ‘Warning “Don’t spread” vs. “Don’t be a spreader” to prevent the COVID-19 pandemic’ for consideration for publication as a Registered Report (Stage 2 manuscript) in *Royal Society Open Science*. Our research aimed to examine whether a subtle linguistic difference in the wording of reminders for preventing the spread of COVID-19 would change people’s behaviour.

We conducted an experiment and analysis following a pre-registered procedure. The paper described the outcomes and constraints of our survey. We did not obtain positive results and thus concluded that highlighting self-identity has little effect on changing intention and behaviour regarding infection prevention. We believe that the present study contributes to the understanding of the psychological tendency regarding infection prevention and disease under the COVID-19 pandemic situation.

We explained deviations from the experimental procedure in a footnote in the paper. Additionally, we have uploaded the dataset and the R code, including analysis output and visualisation of results in OSF. All the authors have read and approved the manuscript, and we declare that it is not under consideration for publication elsewhere. Furthermore, the authors have neither conflicting interests nor any issues related to the journal’s policies. On behalf of my co-authors, I will be acting as the corresponding author., We thank you very much for your consideration.

Yours sincerely,

Fumiya Yonemitsu

Graduate School of Human-Environment Studies, Kyushu University

744 Motoooka, Nishi-ku, Fukuoka 819-0395, Japan

E-mail: y.fumiya.0408@gmail.com

Tel/Fax: +81-92-642-2418

Appendix I

Review RSOS Stage 2

Overview:

As in my round 1 reviews, my main concern about the study is the lack of interpretability of the null results, given that I do not believe the manipulation check is sufficient to show that participants read and internalized the public health message (i.e. the manipulation – “don’t be a spreader” versus “don’t spread” or no message). As I pointed out in Stage 1, the manipulation check the authors used was more of an attention check and did not assess whether participants had actually read and understood the manipulation message. Additionally, as I flagged in Stage 1, the fact that the measurement scales were not piloted or validated means that the resulting ceiling effects (or other possible noise in the data) may have affected the interpretability of the results. However, I acknowledge that the editor accepted the manuscript in principle, including the manipulation check and measurement scales, so this cannot be held against the authors at this stage since they carried out their study as approved.

As detailed below, I believe the other most important issues to address are the way the main results are interpreted (need to make it clear that the results are not conclusive either way) and to make the discussion section more clear, understandable, and relevant.

RR Specific review questions:

Whether the data are able to test the authors’ proposed hypotheses by passing the approved outcome-neutral criteria (such as absence of floor and ceiling effects or success of positive controls)

As detailed in Stage 1 review, I do not believe the manipulation check was sufficient, and the authors also report a possible ceiling effect in one of their scales. These need to be acknowledged in the discussion at least, as unfortunately they weaken the evidential value of the study in my view.

Whether the Introduction, rationale and stated hypotheses are the same as the approved Stage 1 submission

I did not have enough time to read through the whole Stage 1 manuscript (and I also had to hunt to find the Stage 1 ms itself!), but a quick glance suggested it was similar at least. The hypotheses were the same.

Whether the authors adhered precisely to the registered experimental procedures

Everything I checked had adhered to the registered procedures, but in some cases the registration was under-specified (for instance, the equivalence test did not specify a SESOI, smallest effect size of interest to test against).

Where applicable, whether any unregistered exploratory statistical analyses are justified, methodologically sound, and informative

The only exploratory analyses were the extra equivalence tests for the comparisons between additional pairs of conditions—I would say these were justified and informative but as noted below they need to report more information on the tests and explain the interpretation of the tests.

Whether the authors' conclusions are justified given the data

The authors need to reframe their conclusions, given the equivalence tests they ran – the data are not conclusive either way, and the authors' conclusions should reflect that.

Data:

I was able to find and download the data and codebooks from the OSF, and at face value they look ok. However I did not actually try re-running the analyses on the data, due to the short turnaround of this review.

Registration:

There is no registration for this manuscript on the OSF link provided. There is a registration-style document on the OSF project from 4 May 2020 (around the time of the initial Stage 1 submission, I believe), and appears to be based on the original first draft of the manuscript **before** Stage 1 revisions. It is in a question and answer format (presumably one of the OSF preregistration formats) rather than in the format of a Stage 1 manuscript, despite the document name (“RSOS_COVID_RRStage 1_MS.pdf”). **This lack of a formal registration (or at least an upload of the full accepted Stage 1 manuscript) makes it impossible for me (or other readers) to check consistency of the Stage 2 manuscript with the in-principle-accepted Stage 1 manuscript, which is one of my main tasks as a reviewer of a Registered Report.** For the purposes of this review I am assuming that the manuscript submitted 6 June 2020 is the in-principle-accepted Stage 1 manuscript, and am comparing it to that. However, I recommend the authors to upload the in-principle-accepted Stage 1 manuscript to the OSF project as well.

Results:

Results: PVD – please report the means, standard deviations, and sample sizes for each sample you compared for each measurement. This could be in a table or the text.

Figure 3 – it would be helpful to label the “before” and “after” labels by the dates collected, e.g. (“Before COVID-19 onset: 1 January 2018”) and (“After COVID-19 onset: 5 June 2020”) or whatever the actual dates were.

On page 15 and page 16 when you discuss the PVD results, it would help to rephrase the results to keep the order of the timepoints constant, e.g., say score A was higher pre-pandemic compared to mid-pandemic, and score B was lower pre-pandemic compared to mid-pandemic. A fast or not-careful reader might misinterpret the results to indicate that both scales were higher in the pre-pandemic timepoint (that was my first impression on a quick reading) because they are just scanning for the words “higher” or “lower”. I was especially confused by this in the wording on page 16, 2nd paragraph of discussion.

Page 15, lines 6-8 – please phrase the results of the equivalence tests as suggested in the TOSTER paper, and report the smallest effect size of interest that you tested for (i.e., the lower and upper bounds of equivalence that the TOST tested against) for each test. The

TOST statistics are not meaningful without reporting that information. I would also request that you interpret/explain what these results mean for your claims, for readers who are not familiar with equivalence tests— in this case, if both the experimental test and the equivalence test have $p > .05$, then it suggests your data cannot conclusively say either that there *is* or *is no* meaningful effect, either way.

Discussion:

Page 16, Discussion (first paragraph) –

- Instead of “reduce” intentions/behaviors, perhaps “affect” would be better as it is more general (you expected the intervention might reduce some intentions/behaviors and increase others)
- The second sentence is confusing and not quite right. Firstly, you tested changes in behavior or intention scores, not the scores themselves. Secondly, the wording makes it sound like you were testing between behavior and intention scores, rather than each separately.
- In sentence 3, after you mention there is no statistical support for Hyp 1 and 2; to fully report your results, you should mention that there is also no statistical support AGAINST Hyp 1 and 2 (as per the non-significant equivalence tests)

Page 16, paragraph 3 – instead of launching straight into possible *theoretical* explanations for lack of conclusive results, I would first outline or emphasize the limitations of *evidence* in this study – that you did not find conclusive evidence in either direction, and so the effect size may be smaller than you guessed, and/or the data may be noisier than you had hoped. I think it is important to acknowledge this first, before speculating about theoretical or methodological concerns.

I really struggled to understand the 4th paragraph of the Discussion, page 16-17.

- The paragraph seems to argue two opposite conclusions – that according to PVD results people might have reduced infectious behavior (fear of infection) but also increased it (lower perceived susceptibility). Are you suggesting both could be true at the same time? I don’t see how that could be the case. And why would that explain why you got the results you did? The first argument (people already reduced high-risk infectious behaviors, as suggested by germ aversion on the PVD) is consistent with the point that you might have ceiling effects. But shouldn’t the second argument (that people might underestimate potential to be a spreader) be working against any ceiling effects of anti-spreading behavior/intentions?
- You wrote “The present study’s data of IP-intention score may tend to be close to a ceiling effect, and therefore germ aversion may influence IP intention rather than behaviour.” Why should a ceiling effect on intention scores indicate that germ aversion had an effect on it? And why should it indicate that germ aversion did NOT affect behavior? This seems like pure speculation. You do not have data for IP intentions or behavior from the previous PVD data collection, so you don’t know how changes in intentions or behavior might be related to germ aversion PVD levels.
- I found the second half of the paragraph hard to follow as well – I don’t understand how the logic leads you to your conclusions.

- Perhaps it would be better to just outline a simpler explanation/speculation in this paragraph --that the ceiling effects left little room for differences in intention, and that the levels of attitudes around disease (as measured by PVD) might have an unknown effect on the results (and any further interpretation would be speculation).

Page 17, paragraph 2—I'm not sure I understand the first sentence, "Moreover, the results of IP may reflect the degree of effect of highlighting self identity." Do you mean that you may have gotten inconclusive results because the true effect size is very small? If so, I would just state that!

Page 18, paragraph 2 (limitations)

- please rethink/rephrase your conclusions about the study's findings. You wrote "However, this indicates that at least the procedures in our study could not change IP-intentions and behaviour." However, the results of the equivalence tests combined with the hypothesis tests indicate that you cannot rule out either outcome – the data are inconclusive. You could instead state that your data could not conclusively show that the procedures changed intentions or behavior. (Or perhaps this sentence could simply be omitted and the paragraph focus solely on the limitations.)
- instead of just considering internal validity (what methodological changes might result in a conclusive effect), what about also considering external validity – how well do the study stimuli reflect the way citizens might encounter such messaging "in real life" if a government or other entity decided to adopt the self identity messaging? The current study design could potentially be considered a merit rather than a limitation in some ways, as perhaps it reflects the passive messaging and sometimes long time lags between message and behavior that you might find in a modest messaging campaign. Likewise, regarding the potential effects you discuss of increasing the frequency of exposure to messaging—instead of just speculating that it might increase likelihood of observing effects, it would be more interesting to me as a reader to consider how it relates to real world situations (e.g., the lab experiment you describe might provide information on how a well-funded, centralized and co-ordinated messaging campaign might affect responses.)
- in the limitations section, I would also suggest highlighting other ways the study methodology might be improved. By "improved," I do not mean better chances of finding a significant effect, but rather a better reflection of whether the effect exists in the real world. What changes might allow you to make more definitive claims about the hypotheses you are testing? For instance, validating and refining the measures might have helped make the data less noisy, or having a better manipulation check might have allowed you to test whether people actually absorbed the message.
- Lastly, I would suggest that the authors also discuss the limitation that the 'manipulation check' cannot actually ascertain whether participants read the message, and the possibility that many participants simply did not read it.

Page 18, last paragraph – again, I do not follow the logic of this paragraph or understand the argument it is attempting to make. Could this be phrased more clearly and directly? For instance, I wasn't sure if the low case numbers were meant as an argument that people

were already following safe behaviors, or that low numbers meant people perceived low risk and therefore the messaging simply didn't work at all. I also am unclear what is meant by "If the threat of COVID-19 spread increases, people can easily follow the reminders." However these were not the only issues with the paragraph -- I'm sorry, I simply didn't understand any of the points or how they follow on from each other.

Page 19, paragraph 1 – again, I am having trouble following the argument of this paragraph. Is the argument that self identity effects have been found in many cultures? Or that they have failed to be found in many cultures? I don't follow the logic of the last several sentences either. Could this be rewritten more clearly and simply, perhaps just stating what evidence there already is about whether self- identity effects are cross cultural.

Page 19, paragraph 2 – again, the data do not quite support this conclusion, that the manipulation "has little effect on changing IP intention and behaviour." Instead, the data simply do not provide support for (or against) the hypotheses

Minor points:

I recommend that, in the codebooks (description of data text files), the authors should add in information on the allowable range of responses (1-7 I believe?) and indicate what those numerical responses corresponded to (e.g., 1=strongly disagree, 7 = strongly agree). This would be very helpful but is not strictly required.

Page 6, top paragraph – the wording is a bit unclear. I suggest something like "As a secondary hypothesis for this research, we additionally predicted that the scores ... [etc.]"

Page 12, line 29 (Main Analysis) – could the first sentence clarify that the 2 mean scores for each measure are the before and after (or time 1 and time 2) measures? I did not understand this at first and had to read it a few times before realizing what it must mean.

Page 12, line 39 – the last word of the paragraph should be variables, plural. You have written variable (singular), which makes it sound like they have been combined – but I gather from the paragraph below that you actually did not combine, but ran separate tests for each.

Page 14, line 17-18 – I don't understand what this sentence means: "However, the total number of the participants excluded by each criterion did not match the total number of excluded participants." Could you briefly explain more what you meant, and why this is important?

Page 14, line 27 – I think the PVD results should be listed last in this section (after the intentions and behaviors results) as it is a secondary hypothesis. Put the most important, central results first.

When referring to the PVD in text (e.g., page 17 line 47, but elsewhere as well), instead of "current situation" it may be useful to say "during pandemic" or "several months into the

COVID-19 pandemic”, or even “June 2020” as it is unclear when exactly “current situation” refers to (and will be even harder for readers in future years to pinpoint).

Appendix J

15 September 2020

Dear Dr. Chambers,

We sincerely appreciate the reviewers' extremely prompt, valuable comments on our manuscript titled 'Warning 'Don't spread' vs. 'Don't be a spreader' to prevent the COVID-19 pandemic' (RSOS-200793.R3), which is being considered for publication as a Registered Report in the *Royal Society Open Science*. Based on the really constructive comments, we have substantially revised the manuscript, especially the results and discussion sections and the supplementary code, and have provided an amended version. All edited portions are in blue. Our individual responses to each comment posited are listed as follows.

Responses to Associate Editor

Comments & Replies

1-1 Three of the original Stage 1 reviewers were available to assess the Stage 2 manuscript. In general, the reviewers find that the Stage 2 criteria are nearly met, but with some work especially required to improve the clarity of reporting and the Discussion especially (including ensuring that the conclusions are justified by the evidence, that the arguments are logically coherent, and highlighting key limitations such as the observed ceiling effect). Concerning the point raised by Reviewer 2 about the unavailability of the accepted Stage 1 protocol, the URL to the Stage 1 protocol was stated as required at the end of the Abstract (and I have discussed this issue already with the reviewer), but I agree that more could be done to flag it for readers. Therefore, please also include in the Method section the sentence in the Abstract that lists the URL to the Stage 1 protocol.

Reply: We sincerely thank the three reviewers for their valuable comments on the manuscript's Stage 2 titled 'Warning "Don't spread" vs. "Don't be a spreader" to prevent the COVID-19 pandemic', following the Stage 1. Based on their comments, we have changed sentences and words that were unclear so as to make them easier to understand for the readers. In this process, we revised some descriptions in Methods, according to the reviewers' comments. Moreover, since there were careless errors in the code of the statistical analyses, we reanalysed the data. As a result, we have modified our claim. Specifically, the effect of the reminders was significantly equivalent among the conditions. Additionally, we added the URL of the final version of the stage 1 manuscript. Our responses to each of the comments posited by the reviewers are listed below.

Responses to Reviewer 2

Comments & Replies

2-1 Overview: As in my round 1 reviews, my main concern about the study is the lack of interpretability of the null results, given that I do not believe the manipulation check is sufficient to show that participants read and internalized the public health message (i.e. the manipulation – "don't be a spreader" versus "don't spread" or no message). As I

pointed out in Stage 1, the manipulation check the authors used was more of an attention check and did not assess whether participants had actually read and understood the manipulation message. Additionally, as I flagged in Stage 1, the fact that the measurement scales were not piloted or validated means that the resulting ceiling effects (or other possible noise in the data) may have affected the interpretability of the results. However, I acknowledge that the editor accepted the manuscript in principle, including the manipulation check and measurement scales, so this cannot be held against the authors at this stage since they carried out their study as approved. As detailed below, I believe the other most important issues to address are the way the main results are interpreted (need to make it clear that the results are not conclusive either way) and to make the discussion section more clear, understandable, and relevant.

Reply: We thank Reviewer 2 for reviewing our paper, and providing detailed and informative suggestions. We have made every effort to address the issues. We hope the revised manuscript is suitable for publication.

RR Specific review questions:

2-2 Whether the data are able to test the authors' proposed hypotheses by passing the approved outcome-neutral criteria (such as absence of floor and ceiling effects or success of positive controls)

As detailed in Stage 1 review, I do not believe the manipulation check was sufficient, and the authors also report a possible ceiling effect in one of their scales. These need to be acknowledged in the discussion at least, as unfortunately they weaken the evidential value of the study in my view.

Reply: As suggested by Reviewer 2, in the 'Discussion' section, we have now described the manipulation checks and a possible ceiling effect as limitations.

2-3 Whether the Introduction, rationale and stated hypotheses are the same as the approved Stage 1 submission

I did not have enough time to read through the whole Stage 1 manuscript (and I also had to hunt to find the Stage 1 ms itself!), but a quick glance suggested it was similar at least. The hypotheses were the same.

Reply: We thank Reviewer 2 for checking the consistency of our hypotheses.

2-4 Whether the authors adhered precisely to the registered experimental procedures

Everything I checked had adhered to the registered procedures, but in some cases the registration was under-specified (for instance, the equivalence test did not specify a SESOI, smallest effect size of interest to test against).

Reply: In both the Stage 1 and 2 manuscripts, there were no statements about the effect size of the equivalence test, including a SESOI. Based on a study by Lakens et al. (2018), we had planned to set the value of equivalence bounds as ± 0.5 , but we realised that we had erroneously stated the value setting as ± 0.05 . We sincerely apologize for this mistake and for omission of the information. In the revised version, we corrected the value of the equivalence bounds which we reset as ± 0.5 and reconducted equivalence tests with two one-sided tests. As a result, the equivalence tests showed significant equivalences across the spreader, spreading, and control conditions. The details of the settings of equivalence tests have been added to the 'Results' section as a deviation because it had not been pre-registered. Similarly, the significant equivalences were treated as being the results of the post-hoc analyses and included under 'Discussion'.

2-5 Where applicable, whether any unregistered exploratory statistical analyses are justified, methodologically sound, and informative

The only exploratory analyses were the extra equivalence tests for the comparisons between additional pairs of conditions—I would say these were justified and informative but as noted below they need to report more information on the tests and explain the interpretation of the tests.

Reply: We have checked Reviewer 2’s comments and revised our manuscript according to each of them. For the details, please refer to the following replies.

2-6 Whether the authors’ conclusions are justified given the data

The authors need to reframe their conclusions, given the equivalence tests they ran – the data are not conclusive either way, and the authors’ conclusions should reflect that.

Reply: We have now confirmed the validity of our conclusions. As already mentioned in our reply to 2-4 of Reviewer 2’s comment, we reanalysed the data and have changed our claim. Now, our conclusions reflect the results.

2-7 Data:

I was able to find and download the data and codebooks from the OSF, and at face value they look ok. However I did not actually try re-running the analyses on the data, due to the short turnaround of this review.

Reply: We thank Reviewer 2 for checking the data codebooks.

2-8 Registration:

There is no registration for this manuscript on the OSF link provided. There is a registration-style document on the OSF project from 4 May 2020 (around the time of the initial Stage 1 submission, I believe), and appears to be based on the original first draft of the manuscript **before** Stage 1 revisions. It is in a question and answer format (presumably one of the OSF preregistration formats) rather than in the format of a Stage 1 manuscript, despite the document name (“RSOS_COVID_RRStage 1_MS.pdf”). **This lack of a formal registration (or at least an upload of the full accepted Stage 1 manuscript) makes it impossible for me (or other readers) to check consistency of the Stage 2 manuscript with the in-principle-accepted Stage 1 manuscript, which is one of my main tasks as a reviewer of a Registered Report.** For the purposes of this review I am assuming that the manuscript submitted 6 June 2020 is the in-principle-accepted Stage 1 manuscript, and am comparing it to that. However, I recommend the authors to upload the in-principle-accepted Stage 1 manuscript to the OSF project as well.

Reply: We thank Reviewer 2 for checking our OSF page. We have added in the Data accessibility section the URL of the final version of the 1st-stage manuscript (<https://osf.io/kz5y4>).

2-9 Results:

Results: PVD – please report the means, standard deviations, and sample sizes for each sample you compared for each measurement. This could be in a table or the text.

Reply: We have reported the means, standard deviations, and sample sizes for each sample in the Results section.

2-10 Figure 3 – it would be helpful to label the “before” and “after” labels by the dates collected, e.g. (“Before COVID-19 onset: 1 January 2018”) and (“After COVID-19 onset: 5 June 2020”) or whatever the actual dates were.

Reply: We appreciate the above-mentioned advice relating to Figure 3. Considering this along with the comment in 2-27, we changed the ‘before’ and ‘after’ in Figure 3 to ‘Before the COVID-19 pandemic (September 22–23, 2018)’ and “During the COVID-19 pandemic (June 11–29, 2020)”, respectively.

2-11 On page 15 and page 16 when you discuss the PVD results, it would help to rephrase the results to keep the order of the timepoints constant, e.g., say score A was higher pre-pandemic compared to mid-pandemic, and score B was lower pre-pandemic compared to mid-pandemic. A fast or not-careful reader might misinterpret the results to indicate that both scales were higher in the pre-pandemic timepoint (that was my first impression on a quick reading) because they are just scanning for the words “higher” or “lower”. I was especially confused by this in the wording on page 16, 2nd paragraph of discussion.

Reply: In response to Reviewer 2’s valuable recommendation, we have maintained consistency in the order of timepoints.

2-12 Page 15, lines 6-8 – please phrase the results of the equivalence tests as suggested in the TOSTER paper, and report the smallest effect size of interest that you tested for (i.e., the lower and upper bounds of equivalence that the TOST tested against) for each test. The TOST statistics are not meaningful without reporting that information. I would also request that you interpret/explain what these results mean for your claims, for readers who are not familiar with equivalence tests– in this case, if both the experimental test and the equivalence test have $p > .05$, then it suggests your data cannot conclusively say either that there *is* or *is no* meaningful effect, either way.

Reply: We have added to the ‘Results’ section, the lower and upper bounds of equivalence for each equivalence test (i.e., ± 0.5) based on the study by Lakens et al. (2018). As stated in our response 2-4, we had mistakenly set an incorrect equivalence bounds value in the previous version of our manuscript. We have now reanalysed the data based on the correct values (the lower and upper bounds of equivalence = ± 0.5) and found that changes in the IP-scores were significantly equivalent in the reminder conditions. In the revised manuscript’s ‘Discussion’ section, we have discussed our findings on the effects of highlighting self-identity based on our reanalyses.

2-13 **Discussion:**

Page 16, Discussion (first paragraph) –

- Instead of “reduce” intentions/behaviors, perhaps “affect” would be better as it is more general (you expected the intervention might reduce some intentions/behaviors and increase others)
- The second sentence is confusing and not quite right. Firstly, you tested changes in behavior or intention scores, not the scores themselves. Secondly, the wording makes it sound like you were testing between behavior and intention scores, rather than each separately.
- In sentence 3, after you mention there is no statistical support for Hyp 1 and 2; to fully report your results, you should mention that there is also no statistical support AGAINST Hyp 1 and 2 (as per the non-significant equivalence tests)

Reply: Reviewer 2’s suggestions were most helpful. In accordance with these suggestions, we have improved the ‘Discussion’ section. Specifically, we have rephrased “reduce intentions ~” to “affect intentions ~”. Moreover, the second sentence has been replaced with “~ there was no difference between the reminder conditions in both the IP-

intention and IP-behaviour scores”. In addition, after sentence 3, we have added ‘the post-hoc equivalence tests showed that there were significant equivalences between the reminder conditions in both the IP-intention and IP-behaviour scores, indicating that there was no support against either Hypothesis 1 or Hypothesis 2’.

2-14 Page 16, paragraph 3 – instead of launching straight into possible *theoretical* explanations for lack of conclusive results, I would first outline or emphasize the limitations of *evidence* in this study – that you did not find conclusive evidence in either direction, and so the effect size may be smaller than you guessed, and/or the data may be noisier than you had hoped. I think it is important to acknowledge this first, before speculating about theoretical or methodological concerns.

Reply: As we mentioned in our reply to 2-4, we reanalysed the data and found that the effect of the reminders was significantly equivalent among the reminder conditions (i.e. we obtained conclusive evidence). These suggestions were very helpful for our claim, and we, therefore, first outlined and emphasised the limitations of the present study in the revised manuscript.

2-15 I really struggled to understand the 4th paragraph of the Discussion, page 16-17.

– The paragraph seems to argue two opposite conclusions – that according to PVD results people might have reduced infectious behavior (fear of infection) but also increased it (lower perceived susceptibility). Are you suggesting both could be true at the same time? I don’t see how that could be the case.

Reply: Reviewer 2 seems to disagree with the PVD results which are understandable because we also did not expect this tendency. However, our obtained results indicate that during the COVID-19 pandemic perceived infectability was lower and germ aversion was higher in comparison with the pre-pandemic ones. While we reconsidered our obtained results based on the reviewers’ comments, at this point of time there is insufficient empirical evidence to specify the causes of the different directions of the PVD subscales, possibly since unknown effects mediate in our results. In the revised manuscript, we added these issues.

– And why would that explain why you got the results you did? The first argument (people already reduced high-risk infectious behaviors, as suggested by germ aversion on the PVD) is consistent with the point that you might have ceiling effects. But shouldn’t the second argument (that people might underestimate potential to be a spreader) be working against any ceiling effects of anti-spreading behavior/intentions?

Reply: As rightly pointed out by Reviewer 2, the second argument could confuse readers. We have, therefore, deleted it.

– You wrote “The present study’s data of IP-intention score may tend to be close to a ceiling effect, and therefore germ aversion may influence IP intention rather than behaviour.” Why should a ceiling effect on intention scores indicate that germ aversion had an effect on it? And why should it indicate that germ aversion did NOT affect behavior? This seems like pure speculation. You do not have data for IP intentions or behavior from the previous PVD data collection, so you don’t know how changes in intentions or behavior might be related to germ aversion PVD levels.

Reply: As pointed out by Reviewer 2, the above issue is just our speculation. However, our speculation is based on the data of the mean scores of IP intention and behaviour in each wave. Specifically, as compared to the mean scores of IP behaviour, those for IP intention were very high and near the scale’s maximum value (i.e. 7). Considering this

as well as germ aversion's higher scores, germ aversion could influence IP intention more strongly than IP behaviour. However, in the absence of pilot data; this idea is just a speculation. We have revised our manuscript to clarify these points.

- I found the second half of the paragraph hard to follow as well – I don't understand how the logic leads you to your conclusions.

Reply: Thanks to Reviewer 2's valuable suggestions, we substantially revised this paragraph. We believe that our revised manuscript has become easy to follow.

- Perhaps it would be better to just outline a simpler explanation/speculation in this paragraph --that the ceiling effects left little room for differences in intention, and that the levels of attitudes around disease (as measured by PVD) might have an unknown effect on the results (and any further interpretation would be speculation).

Reply: According to these comments, we have discussed our findings using simpler explanations/speculations in the revised manuscript. At this point of time, it is difficult to specify the factor causing the present results from the point of view of attitudes toward the disease and most of our interpretations are speculations. Based on all the comments, we have substantially revised the entire 'Discussion' section.

- 2-16** Page 17, paragraph 2—I'm not sure I understand the first sentence, "Moreover, the results of IP may reflect the degree of effect of highlighting self identity." Do you mean that you may have gotten inconclusive results because the true effect size is very small? If so, I would just state that!

Reply: After careful consideration, we deleted the paragraph containing this sentence in the revised manuscript.

- 2-17** Page 18, paragraph 2 (limitations)

- please rethink/rephrase your conclusions about the study's findings. You wrote "However, this indicates that at least the procedures in our study could not change IP-intentions and behaviour." However, the results of the equivalence tests combined with the hypothesis tests indicate that you cannot rule out either outcome – the data are inconclusive. You could instead state that your data could not conclusively show that the procedures changed intentions or behavior. (Or perhaps this sentence could simply be omitted and the paragraph focus solely on the limitations.)

Reply: As we replied in 2-4, the equivalence test indicates that the effect of the reminders is significantly equivalent among the reminder conditions. Therefore, we believe that we are now able to argue that the procedures in our study could not change IP intentions and behaviours. However, we added as one of the limitations that the interpretation of the equivalence test should be done with caution because setting the value of equivalence bounds was a deviation in the revised manuscript.

- instead of just considering internal validity (what methodological changes might result in a conclusive effect), what about also considering external validity – how well do the study stimuli reflect the way citizens might encounter such messaging "in real life" if a government or other entity decided to adopt the self identity messaging? The current study design could potentially be considered a merit rather than a limitation in some ways, as perhaps it reflects the passive messaging and sometimes long time lags between message and behavior that you might find in a modest messaging campaign. Likewise, regarding the potential effects you discuss of increasing the frequency of exposure to messaging—instead of just speculating that it might increase likelihood of

observing effects, it would be more interesting to me as a reader to consider how it relates to real world situations (e.g., the lab experiment you describe might provide information on how a well-funded, centralized and co-ordinated messaging campaign might affect responses.)

Reply: As Reviewer 2 mentioned, discussions from the point of view of a real message campaign might be helpful. The present study showed that highlighting self-identity by manipulating linguistic expressions had no impact on IP intention and behaviour. Although both messages in our stimuli are general and governments often use similar message, our manipulation would not contribute to a real messaging campaign based on our findings. Based on this admission, it might be beneficial to consider the effectiveness of reminders for encouraging IP intention and behaviour in a real messaging campaign. In the revised manuscript, we proposed two possible and simple ways to meet this purpose (e.g., repetitive presentation and self-generation of reminder messages). We added these issues to the revised manuscript. For your information, we did not discuss the issues regarding the lags between the message and behaviour here because we have discussed this point in another paragraph of the Discussion section.

- in the limitations section, I would also suggest highlighting other ways the study methodology might be improved. By “improved,” I do not mean better chances of finding a significant effect, but rather a better reflection of whether the effect exists in the real world. What changes might allow you to make more definitive claims about the hypotheses you are testing? For instance, validating and refining the measures might have helped make the data less noisy, or having a better manipulation check might have allowed you to test whether people actually absorbed the message.

Reply: We agree to Reviewer 2’s suggestions. We added these issues in the Limitation section of the revised manuscript.

- Lastly, I would suggest that the authors also discuss the limitation that the ‘manipulation check’ cannot actually ascertain whether participants read the message, and the possibility that many participants simply did not read it.

Reply: We also incorporated Reviewer 2’s suggestions on ‘manipulation check’ to the ‘Limitation’ section.

2-18 Page 18, last paragraph – again, I do not follow the logic of this paragraph or understand the argument it is attempting to make. Could this be phrased more clearly and directly? For instance, I wasn’t sure if the low case numbers were meant as an argument that people were already following safe behaviors, or that low numbers meant people perceived low risk and therefore the messaging simply didn’t work at all. I also am unclear what is meant by “If the threat of COVID-19 spread increases, people can easily follow the reminders.” However these were not the only issues with the paragraph -- I’m sorry, I simply didn’t understand any of the points or how they follow on from each other.

Reply: In this paragraph, we discussed whether there was a possibility of the timing of the survey mediating our results. Our claims are listed below. As mentioned in the first version of our manuscript, the number of cases were relatively low during our survey. Thus, it is unclear whether the obtained results would be true in the event of the COVID-19 situation becoming severe. Based on this view, further investigations relating to severe situations would be warranted to confirm whether slight differences in linguistic expressions generally have no effect on our IP intentions and behaviours. As Reviewer

2 pointed out, our descriptions deviated from our claims in the previous version of our manuscript. In the revised manuscript, we clarified these points.

2-19 Page 19, paragraph 1 – again, I am having trouble following the argument of this paragraph. Is the argument that self identity effects have been found in many cultures? Or that they have failed to be found in many cultures? I don't follow the logic of the last several sentences either. Could this be rewritten more clearly and simply, perhaps just stating what evidence there already is about whether self- identity effects are cross cultural.

Reply: In this paragraph, we had intended to discuss the speculation regarding the effect of highlighting of self-identity on IP intention and behaviour in countries other than Japan. While the present study collected data only in Japan; the effects of highlighting self-identity have been found in America and Israel (Bryan et al., 2013; Savir & Gamliel, 2019). Based on these previous findings, the effects of highlighting of self-identity on IP intention and behaviour might occur in these countries. Perhaps, the null effects of highlighting of self-identity on IP intention and behaviour in the present study are due to potential factors peculiar to Japan, and related to the severity of COVID-19, the quality of the medical system, and so on. Further investigations in countries other than Japan would confirm this speculation. However, this argument is unclear and confusing to the reader. In the revised manuscript, we deleted this paragraph and substituted it through a brief discussion of this point in the Limitation section.

2-20 Page 19, paragraph 2 – again, the data do not quite support this conclusion, that the manipulation “has little effect on changing IP intention and behaviour.” Instead, the data simply do not provide support for (or against) the hypotheses

Reply: We found significant equivalence among the reminder conditions in our reanalyses (for details, please refer to 2-4). Now, we can argue that highlighting self-identity has little effect on changing IP intention and behaviour.

2-21 Minor points:

I recommend that, in the codebooks (description of data text files), the authors should add in information on the allowable range of responses (1-7 I believe?) and indicate what those numerical responses corresponded to (e.g., 1=strongly disagree, 7 = strongly agree). This would be very helpful but is not strictly required.

Reply: We have added this information to the codebooks.

2-22 Page 6, top paragraph – the wording is a bit unclear. I suggest something like “As a secondary hypothesis for this research, we additionally predicted that the scores ... [etc.]”

Reply: We have modified the text as per Reviewer 2's suggestion.

2-23 Page 12, line 29 (Main Analysis) – could the first sentence clarify that the 2 mean scores for each measure are the before and after (or time 1 and time 2) measures? I did not understand this at first and had to read it a few times before realizing what it must mean.

Reply: According to this comment, we have clarified having calculated the mean scores of each of IP intention and behaviour in each wave.

2-24 Page 12, line 39 – the last word of the paragraph should be variables, plural. You have written variable (singular), which makes it sound like they have been combined – but I

gather from the paragraph below that you actually did not combine, but ran separate tests for each.

Reply: We have modified the wording based on Reviewer 2's suggestion.

2-25 Page 14, line 17-18 – I don't understand what this sentence means: "However, the total number of the participants excluded by each criterion did not match the total number of excluded participants." Could you briefly explain more what you meant, and why this is important?

Reply: We thought that some of the participants would infringe on more than one criterion of exclusion and therefore, the sum of the exclusions due to each criterion did not match with the data that we did not use in the statistical analyses. Since Reviewer 2 pointed out that this sentence was unnecessary and confusing to the reader, we deleted it and only reported the number of data used in the analyses.

2-26 Page 14, line 27 – I think the PVD results should be listed last in this section (after the intentions and behaviors results) as it is a secondary hypothesis. Put the most important, central results first.

Reply: Following Reviewer 2's recommendation, we shifted the PVD results to the end of the 'Results' section.

2-27 When referring to the PVD in text (e.g., page 17 line 47, but elsewhere as well), instead of "current situation" it may be useful to say "during pandemic" or "several months into the COVID-19 pandemic", or even "June 2020" as it is unclear when exactly "current situation" refers to (and will be even harder for readers in future years to pinpoint).

Reply: We reworded 'current situation' to "during the COVID-19 pandemic".

Responses to Reviewer 3

**Comments & Replies**

3-1 -Whether the data are able to test the authors' proposed hypotheses by passing the approved outcome-neutral criteria (such as absence of floor and ceiling effects or success of positive controls)

The check of comparing pre-PVD to current PVD was done. There may be ceiling effects, as the authors discuss at length. I am not sure there are ceiling effects as opposed to the manipulation just not being effective, however.

Reply: At this point of time, we should refrain from concluding that ceiling effects mediated in the present results. As Reviewer 3 mentioned, there might have been no ceiling effect, but our manipulation was ineffective. We had mentioned in the previous version of the manuscript that the IP intention and behaviour scores of were very high as, and that these results were consistent with ceiling effects. Considering that raising any possible and potential factors related to the results could be informative for the readers, we have discussed the possibility of mediation of ceiling effect in the present results. In the revised manuscript, mediation of ceiling effect in the present results was described as one of speculation in the 'Discussion' section.

3-2 - Whether the Introduction, rationale and stated hypotheses are the same as the approved Stage 1 submission

They appear to be consistent

Whether the authors adhered precisely to the registered experimental procedures

They appear to have adhered well

Where applicable, whether any unregistered exploratory statistical analyses are justified, methodologically sound, and informative

This appears consistent with the rest of the manuscript

Reply: We are grateful to Reviewer 3 for carefully reviewing our manuscript to ensure that it meets the criteria for stage 2 of the registered report.

3-3 - Whether the authors' conclusions are justified given the data

When I reviewed this manuscript as a pre-registered report, the authors insisted that pre-registered reports should only test the direct hypotheses, and thus they should not include additional measures to try to answer why the primary manipulation failed. Now the primary manipulation failed, and the authors are trying to piece together why. However, I am not sure that much of the speculation is warranted given the paper was only designed to answer the primary hypothesis. The discussion is a bit long, accordingly.

There are a few paragraphs where lack of measurement of relevant variables seem to play a major role in the speculation - "In our speculation, cultural differences may cause this gap between findings"

Reply: As we mentioned in our reply to 3-1, we believe that discussing any possible and potential factors related to our null effect could be informative for interpretation and subsequent studies. Since some explanations were redundant, it made the entire 'Discussion' section difficult to understand. In line with the reviewers' valuable comments, we substantially revised the Discussion section which has now become more useful and easy to understand.

3-4 - "The present study has several limitations on generalizing the findings"

I think most of these caveats in this section are fine and reasonable. Maybe the other speculations should be listed as limitations if they are unanswerable given the current data.

Reply: After careful consideration, we decided not to add any further speculation. As Reviewer 3 mentioned at 3-3, we have already indulged in a lot of speculations and further speculations will confuse the readers.

3-5 - As a side note, while I missed a revision of the pre-registered part of the report, it seems that basically nothing was done to assess the psychometrics of the scale, and whether it makes sense to take means across all of the items in these scales.

Reply: We were interested in whether the differences in the reminders affected the compliance with this IP-scale guidance issued by the Japanese government. Thus, we did not assess the psychometrics of the scale itself. In addition, we also used the means across all items in our analysis because other reviewers had indicated that the items should be taken as means. However, based on Reviewer 3's suggestion, we have conducted a post-hoc

analyses of the data. The results showed that there were no differences in the effect of reminders in all the items. We uploaded the R code of these results to the OSF.

3-6 - There are many writing issues in the new text (discussion, results, etc). I am not going to list them all, but the writing should be clearer prior to publishing.

Reply: According to this valuable suggestion, we had a native speaker of English to proofread our manuscript again.

3-7 - Abstract-
"Practise"
Is misspelled

Reply: We have used British English in our paper based on the guidelines of the Royal Society Open Science, wherein the verb form is 'practise' and 'practice' is the noun.

3-8 - Discussion-
"the reminder hardly changes IP-intention"
Informal, and hardly is not really the result. The result is does not change IP-intention or behavior

Reply: We have modified the expression to “the reminder does not change IP-intention or behaviour.”

3-9 - "Given the small nature of the effect size, it is quite likely that the effect of highlighting self-identity was not observed depending on experimental settings such as the present"
Difficult to parse

Reply: After careful consideration, we deleted this sentence in the revised manuscript.

3-10 - "performed meta-analysis"
Performed a

Reply: As we mentioned in our reply to 3-9, we deleted the paragraph containing this sentence.

3-11 -"These findings show that it is not only Japanese people who experience little effect of highlighting self-identity."
This sentence seems to say the opposite of what is intended, as it suggests that Japanese people and other groups are unlikely to show an effect of highlighting self-identity. I think it means to say there is evidence that they do show such an effect.

Reply: As Reviewer 3 has pointed out, this sentence states the opposite of what we want to convey. According to the other reviewer’s comment (2-19), we changed the entire paragraph substantially, and deleted this sentence.

Responses to Reviewer 5

Comments & Replies

5-1 I have read the stage 1 version of this registered report and was very curious to read about the outcome of this research project and the authors' conclusions. Overall, regarding all relevant aspects, the authors did comply to the registered procedures and I also could not detect deviations from the introduction or the at stage 1 registered hypotheses.

In general, I would recommend this paper for publication, however, there are several aspects which in my opinion would benefit from revision.

Reply: We sincerely thank Reviewer 5 for the positive feedback and suggestions on our manuscript. We have replied to Reviewer 5's comments.

5-2 TOST-Approach: I appreciate that the authors use this approach for being able to draw conclusions about the absence of an effect because that's usually not possible within a frequentist framework. However, as I think TOST generally is not a standard analysis, I would appreciate it if the authors could explain their approach here in more detail. Especially, as far as I have understood it, in the TOST it is necessary to define effect sizes as boundaries so that it's possible to reject effects outside this boundary. I wonder which effect sizes the authors used? Also, I don't know what the phrase "there was no significant equivalence" exactly means. What are the implications of this result? Again, I value that the authors used this approach, but I think the authors need to add more information about what these findings actually imply.

Reply: We thank Reviewer 5 for the positive comments on our statistical approach. As we mentioned in our replies to the other reviewer's comment (2-4), we did not register effect sizes for TOST. Moreover, there was a careless error in the code of the statistical analyses: Although we planned to adopt ± 0.5 as the boundaries, we inadvertently put the input as ± 0.05 . Considering these, we added the descriptions of the boundaries in the Deviations section. Moreover, we reanalysed the data in the case of setting ± 0.5 as the boundaries and found that the changes in IP-scores were statistically equivalent (i.e., significantly equivalent) among the reminder conditions. This result more strongly supported no differences of the change in IP-scores among the reminder conditions. We revised the Results and Discussion sections based on this reanalyses.

5-3 PVD-Scale: The two subscales of the PVD-Scale point in different directions (although it should be highlighted that the effect of the germ aversion scale is really small, $d < 0.1$). In their hypothesis concerning the PVD-scale, it read to me as the authors would assume coherent effects for the two subscales, so it seems a rather unusual result that they diverge. To me the conclusion of these findings is as follows: Participants felt more susceptible to infections before the outbreak of COVID-19, but now they experience more discomfort in situations with potentially high germ-exposure (although again: to a very small extent). I'm aware that the authors attempt to find explanations for this oddity on p. 16/17, as they argue (if I understand it correctly) that because people experience more germ aversion, they avoid certain events and thus they feel less susceptible to infections disease as before. That's valid reasoning, but I think it does not explain why the effect size of perceived infectability is larger than germ aversion. I think there might be some more processes involved, and I think it would be valuable to further elaborate on that. I'd also be curious about the exact statistics of the two scales (means, SDs, inter-items correlations) as well as the exact wording of the items in the appendix since I think that warrants a better assessment of the results.

Reply: We agree to Reviewer 5's comment regarding poor explanations of different directions of the two subscales of the PVD. Reviewer 2 also pointed out that the logic in this paragraph was unclear. After we carefully considered both comments, we realised that we should avoid specifying causes of divergence in the PVD subscales because there is insufficient empirical evidence discuss these issues. Thus, based on this admission, we stated in the revised manuscript that further investigations would be necessary to specify causes of divergence in the PVD subscales. We have reported means, SDs, and inter-item correlations of the two scales in results, under the heading *The PVD scores during the COVID-19 pandemic and prior to its onset*. We also added the Japanese

versions of the IP-intention, IP-behaviour, and PVD scales to the OSF datasets (<https://osf.io/dc7rs/files/>).

5-4 p. 17: I don't really understand the reasoning about the role of effect sizes in prior studies. It is about effect sizes or context effects? I agree that it's important to consider not only p-values but effect sizes, but what exactly is the link between effect sizes in prior studies and the results of the present study?

Reply: We thought that the effect sizes in the previous empirical studies and the meta-analysis (Bryan et al., 2013; Guo et al., 2020; Savir & Gamliel, 2019) would be helpful for considering why the reminder had no effect in the present study. If the reported effect sizes were large, the phenomenon is expected to steadily and robustly occur. Conversely, the small effect size indicates that the effect possibly has plasticity. From this perspective, we discussed the previous and present findings together. As a result of reviewing the previous studies, we found that the effect sizes were usually small (Bryan et al., 2013: did not report η^2 ; Guo et al., 2020: $f = 0.123$; Savir & Gamliel, 2019: $d = 0.17$). This tendency is consistent with the results of the meta-analysis (Savir & Gamliel, 2019). Based on these, we considered that the effect of highlighting self-identity possibly has plasticity and might not be robust enough to be observed in any real-life situations. However, as mentioned in our reply 3-9, we deleted this speculation. Instead, we have added these statements as limitations to paragraph 3 in the 'Discussion' section.

5-5 I think a very likely explanation for the results on the IP intention is the existence of ceiling effects, which the authors also adequately discuss. Irrespective of that the authors could not detect effects of highlighting self-identity, I find the results worth publishing as they can provide important insights for future studies. Especially because the study was conducted in Japan and cultural factors most likely influence health-related behavior, the study can make an important contribution to the understanding of health-related behavior.

Reply: We greatly appreciate Reviewer 5's positive comments and detailed review of our paper.

We once again express our sincere gratitude to the editor and the reviewers for their prompt, thoughtful, and constructive comments. We hope that our revised manuscript is now suitable for publication as a Registered Report in the *Royal Society Open Science*.

Sincerely,

Fumiya Yonemitsu
Kyushu University
744 Motoooka, Nishi-ku, Fukuoka 819-0395, Japan
Phone/Fax number: ++81-92-642-2418
E-mails: y.fumiya.0408@gmail.com